# Stress granules plug and stabilize damaged endolysosomal membranes

Claudio Bussi[1 ✉], Agustín Mangiarotti[2], Christian Vanhille-Campos[3,4], Beren Aylan[1,6], Enrica Pellegrino[1,6], Natalia Athanasiadi[1], Antony Fearns[1], Angela Rodgers[1], Titus M. Franzmann[5], Anđela Šarić[3], Rumiana Dimova[2] & Maximiliano G. Gutierrez[1 ✉]

Endomembrane damage represents a form of stress that is detrimental for eukaryotic cells[1,2]. To cope with this threat, cells possess mechanisms that repair the damage and restore cellular homeostasis[3–7]. Endomembrane damage also results in organelle instability and the mechanisms by which cells stabilize damaged endomembranes to enable membrane repair remains unknown. Here, by combining in vitro and in cellulo studies with computational modelling we uncover a biological function for stress granules whereby these biomolecular condensates form rapidly at endomembrane damage sites and act as a plug that stabilizes the ruptured membrane. Functionally, we demonstrate that stress granule formation and membrane stabilization enable efficient repair of damaged endolysosomes, through both ESCRT (endosomal sorting complex required for transport)-dependent and independent mechanisms. We also show that blocking stress granule formation in human macrophages creates a permissive environment for *Mycobacterium tuberculosis*, a human pathogen that exploits endomembrane damage to survive within the host.

Stress granules are condensates of non-translating messenger ribonucleoproteins (mRNPs) that originate from mRNAs stalled in translation initiation via a network of interactions involving the RNA binding protein G3BP1 and its homologue G3BP2[8–10] (collectively referred to as G3BP). In addition to mRNPs, these cytoplasmic condensates contain a broad and heterogenous group of proteins whose identity varies in a context and cell type specific manner[11,12]. Despite many advances in our understanding of the molecular mechanisms driving stress granule formation, the biological function of these biomolecular condensates remains largely unknown[12]. Membrane-bound organelle–biomolecular condensate interactions have been reported to have a role in cellular organization[13,14]. Stress granules and other biomolecular condensates are considered to be membraneless organelles, but they also associate with lipid membranes. Therefore, membraneless and membrane-bound condensates probably correspond to different functional states[15–17]. Nevertheless, our understanding of the mechanisms and functional implications of biomolecular condensate–membrane interactions is still limited[18].

Lysosomal damage is a common feature of many diseases, and poses major challenge in maintaining cellular health. Despite its importance, the identification of a mechanism that can rapidly stabilize the ruptured lysosomal membrane, preventing leakage and enabling effective repair, remains unknown. Emerging evidence indicates that G3BP proteins can associate with the lysosomal membrane under homeostatic conditions[19,20]. In addition to the broad range of stimuli that trigger stress granule formation, lysosomal damage also induces stress granule formation[21,22]. However, it remains unclear whether this response is primarily caused by the damaged membrane itself or whether stress granule formation is spatially restricted to the site of damage. Of note, the biological function of stress granules in the context of endolysosomal membrane damage (that is, loss of membrane integrity through rupture or poration) remains unknown.

Physiological roles for limited endolysosomal damage have been described in the context of chromosome segregation, antigen presentation and host–pathogen interactions[23–25]. Upon membrane damage, pores form, making the vesicles unstable and prone to collapse if the ruptured area cannot be sealed[26]. Given that this process is lethal, cellular repair mechanisms act rapidly to restore membrane integrity, avoiding extensive membrane damage, cytosolic protease leakage and cell death. Limited endolysosomal damage can be repaired through the ESCRT-dependent and independent pathways[3,4,6,7], but the mechanisms that first stabilize the damaged membrane to enable the recruitment of the repair machinery are poorly understood.

Here we investigated the role of biomolecular condensates and stress granules in response to endomembrane damage induced by chemical and physical agents, and by infection with *M. tuberculosis* (Mtb) in human macrophages. Our findings revealed that stress granules nucleate in the proximity of damaged endolysosomes, providing a protective plug that stabilizes the ruptured membrane and promotes efficient repair. Notably, this process is critical for the repair of Mtb phagosomes and contributes to the containment of this human pathogen, which induces endomembrane damage. Our study reveals a biophysical mechanism underlying the stabilization of damaged membranes and uncovers a biological function for stress granules in the context of endolysosomal damage.

[1]The Francis Crick Institute, London, UK. [2]Max Planck Institute of Colloids and Interfaces, Potsdam, Germany. [3]Institute of Science and Technology Austria, Klosterneuburg, Austria. [4]Department of Physics and Astronomy, Institute for the Physics of Living Systems, University College London, London, UK. [5]Center for Molecular and Cellular Bioengineering, Biotechnology Center, Technische Universität Dresden, Dresden, Germany. [6]These authors contributed equally: Beren Aylan, Enrica Pellegrino. ✉e-mail: claudio.bussi@crick.ac.uk; max.g@crick.ac.uk

## Stress granules condense on damaged endolysosomes

We first investigated the dynamics of stress granule–endomembrane damage interactions in human induced pluripotent stem cell-derived macrophages (iPSDMs) using superresolution live-cell imaging. After inducing endolysosome damage with L-leucyl-L-leucine methyl ester (LLOMe), a widely used lysosomotropic compound that forms a membranolytic polymer[27,28], we observed a marked increase in G3BP1-positive granules that localized in the proximity of lysosomes positive for galectin-3 (GAL-3), a cytosolic lectin that binds to lysosomal glycans that are exposed only after membrane damage[28] (Fig. 1a and Supplementary Videos 1 and 2). We found that approximately 90% of the GAL-3-positive events were associated with G3BP1-positive structures and formed rapidly after adding LLOMe, reaching a maximum after 20 min of treatment (Fig. 1b–d and Supplementary Videos 1 and 2). In addition, we confirmed that stress granule formation at lysosomal damage sites occurred at a higher rate than expected by random chance by conducting spatial point pattern analysis (Fig. 1e). A 3D image analysis showed that in more than 70% of the G3BP1 and GAL-3-positive events, the granules formed a plug pattern closely associated with the damaged endolysosome from where later the condensate continued to grow (Fig. 1f,g and Supplementary Videos 3–8). At a higher temporal resolution (around 1 s), we determined that G3BP1-positive structures and their accumulation preceded the detection of endomembrane damage by GAL-3 (Fig. 1h,i and Supplementary Video 8).

G3BP1-positive granules triggered by LLOMe treatment were positive for bona fide stress granule markers[29] such as poly(A)-RNA, EIF3B, EIF4G, G3BP2, TIA1 and PABPC1[8–11] (Extended Data Fig. 1a–g). The response to LLOMe was different from the response to the stress granule inducer sodium arsenite[30] (NaAsO$_2$), since NaAsO$_2$ did not trigger endomembrane damage (Extended Data Fig. 2a,b). Moreover, LLOMe treatment rapidly increased the phosphorylation of the eukaryotic translation initiation factor 2α subunit (eIF2α) (Extended Data Fig. 2c,d), which regulates canonical stress granule formation[21,31]. In line with these results, the protein synthesis inhibitors cycloheximide and emetine[32] blocked LLOMe-induced stress granule formation without affecting lysosomal damage induction (Extended Data Fig. 2e,f).

We then investigated the effect of blocking lysosomal acidification using bafilomycin A1 (BAFA1), an inhibitor of the vacuolar ATPase. Given that BAFA1 could interfere with the lysosomal processing of LLOMe[27], we tested its effect on stress granule formation induced by silica crystals (Extended Data Fig. 2g), another endomembrane-disrupting agent[28,33]. BAFA1 significantly decreased stress granule formation after inducing lysosomal damage with silica crystals (Extended Data Fig. 2g,h), suggesting that rapid and localized changes in the cytosolic pH after limited lysosomal leakage could trigger condensate formation, as previously reported with G3BP1 condensates in vitro[10] and in other systems[34]. Distinct types of interactions between G3BP1-positive stress granules and LAMP1-positive endolysosomal membranes were observed after endomembrane damage, when stress granules were found both in the proximity of vesicular membrane and in the lumen or associated to intraluminal vesicles (Extended Data Fig. 2i–k). Consistent with these results, live-cell imaging and 3D surface-rendering analysis of iPSDMs expressing LAMP1–eGFP showed G3BP1 foci adjacent to membrane disruption sites, where it further accumulated into LAMP1-positive vesicles (Fig. 1j,k). Together, these results indicate that stress granules are formed rapidly and plug endolysosomal damage sites.

## Condensates stabilize damaged membranes

To confirm that protein condensation can occur directly at the damage site, we used giant unilamellar vesicles (GUVs) and G3BP1–RNA condensates as a model system. First, we reconstituted G3BP1–RNA granules in vitro as previously described[10] (Extended Data Fig. 3a–d).

Then, we generated GUVs encapsulating G3BP1 as a homogeneous solution (50 mM HEPES, 100 mM NaCl, 4 mM MgCl$_2$, pH 7.5) and used a microfluidic device for a rapid and complete exchange of the external milieu (Extended Data Fig. 3e,f). To mimic the difference in pH found between the cytosolic and endolysosomal compartments in cellulo, we exchanged the pH of the external solution from 7.5 to 5 (Methods). The membrane damage was induced via a hypotonic shock, generating nano- to micro-sized pores in the GUVs, allowing localized mixing of the exterior and interior solutions (Fig. 2a–d, Extended Data Fig. 3g–i and Supplementary Video 9). Membrane poration and phase separation of G3BP1 was triggered by exchanging the external solution to one of lower osmolarity and pH and containing poly(A)-RNA (20 mM HEPES, 100 mM NaCl, 4 mM MgCl$_2$, 200 ng μl$^{-1}$ poly(A)-RNA, pH 5). Notably, G3BP1–RNA condensates formed at membrane damage sites triggering wetting and stabilization of the rim of the pore, and preventing membrane collapse and further content leakage (Fig. 2a–g, Extended Data Fig. 3g–i and Supplementary Videos 9 and 10). These observations were consistent with the pattern observed in cells (Fig. 1), which preserved vesicle integrity. By contrast, if pore formation occurred when condensates were not able to form, the vesicles collapsed (Fig. 2e–g). Membrane wetting[15–17] by the condensate was observed (Fig. 2b,c and Extended Data Fig. 3h,n) ensuring local droplet spreading and stabilization of the pore, effectively immobilizing the ruptured membrane. Moreover, the percentage of GUVs that survived the hypotonic shock was higher when condensates were formed, compared to when there was no phase separation (Fig. 2g).

Given that we are proposing a function for biomolecular condensates as stabilizers of membrane damage, we tested whether these results could be extended to other proteins that form condensates. We used glycinin, a plant storage protein that constitutes a robust model system for condensate formation and whose interactions with membranes have been described recently[17]. We triggered condensation by externally decreasing pH or increasing salinity and vesicle poration (Extended Data Fig. 3j–o and Supplementary Video 11). Glycinin experiments further confirmed that condensate formation can occur selectively at membrane disruption sites, plugging membrane pores.

We next analysed the physical mechanism of plugging using a coarse-grained molecular dynamics model and observed that condensates formed spontaneously at the site of membrane damage owing to mixing of the solutions in the inner (reflecting the endosomal lumen) and outer (reflecting the cytosolic side) compartments (Fig. 2h–j). The rapid nucleation of these condensates resulted in a marked drop of the flux across the pore and prevented further mixing of the two solutions (Fig. 2i and Extended Data Fig. 4a,b). Similar to the pattern observed in cells (Supplementary Videos 3–8), by tuning the protein–membrane interaction, we found that membrane wetting by the condensate was required to stabilize the membrane. As observed in vitro, wetting provided the favourable interaction energy required to drive passive pore sealing (Fig. 2h–j) and engulfment of the droplet after membrane wrapping (Supplementary Video 12). In the absence of wetting, the droplets disrupted the pore as they were able to grow to saturation (Supplementary Video 13), but most of the plugging effect was retained, with the condensate acting as a sink for the solutes that would otherwise mix. Finally, removing all condensate-forming interactions from the system resulted in extensive mixing of the two solutions and long-lasting membrane damage (Supplementary Video 14). In addition, we found that efficient plugging (maximum inhibition of the mixing fluxes) requires fast droplet nucleation at the damage site, which is determined by the concentration of its components (Extended Data Fig. 4a,b). We further evaluate the spatial distribution of droplet nucleation and growth over time. In line with the in cellulo results, we observed that condensate droplets initially form in the vicinity of the pore and that most of the condensate material remains in this region as the plug

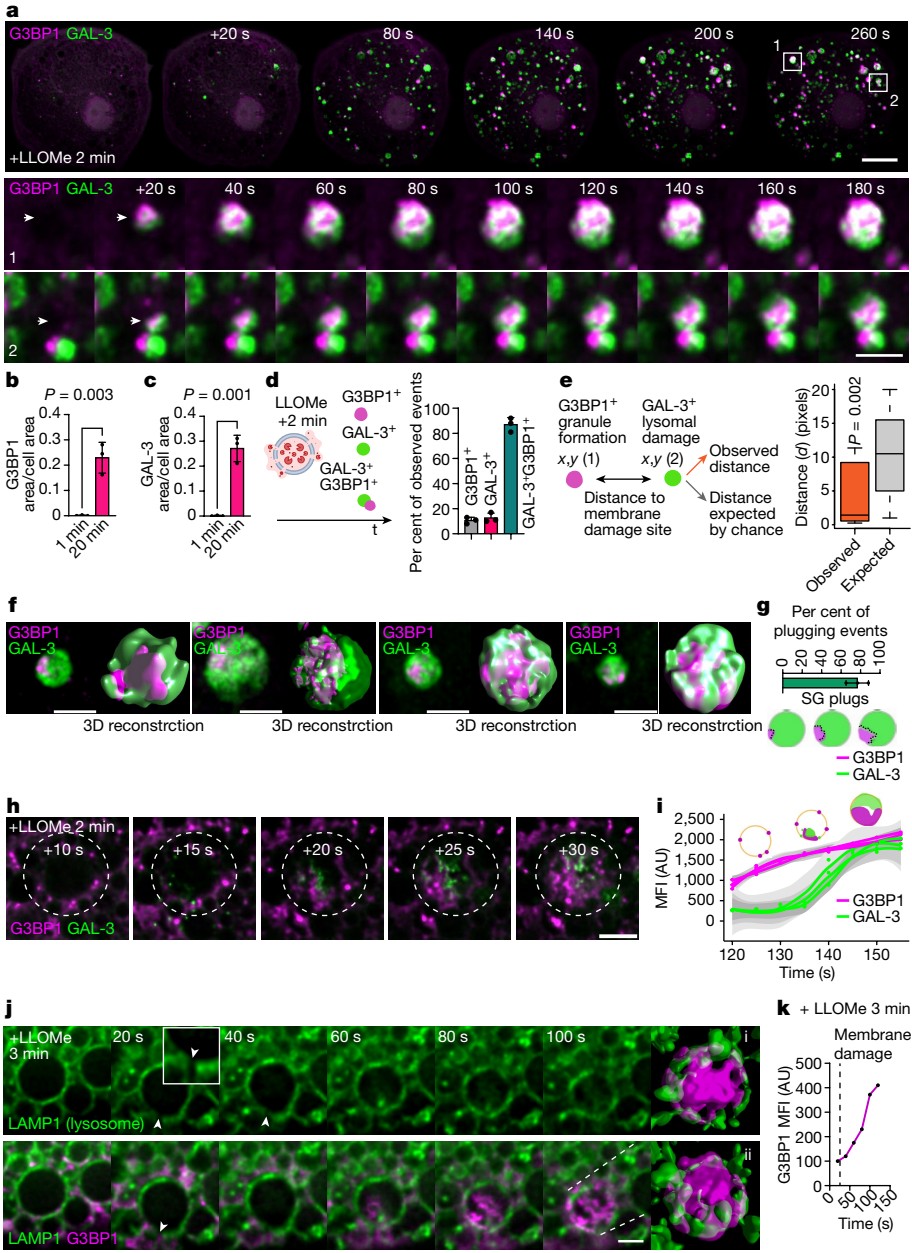

**Fig. 1 | Stress granules form in the proximity of endomembrane damage sites. a**, Live-cell imaging sequence of iPSDMs expressing G3BP1–GFP and GAL-3–RFP after 2 min of LLOMe treatment (1 mM). Outlined regions in the top row are magnified in the bottom two rows. Arrowheads highlight the time frame before and after a G3BP1/GAL-3 event is detected. **b,c**, Quantification of area of G3BP1 (**b**) and GAL-3 (**c**) puncta normalized to cell area from live-cell imaging experiments as described in **a**. *n* = 30 cells examined over 3 independent experiments; two-tailed *t*-test. **d**, Scheme summarizing the different type of events observed in the first 20 min after inducing lysosomal damage. Graph shows the percentage of single G3BP1⁺ events, GAL-3⁺ events or combined GAL-3⁺G3BP1⁺ events observed by live-cell imaging. **e**, Spatial point pattern analysis applied to G3BP1⁺ and GAL-3⁺ events. In box plots, the centre line is the median, boxes delineate the interquartile range (IQR), and whiskers represent the data range within 1.5 times the IQR. *n* = 40 events examined over 3 independent experiments; *P* value by Monte Carlo simulation-based approach. **f**, G3BP1 polarized 'plug pattern' on GAL-3⁺ damaged vesicles and the corresponding 3D *z*-stack reconstruction. **g**, Quantification and schematic of plugging events shown in **f**. *n* = 40 events examined over 3 independent experiments. SG, stress granule. **h**, Image sequence (1 s frame rate, shown at 5 s intervals) of iPSDMs expressing G3BP1–GFP (magenta) and GAL-3–RFP (green) after 2 min of LLOMe treatment. **i**, Mean fluorescence intensity (MFI) over time of G3BP1⁺ and GAL-3⁺ events as shown in **h**. Graphs show traces over time and standard error. *n* = 30 events examined over 3 independent experiments. AU, arbitrary units. **j**, Live-cell imaging sequence of iPSDMs expressing LAMP1–RFP and G3BP1–GFP. Arrowhead indicates membrane rupture site. The panel on the far right shows a 3D reconstruction. **k**, MFI of G3BP1–GFP over time for sequence shown in **j**. Bar plots indicate mean ± s.e.m. of at least three independent experiments. Scale bars: 10 µm (**a**, top), 2 µm (**a**, middle, bottom, **f,h,j**).

grows (Extended Data Fig. 5a,b). Together, these results show that upon membrane pore generation, interaction of the solutions inside and outside of the lumen triggers local biomolecular condensate formation, which in turn stabilizes the damaged membrane and promotes vesicle survival.

## Stress granules facilitate endomembrane repair

To investigate whether stress granule-mediated membrane stabilization was functionally linked to endomembrane repair, we targeted G3BP1 and G3BP2 in iPSDMs using CRISPR–Cas9 delivered as

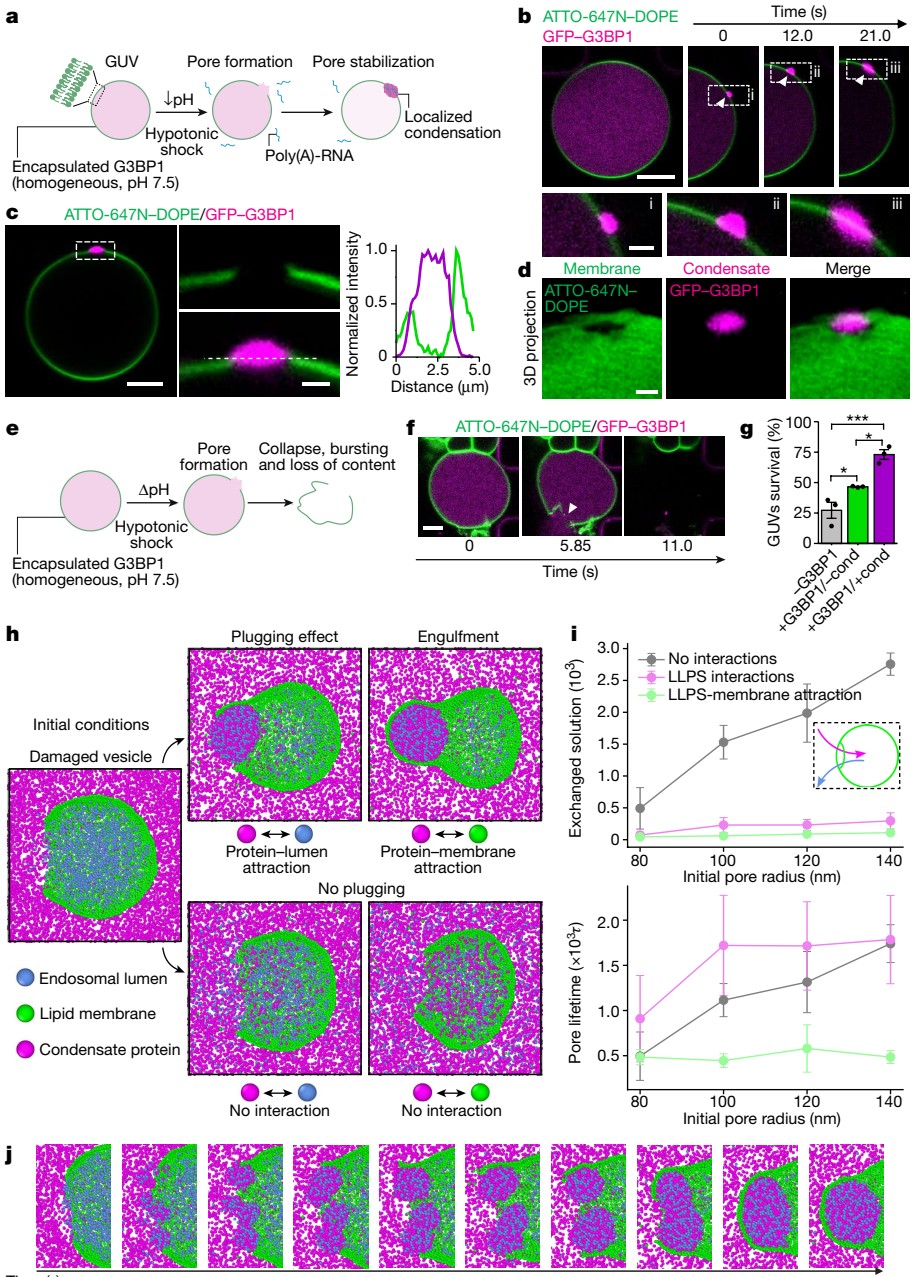

**Fig. 2 | Localized condensate formation stabilizes damaged membranes.**
**a**, Experimental design. Lipid vesicles (green) encapsulating G3BP1 (magenta) are exposed to a hypotonic shock and pH decrease in the presence of poly-(A) RNA in the external medium. GUV, giant unilamellar vesicles. **b**, Confocal time-lapse. A damaged vesicle (green) stabilized by G3BP1–RNA condensate. White arrowhead marks condensate formation and growth. Scale bars: 5 µm (top) and 2 µm (bottom). **c**, Left, final state of GUV condensate from **b** (scale bar, 5 µm). Middle, magnified view showing membrane discontinuity and G3BP1–RNA condensate at the pore (scale bar, 2 µm). Right, intensity profile shows that the condensate wets the pore rim. **d**, 3D projection of pore region from **c**. Scale bar, 2 µm. **e**, Without the triggering of condensation, membrane damage causes bursting and content loss. **f**, Vesicle destabilization sequence. Bursting upon poration is followed in the absence of RNA. White arrowhead indicates pore and content loss. Scale bar, 10 µm. **g**, GUV survival under hypotonic shock. GUVs without encapsulated protein (−G3BP1), with protein (+G3BP1/−cond), and with protein and triggering of condensate formation with poly(A)-RNA (+G3BP1/+cond). Data are mean ± s.e.m. $n = 3$ independent experiments. One-way ANOVA. **h**, Simulated damaged model vesicle (green) with condensation-prone protein solution outside and solute particles inside, resulting in different outcomes based on the interactions between these three components. **i**, Droplet plug formation. Exchanged solution (particles) between vesicle inside and outside (top) and pore lifetime (bottom) with different pore radii, for three interaction scenarios. No interactions, increased solution exchange with pore size and no droplets; Upon condensation (liquid-liquid phase separation (LLPS) interactions), droplets plug exchange and pores are long-lived; Condensate-membrane (LLPS–membrane attraction), droplet wets vesicle, efficient plug and accelerated pore closure. Data are mean ± s.d. $n = 10$ replicas. **j**, Droplet formation sequence. Mixing two solutions leads to rapid small-droplet formation near the pore. Over time, these merge into one large condensate, plugging the pore and inducing sealing with wetting interactions. *$P < 0.05$, ***$P < 0.001$.

a ribonucleoprotein (RNP) complex by nucleofection[35] and achieved near population-level genetic knockout (Extended Data Fig. 6a,b). Given that LLOMe is processed into a membranolytic polymer primarily

dependent on lysosomal protease activity[27], we first evaluated the lysosomal volume and proteolytic activity in iPSDMs subjected to RNP nucleofection (G3BP$^{nf}$ iPSDMs). There were no differences in the

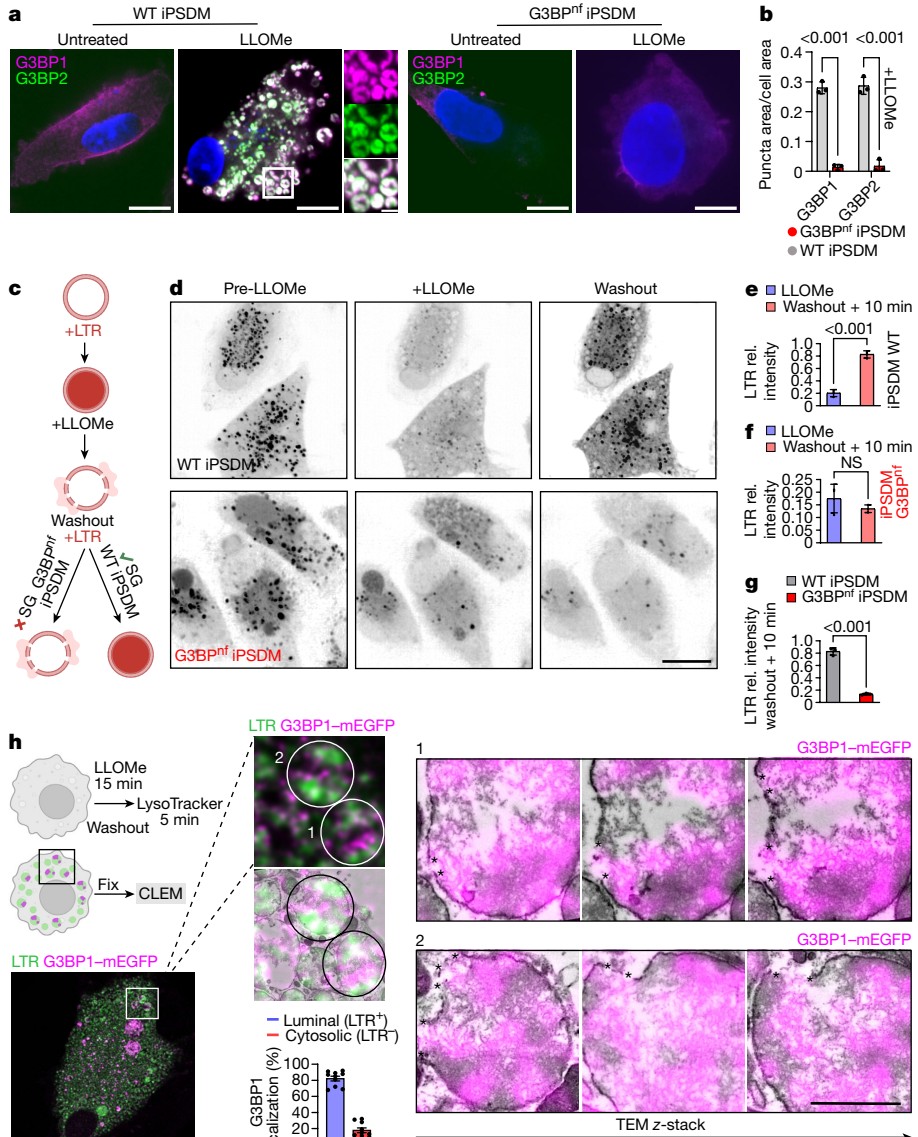

**Fig. 3 | Stress granules facilitate endomembrane repair. a**, G3BP1 and G3BP2 staining in wild-type and G3BP^nf iPSDMs left untreated or treated with LLOMe (1 mM, 30 min). $n = 3$ independent experiments. Scale bars: 10 μm (main images), 2 μm (enlarged area). **b**, Stress granule quantification in iPSDMs treated as described in **a**. $n = 30$ cells examined over 3 independent experiments. Two-way ANOVA with Šídák's multiple comparisons test. **c**, Schematic illustrating the lysosomal recovery assay using LysoTracker (LTR). **d**, Image sequence of wild-type and G3BP^nf iPSDMs incubated with LysoTracker (black puncta) before adding LLOMe (left), and 2 min (middle) or 10 min after (right) washout. Scale bar, 10 μm. **e,f**, Intensity of LysoTracker puncta relative (rel.) to basal values (pre-LLOMe) for the indicated conditions in wild-type (**e**) and G3BP^nf (**f**) iPSDMs. $n = 3$ independent experiments. Two-tailed $t$-test. **g**, LysoTracker intensity at 10 min after washout in wild-type iPSDMs compared with G3BP^nf iPSDMs. Data are mean ± s.e.m. $n = 3$ independent experiments. Two-tailed $t$-test. **h**, Schematic of the CLEM experiment on recovered lysosomes. mEGFP–G3BP1 (magenta) iPSDMs were treated with LLOMe (1 mM) for 15 min. After a washout step, cells were incubated with LysoTracker (green) before fixation and CLEM processing. Bottom right, percentage of G3BP1^+ puncta localizing to LysoTracker^+ and LysoTracker^- areas, quantified by electron microscopy. $n ≥ 10$ fields of view over 3 independent experiments. The image sequence shows the G3BP1^+LysoTracker^+ regions of interest (1 and 2) analysed by CLEM and the corresponding fluorescence–electron microscopy image overlap across three different regions in $z$ (serial stack). Arrowheads indicate G3BP1^+ areas surrounding membrane disruption sites. NS, not significant.

lysosomal volume and proteolytic activity of G3BP^nf iPSDMs compared with wild-type iPSDMs (Extended Data Fig. 6c–e). In agreement with previous reports showing that G3BP is a central node in the regulation of stress granule formation[8–10,36,37], we observed that G3BP^nf iPSDMs did not form stress granules after inducing lysosomal damage (Fig. 3a,b and and Extended Data Fig. 6f). Single-cell high-content imaging of the lysosomotropic dye LysoTracker after lysosomal damage was used as a probe to monitor lysosomal integrity dynamics[38]. There were no differences in the fluorescence intensity of LysoTracker between untreated G3BP^nf iPSDMs and control iPSDMs, indicating that the absence of G3BP does not affect lysosomal pH in basal conditions (Extended Data Fig. 6g). In addition, damaged endolysosomes in G3BP^nf iPSDMs and control iPSDMs exhibited similar leakage of LysoTracker after 2 min of LLOMe treatment, suggesting that there was no difference in the extent of lysosomal damage induction after LLOMe (Fig. 3c–g). However, G3BP^nf iPSDMs did not recover the lysosomal population after removal of LLOMe (Fig. 3c–g). Similar results were observed in G3BP1 and G3BP2 (G3BP1/2)-nucleofected human monocyte-derived macrophages (HMDM G3BP^nf) and in HeLa cells transfected with short interfering RNA (siRNA) targeting G3BP1/2 (Extended Data Fig. 6h–p). To visualize stress

granules in cellulo at the ultrastructural level, we performed correlative light–electron microscopy (CLEM) studies. By analysing serial transmission electron microscopy sections of the recovered G3BP1-positive and LysoTracker-positive vesicles after lysosomal damage, we observed that—at the timepoint evaluated—most of the G3BP1-positive signal was associated with aggregates that localized to LysoTracker-positive vesicles and G3BP1-positive regions enriched in the areas of endomembrane disruption (Fig. 3h). These results are consistent with our observations from live-cell imaging and the plug-engulfment model that emerged from the computational simulations.

To further validate that blocking stress granule formation impairs lysosomal repair, we used previously characterized G3BP1 and G3BP2 double-knockout U2OS (G3BP-DKO) cells[37] (Extended Data Fig. 7a–c). First, we observed similar G3BP1–GAL-3 dynamics after inducing lysosomal damage in wild-type U2OS cells (Extended Data Fig. 7d–f) and found no differences in the lysosomal proteolytic activity or basal LysoTracker intensity levels compared to the G3BP-DKO cells (Extended Data Fig. 7g). Notably, the lysosomal population in G3BP-DKO U2OS cells did not recover after LLOMe removal (Extended Data Fig. 7h–l and and Supplementary Video 15). We confirmed these results using 10K-dextran particles, which—unlike LysoTracker—accumulates in lysosomes in a pH-independent manner[3] (Extended Data Fig. 7m–q).

Lysosomal damage has been shown to be repaired by ESCRT-dependent[7,38] as well as ESCRT-independent pathways[4] such as PI4K2A[5,6] and annexin A1 and A2[3] (ANXA1/A2)-mediated mechanisms. However, in cases of extensive damage, a coordinated system mediated by GAL-3 and autophagy adapters[1,39] such as TBK1 and p62 targets these damaged lysosomes to the autophagy pathway[1,40,41]. Consistent with the proposed membrane-stabilizing role for stress granules, we observed rapid accumulation of ESCRT-III components CHMP2A, CHMP4B and the CHMP4A-interacting protein ALIX in G3BP-DKO U2OS cells after inducing lysosomal damage (Extended Data Fig. 8a,b). Notably, while the involvement of GAL-3 and autophagy adapters have been reported at later time points (30 min or beyond) following LLOMe treatment in wild-type U2OS cells[7] and other cell lines[39], G3BP-DKO U2OS cells displayed accumulation of these proteins as early as 5 min after lysosomal damage (Extended Data Fig. 8c–f). Similar results were observed with the recruitment of ESCRT-independent components such as ANXA1/A2[3] (Extended Data Fig. 9a–d) and PIK42A–ORP9[5,6] (Extended Data Fig. 10a–d). Consistent with our in vitro and in silico observations, our results indicate that the inability to form a plug after lysosomal damage increases the extent of damage and lysosomal membrane instability. Even though the recruitment of the repair machinery is increased, the absence of a stress granule-stabilized membrane renders the process inefficient. Consequently, damaged lysosomes accumulate and become targeted for degradation through the autophagy pathway[2]. Collectively, these findings indicate that stress granules stabilize damaged endomembrane and contribute to efficient endomembrane repair in cellulo.

## Stress granule formation restricts Mtb infection

Mtb induces membrane damage in macrophages via the action of the ESX-1 type VII secretion system and other factors[42]. In an RNA-sequencing study of Mtb-infected iPSDM[42], we identified a significant increase in the transcript levels of several stress granule markers after infection with wild-type Mtb but not the ESX-1-deficient Mtb ΔRD1 mutant, which is severely restricted in its ability to induce endomembrane damage[33,42,43] (Fig. 4a). Notably, *ZNFX1*—a gene associated with Mendelian susceptibility to mycobacterial disease or tuberculosis with intermittent monocytosis, and that has been previously identified in stress granules[44]—was also upregulated in a RD1-dependent manner (Fig. 4a). In agreement with a membrane damage-dependent phenotype, infection with wild-type Mtb but not with Mtb ΔRD1 increased the levels of phosphorylated eIF2α (p-eIF2α) (Fig. 4b,c). In human macrophages, wild-type Mtb triggered stress granule formation in

the proximity of bacteria that are inducing membrane damage, as visualized by GAL-3 immunostaining (Fig. 4d–g). Moreover, we found that ZNFX1 presented a high degree of co-localization with G3BP1 and localized to sites of membrane damage in close proximity to wild-type Mtb (Fig. 4f–h). Notably, when we investigated whether stress granules are formed during Mtb infection in a mouse model of Mtb infection, we found that stress granule- and GAL-3-positive structures were present in the lungs of C3H/HeJ mice infected with wild-type Mtb—but not in those infected with Mtb ΔRD1 (Fig. 4i–k). To identify functional outcomes of stress granule formation after Mtb infection, we analysed Mtb replication in control and G3BP[nf] iPSDMs. We found increased replication of wild-type Mtb after 48 h of infection, and replication of both wild-type Mtb and Mtb ΔRD1 in G3BP[nf] iPSDMs increased after 72 h of infection (Fig. 4l,m). The difference in kinetics observed between wild-type Mtb and Mtb ΔRD1 is in agreement with data showing that although Mtb ΔRD1 exhibits a significantly diminished capacity to localize in the cytosol, a certain degree of endomembrane damage can still occur through the bacterial lipid phthiocerol dimycocerosate[43]. Collectively, these results suggest that an impairment in endolysosomal membrane stabilization and repair, driven by the blockade of stress granule formation, critically affects the outcome of Mtb infection in macrophages.

## Discussion

By combining in vitro, in silico and in cellulo studies, we uncovered a function for biomolecular condensates whereby stress granules form selectively in the proximity of damaged membrane as a plug that facilitates vesicle stabilization and survival. Stress granules condense rapidly in the vicinity of endolysosomal damage sites, enabling endomembrane stabilization and repair, proving pivotal for the macrophage host defence against intracellular pathogens that damage endomembranes. Our in vitro results showing that other biomolecular condensates can also stabilize and plug ruptured membranes suggest that that this could be an evolutionarily conserved mechanism, considering similar observations for fungal septal pore-clogging proteins that aggregate during wound healing in *Neurospora crassa*[45].

We propose a model in which stress granules nucleate at the pore location after local changes in milieu conditions (for example, decrease in pH[10]) and mixing of endolysosomal and cytosolic contents. The phase-separated droplets rapidly grow to span the whole pore, plugging the mixing of fluxes, such that they display different sizes depending on the size of the pore (Fig. 2h,j and Extended Data Fig. 4). Furthermore, wetting interactions between the condensate and the membrane stabilize both the pore and the droplet, enhancing the plug effect. From a biophysical perspective, a beneficial energy contribution from this wetting interaction counteracts the destabilizing forces acting on the vesicle and can drive pore sealing and engulfment of the condensate by wrapping of the membrane around the droplet. This is more likely to happen in membranes that are subject to small lateral stress or possess some area reservoir, whereas only partial engulfment would be expected if the membrane is under tension. Therefore, engulfment of the condensate could be mediated by lipid addition to the membrane—as previously described[6]—even though the contribution of this pathway remains to be investigated. Finally, complete engulfment of the droplet could lead to budding of the membrane-wrapped condensate, resulting in complete membrane repair. This could happen spontaneously (as observed in simulations for strong condensate–membrane interactions; Extended Data Fig. 4) or could be mediated by a specialized mechanism such as the ESCRT machinery in a subsequent step. Stress granules are thought to have a role in the protection of cells from stress, but they can also contribute to the aggregation of misfolded proteins[46]. Our findings revealing a role of stress granules in stabilizing damaged membranes may contribute to the identification of molecular targets for the treatment of neurodegenerative diseases, such as synucleinopathies and

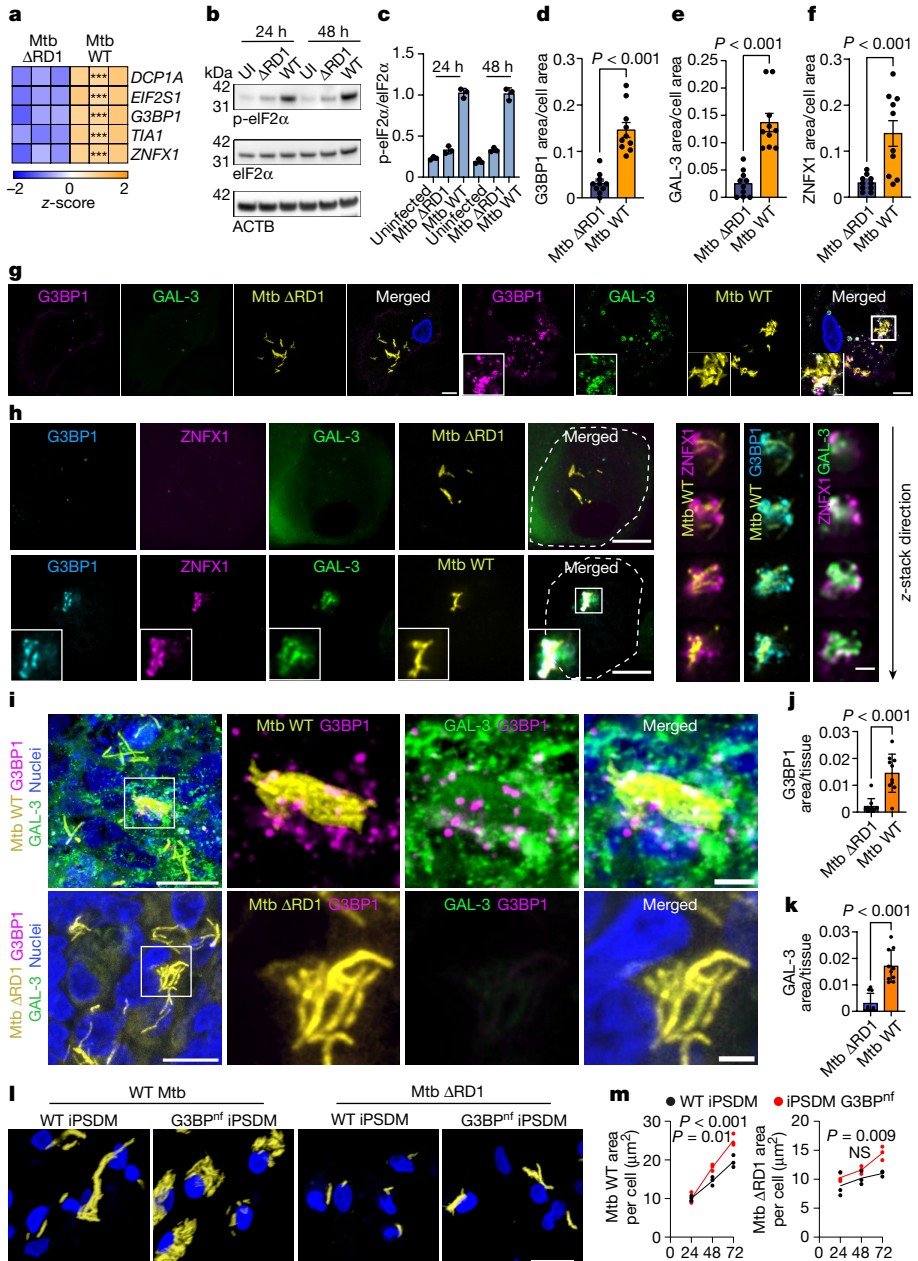

**Fig. 4 | Stress granules are required for Mtb restriction in human macrophages. a**, Heat map of *z*-scores for transcripts associated with stress granules in iPSDMs infected with wild-type Mtb or Mtb ΔRD1 (multiplicity of infection (MOI) 2, 48 h). *P* value was calculated using Wald statistic and adjusted for false discovery rate (α = 0.05, ***$P < 0.001$). **b**, Immunoblot for phosphorylated and total eIF2α in iPSDMs that are uninfected (UI) or infected with wild-type Mtb or Mtb ΔRD1 for 24 or 48 h (MOI, 2). Beta actin (ACTB) was used as a loading control. **c**, eIF2α protein levels relative to ACTB. Data are mean ± s.e.m. *n* = 3 independent experiments. **d**–**f**, Area of G3BP1 (**d**), GAL-3 (**e**) and ZNFX1 (**f**) puncta normalized to cell area of iPSDMs infected with wild-type Mtb and Mtb ΔRD1 (MOI 2, 48 h). Data are mean ± s.e.m. of at least 10 cells from one out of 3 independent experiments (*n* = 3). Two-tailed *t*-test. **g**, Representative images of iPSDMs infected with Mtb (MOI 2, 48 h) and stained for G3BP1 and GAL-3. **h**, Left, representative images of iPSDMs infected with Mtb (MOI 2, 48 h) and stained for G3BP1, ZNFX1 and GAL-3. Right, *z*-stack images of the outlined area. **i**, Immunofluorescence images Mtb, G3BP1 and GAL-3 in lung from mice infected with wild-type Mtb and Mtb ΔRD1. **j**,**k**, Area of G3BP1 (**j**) and GAL-3 (**k**) puncta normalized to tissue area. Data are mean ± s.e.m. *n* = 10 areas analysed from at least 3 tissue sections. Two-tailed *t*-test. Images shown are *z*-stack projections. **l**, Images of wild-type and G3BP[nf] iPSDMs infected with wild-type Mtb and Mtb ΔRD1 for 72 h (MOI, 1). **m**, Quantification of Mtb replication (bacteria area per cell) in wild-type and G3BP[nf] iPSDMs at 24, 48 and 72 h after infection. *n* = 3 independent experiments. Two-way ANOVA with Šídák's multiple comparisons test.

tauopathies[47], in which lysosomal damage has been shown to be triggered by abnormal protein aggregates[40,48,49].

G3BP1-dependent stress granule formation seems to be critical for repairing Mtb phagosomes and the control of infection and provides a mechanistic explanation for clinical evidence showing that patients with inherited deficiency of the stress granule protein ZNFX1 exhibit impaired immunity to mycobacteria[44]. Together, our results reveal a physical mechanism by which spontaneous stress granule condensation at the site of membrane damage mediates vesicle stabilization and subsequent repair. Our observations could be relevant for diseases associated with lysosomal membrane damage such as neurodegenerative disorders, infection and cancer[1].

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

## Methods

### Animal infections

Mice were bred and housed in specific pathogen-free facilities at The Francis Crick Institute. All protocols for breeding and experiments were approved by the Home Office (UK) under project license P4D8F6075 and performed in accordance with the Animal Scientific Procedures Act, 1986.

### Plasmids

All DNA constructs were produced using *Escherichia coli* DH5a (Thermo Fisher Scientific) and extracted using a plasmid midiprep kit from Qiagen. The plasmids used in this study were: RFP-GAL-3[50], LAMP1-RFP (Addgene, 1817) and G3BP1-mEGFP (Addgene, 135997).

### Cells

**iPS cell and iPSDM culture.** KOLF2 human induced pluripotent stem (iPS cells were sourced from Public Health England Culture Collections (catalogue number 77650100). mEGFP-G3BP1 human iPS cells (used for CLEM studies) were sourced from Coriell Institute (catalogue number AICS-0082 cl.1). iPS cells were maintained in Vitronectin XF (StemCell Technologies) coated plates with E8 medium (Thermo Fisher Scientific). Cells were authenticated by STR profiling upon receipt and are checked monthly for Mycoplasma contamination by PCR. Cells were passaged 1:6 once at 70% confluency using Versene (Gibco). Monocyte factories were set up following a previously reported protocol[42]. In brief, a single-cell suspension of iPS cells was produced with TryplE (Gibco) at 37 °C for 5 min and resuspended in E8 plus 10 μM Y-27632 (Stem Cell Technologies) and seeded into AggreWell 800 plates (Stem-Cell Technologies) with $4 \times 10^6$ cells per well and centrifuged at 100*g* for 3 min. The forming embryonic bodies were fed daily with two 50% medium changes with E8 supplemented with 50 ng ml⁻¹ human BMP4 (Peprotech), 50 ng ml⁻¹ human VEGF (Peprotech) and 20 ng ml⁻¹ human SCF (Peprotech) for 3 days. On day 4, the embryonic bodies were collected by flushing out of the well with gentle pipetting and filtered through an inverted 40-μm cell strainer. Embryonic bodies were seeded at 100–150 embryonic bodies per T175 or 250–300 per T225 flask in factory medium consisting of X-VIVO15 (Lonza) supplemented with Glutamax (Gibco), 50 μM β-mercaptoethanol (Gibco), 100 ng ml⁻¹ hM-CSF (Peprotech) and 25 ng ml⁻¹ human IL-3 (Peprotech). These monocyte factories were fed weekly with factory medium for 5 weeks until plentiful monocytes were observed in the supernatant. Up to 50% of the supernatant was collected weekly and factories fed with 10–20 ml factory medium. The supernatant was centrifuged at 300*g* for 5 min and cells resuspended in X-VIVO15 supplemented with Glutamax and 100 ng ml⁻¹ human M-CSF and plated at $4 \times 10^6$ cells per 10-cm petri dish to differentiate over 7 days. On day 4, a 50% medium change was performed. To detach cells, iPSDM plates were washed once with PBS then incubated with Versene for 15 min at 37 °C and 5% CO₂ before diluting 1:3 with PBS and gently scraping. Macrophages were centrifuged at 300*g* and plated for experiments in X-VIVO15.

**U2OS and HeLa cells.** U2OS and HeLa cells were cultured using standard conditions. Both cell lines were maintained in Dulbecco's Modified Eagle Medium (DMEM) supplemented with 10% fetal bovine serum (FBS). The cells were incubated at 37 °C in a humidified atmosphere containing 5% CO₂. U2OS wild-type and G3BP1/2 KO U2OS cell lines were from the laboratory of P. Anderson[37]. Cells were negative for mycoplasma contamination (checked by PCR).

### iPSDM electroporation

Plasmid DNA was electroporated into iPSDM using the Neon system (Invitrogen). iPSDM were resuspended at $1.5 \times 10^6$ cells in 100 μl buffer R. Ten microlitres of cells per 1 μg plasmid DNA mix was aspirated into a Neon pipette and electroporated in electroporation buffer E at 1,500 V

for 30 ms with 1 pulse. Cells were then plated in ViewPlate glass bottom 96 well plates (6005430, PerkinElmer) for high-content analysis or in IBIDI μ-Slide 18-well glass bottom coverslips (81817) for confocal imaging studies.

### iPSDM nucleofection

iPSDM were washed twice with PBS and electroporated in the appropriate primary nucleofection solution (AmaxaTM Human Monocyte Nucleofector Kit, VPA-1007) using the Lonza 2b Nucleofector (Nucleofector 2b Device, AAB-1001). Five million iPSDMs were used per reaction and resuspended in 100 μl of primary nucleofection solution containing 4 μg of Cas9 from *Streptococcus pyogenes* (SpCas9) (IDT) mixed with a total of 12 μg of targeting synthetic chemically modified single guide RNA (sgRNA) (Synthego) (outlined below). iPSDMs were then nucleofected with two sgRNAs targeting the *G3BP1* genes and two sgRNAs targeting the gene *G3BP2*, and the Cas9–RNP mix using the Y001 program. Nucleofected cells were cultured in pre-warmed X-VIVO15 in a 35 mm Sterilin petri dish plate (121 V, Thermo Scientific). At 2 h post nucleofection, 100 ng ml⁻¹ human M-CSF was added to the cells. Dishes were incubated in a humidified 37 °C incubator with 5% CO₂. After 3 days, an equal volume of fresh complete media including 100 ng ml⁻¹ human M-CSF was added. Six days after the initial isolation, differentiated macrophages were detached in Versene and plated for experiments in X-VIVO15.

sgRNAs used were: G3BP2_g1: CGCCCTACAAGCAGCGGACT; G3BP2_g2: AAGCTCCGGAATATTTACAC; G3BP1_g1: CGCCCGACCAGCAGGGGACT; and G3BP1_g2: AGGCCCCAGACATGCTGCAT.

### siRNA transfection of HeLa cells

HeLa cells were cultured in Dulbecco's Modified Eagle Medium (DMEM) supplemented with 10% FBS. For short interfering RNA (siRNA) transfection, SMARTpool siRNA targeting *G3BP1* and *G3BP2* (obtained from Dharmacon) were diluted in Opti-MEM medium, and DharmaFECT transfection reagent was mixed with the siRNA to achieve a final concentration of 25 nM. The siRNA transfection complex was incubated for 5–10 min at room temperature. HeLa cells were plated at a density allowing for 50–70% confluency at the time of transfection. The siRNA transfection complex was then added to the cells plated in a 6 well plate (1.5 ml final volume) and incubated for 24 h before repeating the same protocol for an extra day (two rounds of transfection). After 24 h, cells were collected for subsequent experiments. As a control siRNA, a Silencer Select Negative Control No. 2 siRNA (4390846, Thermo Fischer) was used following the same protocol.

The targeting siRNAs were as follows:

ON-TARGETplus SMARTpool siRNA J-012099-06, *G3BP1* target sequence GUGGUGGAGUUGCGCAUUA; ON-TARGETplus SMART-pool siRNA J-012099-07, *G3BP1* target sequence AGACAUAGCUCAGACAGUA; ON-TARGETplus SMARTpool siRNA J-012099-08, *G3BP1* target sequence GAAGGCGACCGACGAGAUA; ON-TARGETplus SMART-pool siRNA J-012099-09, *G3BP1* target sequence GCGAGAACAACGAAUAAAU; ON-TARGETplus SMARTpool siRNA J-015329-09, *G3BP2* target sequence UGAAUAAAGCUCCGGAAUA; ON-TARGETplus SMART-pool siRNA J-015329-10, *G3BP2* target sequence GAAUUUAAGUCUGGGACGA; ON-TARGETplus SMARTpool siRNA J-015329-11, *G3BP2* target sequence ACAACGACCUAGAGAACGA; ON-TARGETplus SMARTpool siRNA J-015329-12, *G3BP2* target sequence: GCGAUGGUCUUGACUAUUA.

### Human monocyte-derived primary macrophage isolation

White blood cells were isolated from leukocyte cones (NC24) supplied by the UK National Health Service blood and transplant service by centrifugation on Ficoll-Paque Premium (GE Healthcare 17-5442-03) for 60 min at 300*g*. Mononuclear cells were washed twice with MACS rinsing solution (Miltenyi 130-091-222) to remove platelets, then remaining red blood cells were lysed by incubation at room temperature with

10 mL RBC lysing buffer (Sigma R7757) per pellet for 10 min. Cells were washed with rinsing buffer and pelleted once more, then resuspended in 80 µl MACS rinsing solution supplemented with 1% BSA (MACS/BSA) and 20 µl anti-CD14 magnetic beads (Miltenyi 130-050-201) per $10^8$ cells and incubated on ice for 20 min. Cells were then washed in MACS/BSA by centrifugation, resuspended in 500 µl MACS/BSA per $10^8$ cells and passed through an LS column (Miltenyi 130-042-401) in the field of a QuadroMACS separator magnet (Miltenyi 130-090-976). The column was washed three times with MACS/BSA, then positively selected cells were eluted, centrifuged and resuspended in RPMI 1640 with GlutaMAX and HEPES (Gibco 72400-02), 10% heat-inactivated FBS and 50 ng ml$^{-1}$ hM-CSF (Preprotech) to a concentration of $10^6$ cells per ml in untreated petri dishes. These were placed in a humidified 37 °C incubator with 5% $CO_2$, with an equal volume of fresh media including human M-CSF added after 3 days. Six days after the initial isolation, differentiated macrophages were detached in 0.5 mM EDTA in ice-cold PBS and $5 \times 10^5$ cells seeded per well of a 12-well plate for western blot experiments.

## Mtb infection of macrophages
Wild-type Mtb H37Rv and Mtb H37Rv ΔRD1 were provided by D. Young and S. H. Wilson. Fluorescent Mtb strains were generated as previously reported[51]. E2-Crimson Mtb was generated by transformation with pTEC19 (Addgene 30178, deposited by L. Ramakrishnan). Strains were verified by sequencing and tested for phthiocerol dimycocerosate positivity by thin layer chromatography of lipid extracts from Mtb cultures. Mtb strains were cultured in Middlebrook 7H9 supplemented with 0.2% glycerol, 0.05% Tween-80 and 10% albumin dextrose catalase (ADC). For macrophage infections, Mtb was grown to an optical density ($OD_{600}$) ~ 0.8 then centrifuged at 2,000$g$ for 5 min. The pellet was washed twice with PBS, then the pellet was shaken with 2.5–3.5 mm glass beads for 1 min to produce a single-cell suspension. The bacteria were resuspended in 10 ml cell culture medium and centrifuged at 300$g$ for 5 min to remove clumps. The $OD_{600}$ was determined, and bacteria diluted to an appropriate OD for the required MOI—assuming $OD_{600} = 1$ equates to $10^8$ bacteria per ml—before adding to cells in a minimal volume. After 2 h, the inoculum was aspirated, cells washed twice with PBS and fresh culture medium added. Cells were then incubated for appropriate time points before collecting for analysis as described in the sections below.

## LLOMe treatment
A 333 mM stock of LLOMe (Cat# 4000725, Bachem) was prepared in methanol and frozen at −20 °C in tightly sealed tubes. For LLOMe treatment, the medium was replaced with X-VIVO15 (iPSDM), RPMI (HMDM) or DMEM (HeLa and U2OS cells) containing 1 mM of LLOMe unless otherwise indicated in the figure legend. Methanol (0.3%) in the respective cell culture media was used in all control conditions.

## Silica treatment
A solution of crystalline silica (MIN-U-SIL-15, US Silica) at 200 µg ml$^{-1}$ was prepared in X-VIVO15 and the cells were stimulated for 3 h, after which they were processed for downstream applications. In experiments with BAFA1 (Merck, B1793-10UG), iPSDM were pre-incubated for 1 h with a 100 nM solution. BAFA1 was kept in the medium during silica treatment.

## Western blotting
For lysis, cells were washed once with PBS, and lysed on ice in RIPA buffer (Millipore) containing complete, EDTA-free protease inhibitor (Roche). The samples were boiled at 95–100 °C for 5 min in LDS sample buffer and reducing agent (NuPAGE, Life Technologies) and run on a NuPAGE 4–12% Bis-Tris gel (Life Technologies). The gels were transferred onto a PVDF membrane using an iBlot 2 Dry Blotting System (Thermo Fischer), program P0. Membranes were blocked in 5% skimmed milk powder in PBS plus 0.05% Tween-20 (PBS-T) for 1 h at room temperature

then incubated with primary antibody overnight at 4 °C. Membranes were washed in PBS-T and incubated with HRP-conjugated secondary antibodies for 1 h at room temperature. Membranes were developed with enhanced chemiluminescence reagent (BioRad) and imaged on an Amersham Imager 680 instrument. Antibodies used: anti-G3BP1 (13057-2-AP) and anti-G3BP2 (16276-1-AP) from Proteintech, anti-p-eIF2α (Ser51) (9721), anti-eIF2α (9722) and anti-β-Actin (8H10D10, 12262) from Cell Signalling Technology; and HRP-conjugated anti-mouse (W4021) and anti-rabbit (W4011) antibodies from Promega. All antibodies were used at 1:1,000 dilution with the exception of HRP-conjugated antibodies that were used at 1:10,000 dilution.

## In vitro condensate–membrane experiments
**Materials.** 1,2-dioleoyl-*sn*-glycero-3-phosphocholine (DOPC) and 1,2-dioleoyl-*sn*-glycero-3-phospho-L-serine (DOPS) were purchased from Avanti Polar Lipids. The fluorescent lipid dye ATTO 647N-DOPE was acquired from ATTO-TEC. Polyvinyl alcohol (PVA, with MW 145000) and chloroform (HPLC grade, 99.8%) were purchased from Merck. Lipid stocks were mixed as chloroform solutions at 4 mM, containing 0.1 mol% dye, and were stored until use at −20 °C. Fluorescein isothiocyanate isomer (FITC), poly (A) RNA, 4-(2-hydroxyethyl)−1-piperazineethanesulfonic acid buffer (HEPES), sucrose, glucose, carbonate buffer tablets, hydrochloric acid (HCl), sodium hydroxide (NaOH), magnesium chloride ($MgCl_2$), dimethyl sulfoxide (DMSO) and sodium chloride (NaCl) were obtained from Sigma-Aldrich. All solutions were prepared using ultrapure water from a SG water purification system (Ultrapure Integra UV plus, SG Wasseraufbereitung) with a resistivity of 18.2 MΩ cm.

**Protein purification.** Preparation of glycinin was achieved as described[52]. In brief, defatted soy flour was dispersed in 15-fold water (weight) and pH was adjusted to 7.5 using a 2 M NaOH solution. Afterwards, the dispersion was centrifuged for 30 min at 9,000$g$ at 4 °C. Dry sodium bisulfite (SBS) was added to the supernatant (0.98 g l$^{-1}$ SBS), and the pH of the solution was adjusted to 6.4 with a solution of 2 M HCl and kept at 4 °C overnight. Then the dispersion was centrifuged during 30 min at 6,500$g$ at 4 °C. The obtained glycinin-rich precipitate was dispersed in fivefold water and the pH was adjusted to 7.4. The glycinin solution was then dialysed against Millipore water for two days at 4 °C and then freeze-dried to acquire the final product with a purity of 97.5%. For glycinin labelling, a 20 mg ml$^{-1}$ of protein solution was prepared in 0.1 M carbonate buffer (pH 9). A FITC solution (4 mg ml$^{-1}$ in DMSO) was slowly added into the protein solution with gentle stirring to a final concentration of 0.2 mg ml$^{-1}$. The sample was incubated in the dark while stirring at 23 °C for 3 h. The excess dye was removed using a PD-10 Sephadex G-25 desalting column (GE Healthcare), and buffer was exchanged for ultrapure water. The pH of the labelled protein solution was adjusted to 7.4 by adding 0.1 M NaOH. For fluorescence microscopy experiments, an aliquot of this solution was added to the working glycinin solution to a final concentration of 4%.

G3BP1 expression and purification were adapted from Guillén-Boixet et al.[10] with modifications. In brief, recombinant His$_6$−G3BP1−MBP and His$_6$−GFP−G3BP1−MBP were expressed in and purified from SF9 insect cells (Expression Systems, 94-001 F) using a baculovirus expression system[53]. Following cell lysis (LM10, Microfluidics) in buffer A (50 mM HEPES/NaOH (pH 7.5), 1 M NaCl, 5% (w/v) glycerol, 2 mM EDTA, 2 mM DTT) supplemented with 1× EDTA-free protease inhibitor cocktail (Roche), the lysate was cleared by centrifugation at 50,000$g$ for 50 min and subsequently the supernatant was applied to a custom casted HiFlow amylose column (New England Biolabs). The column was washed with 10 column volumes buffer A, followed by 10 column volumes buffer B (50 mM HEPES/NaOH, 100 mM NaCl, 5% gylcerol (w/v), 2 mM EDTA, 2 mM DTT, pH 7.5). 150 µl of 1 mg ml$^{-1}$ His-tagged 3 C protease (home-made) in buffer B were applied for on-column tag removal and incubated for 8 h at room temperature. The tag-free product

was washed out with buffer B, applied to a 5 ml HiTRAP Q HP column (Cytiva) equilibrated in buffer B and eluted with a linear gradient against buffer A. Fractions containing the target protein were applied to a HiLoad Superdex 200-pg 26/600 column (Cytiva) equilibrated in buffer C (50 mM HEPES/NaOH, 150 mM NaCl, 5% gylcerol (w/v), 2 mM EDTA, 2 mM DTT pH 7.5). G3BP1 fractions were pooled, concentrated, flash frozen with liquid nitrogen and stored at −80 °C.

**Vesicle preparation.** DOPC and DOPC:DOPS 9:1 GUVs containing 0.1 mol % ATTO 647N-DOPE were prepared by the PVA gel-assisted method[54] which allows protein encapsulation during vesicle swelling. In brief, 2 coverslips were cleaned with water and ethanol and dried with nitrogen. PVA solution was prepared by diluting PVA in deionized water at 60 °C to a final concentration of 40 mg ml[−1]. A small aliquot (20–50 µl) of the PVA solution was spread on the glass slides and dried for 1 h at 60 °C. Lipid stock solution (3–4 µl) were spread on the PVA-coated glasses and kept for 1 h under vacuum at room temperature. The chamber was assembled using the two coverslips and a 2-mm-thick Teflon spacer, and filled as follows according to the different experiments:

For G3BP1 encapsulation in GUVs, the swelling buffer consisted in 10 µM GFP–G3BP1 in 50 mM HEPES pH 7.5, 100 mM NaCl, 4 mM MgCl₂.

For glycinin encapsulation, the swelling solution contained 10 mg ml[−1] glycinin in 15 mM NaCl, 10 mM sucrose at pH 7.4 for the pH-triggered phase separation experiments. For inducing salt-driven phase separation the solution consisted of 300 mM sucrose at pH 7.4.

In all cases pH was adjusted using a 2 M NaOH solution. After 30 min, the chamber was opened, and the vesicles were collected carefully to prevent PVA detaching from the cover glasses. Osmolarities were measured and adjusted using a freezing point osmometer (Gonotec Osmomat 3000).

Encapsulation of the proteins was preferred over having the proteins in the external solution due to several experimental constraints. First, protein concentration is a limiting factor and having the protein inside the vesicles requires much less protein than having it in the external solution, which in the microfluidic experiments is under a constant flow. Second, if the proteins were in the external solution, the labelled proteins would make the complete field of view fluorescent, which would hinder visualization of the experiment and high-precision imaging. Finally, it is preferable to have the lower pH on the outside, because GUV formation leads to higher yield and better quality at neutral rather than acidic pH. It is important to note that the results are independent of the experimental choice for solution location (inside versus outside), since condensate formation is driven by the external and internal solutions mixing at the pore (as shown in Fig. 2 and Extended Data Fig. 4), and the outcome will not depend on the positional order of the solutions across the membrane. Furthermore, because of the large size of the GUVs, the membrane is almost flat with negligible curvature (lower than 0.1 µm[−1]) and not expected to have a role.

**Microfluidics and solution exchange.** The microfluidic device consists of a cascade GUV trapping system, which is described in detail elsewhere[55]. It was produced using poly(dimethylsiloxane) (PDMS) pre-cursor and curing agent (Sylgard 184, Dow Corning GmbH), at a mass ratio of 10:1. After polymerization at 80 °C for 3 h, inlet and outlet holes were punched with a biopsy punch with a plunger system (Kai Medical). Then the PDMS chips and glass coverslips were treated with plasma for 1 min using high-power expanded plasma cleaner (Harrick Plasma), and then bonded together. Before the experiments, the desired amount of solution was filled into the microfluidic device by centrifugation at 900 relative centrifugal force (Rotina 420 R, Hettich). Solution exchange was performed with a NeMESYS high-precision syringe pump. The flow speed for the initial washout of the remaining labelled protein in the exterior (Extended Data Fig. 4f,g) was set to 1 µl min[−1] for 40 min to ensure at least 10 times exchange of the

internal volume of the microfluidic device (~4 µl). To change the pH and introduce the hypotonic solutions, a higher speed of 2 µl min[−1] was used to ensure rapid exchange, to avoid phase separation due to passive proton permeation[56–58]. At this high flow, vesicles were pushed against the posts of the microfluidic device and small deformations could be observed (Fig. 2a). After complete solution exchange, the flow speed was lowered to 0.035 µl min[−1] to prevent vesicle movement and facilitate imaging.

The hypotonic buffer consisted in 20 mM HEPES, 100 mM NaCl, 4 mM MgCl₂, pH 5 for G3BP1 experiments. For glycinin pH-driven phase separation the external hypotonic solution was deionized water at pH 4.8, with the pH adjusted using a solution of 2 M HCl. For the experiments with phase separation triggered by NaCl, the hypotonic buffer consisted of a 100 mM NaCl solution, pH 7.4.

**In vitro G3BP1–RNA granules formation.** G3BP1–poly(A)-RNA condensates were reconstructed following Guillén-Boixet et al.[10]. To form the condensates, 20 µl of a solution containing a final concentration of 5 µM GFP–G3BP1 and 200 ng µl[−1] of poly(A)-RNA in 50 mM HEPES, 100 mM NaCl, 4 mM MgCl₂ was directly placed on a glass slide and immediately visualized by confocal microscopy.

## Molecular dynamics

To simulate poration and plugging of lipid vesicles, we consider a minimal coarse-grained molecular dynamics system where the solution of proteins in and out of the vesicle is described by beads dispersed in implicit solvent. The membrane is described as a one-bead-thick fluid elastic layer of mechanical properties that mimic biological membranes. We tune the interactions between solution beads in the different compartments to capture the differences between the cytoplasmic and endosomal protein solutions and explore how the system responds to different degrees of membrane poration. Details on how the molecular dynamics was performed are in the Supplementary Information and Supplementary Table 1.

## Molecular dynamics analysis

To characterize and understand the plugging effect of protein condensates on damaged vesicles we can track how the vesicle surface evolves and how the particles which are initially in each compartment (inside or outside) mix throughout the course of the simulation.

To do so, a first essential step is reconstructing the membrane surface from the model particle positions and orientations, for which we use custom Python code[59] based on the ball pivot algorithm[60]. Our analysis provides an estimate for the vesicle surface area and local boundaries and orientations together with a collection of pores and their perimeter, which allows us to evaluate pore closure.

Once we have defined the vesicle surface, we evaluate condensation by running a clustering algorithm on the solute and protein particles (part of the Ovito library in Python[61]). This allows us to distinguish between condensate particles and solution particles and to calculate droplet properties (such as mass or radius of gyration) as well as the total droplet growth rate over time. Once we have identified the particles that remain in solution, we can then determine which compartment they belong to (inside the vesicle or outside) and by comparing with the initial conditions we can determine solution exchange fluxes.

To classify the particles as inner lumen or outer lumen we determine each particle's closest six surface triangles (using a neighbour list algorithm from the Freud library for Python[62]) and compute the unit vector along the line connecting the particle and each of its neighbour triangles centres. We then compute the dot products $\rho$ of these unit vectors with the local normal of the surface (previously identified using the ball pivot algorithm). Finally, if the average dot product $\langle \rho \rangle > 0.5$ we classify the particle as being inside the vesicle. Conversely, if $\langle \rho \rangle < -0.5$ we classify the particle as being outside the vesicle. In this analysis, particles that are close to the pore present dot products close to zero

($|\langle\rho\rangle| < 0.5$), and in this case we compare their position to the pore's centre of mass to determine their compartment.

Once we have a list of inner particles, outer particles and droplet particles, we can define fluxes by comparing these values across the simulation and characterize the mixing of solutions and the condensate growth in different conditions.

### Imaging

**Stress granules and GAL-3 staining.** After the indicated treatment, cells were washed once with PBS and fixed with 4% methanol-free paraformaldehyde (PFA) in PBS for 15 min. After 3 washes with PBS, cells were permeabilized using a 0.5% Triton X-100 (Sigma)/PBS solution for 10 min. Cells were then immunostained using the corresponding antibodies. Antibodies were: anti-G3BP1 (13057-2-AP) or alternatively, anti-G3BP1 (66486-1-Ig, sc-365338 AF546), anti-G3BP2 (16276-1-AP) anti-TIA1 (12133-2-AP), anti-PABPC1 (10970-1-AP), anti-ALIX (12422-1-AP), anti-CHMP2a (10477-1-AP), anti-CHMP4b (13683-1-AP), anti-annexin A1 (66344-1-Ig), anti-annexin A2 (66035-1-Ig), anti-EIF3B (10319-1-AP), and anti-EIF4G1 (15704-1-AP) from Proteintech. Anti-GAL-3 (125410) and anti-LAMP-1 (121610) from Biolegend. Anti-p62 (GTX111393) from GeneTex, anti-phospho TBK1 (5483 T) from Cell Signalling Technologies. Anti-PI4K2a (B-5, sc390026) and anti-ORP9 (A-7, sc398961) from Santa Cruz. Antibodies were used at 1:200 dilution. Images were acquired on Ibidi glass bottom slides (81817) using a Leica SP8 confocal microscope (for Mtb fixed imaging and silica crystals experiments) or using a VT-iSIM superresolution imaging system (Visitech International). Nuclear staining was performed using 300 nM DAPI (Life Technologies, D3571) in PBS for 5 min. The area of GAL-3 and stress puncta per cell was evaluated using the image analysis software FIJI/ImageJ as described below.

**Fluorescence recovery after photobleaching.** Fluorescence recovery after photobleaching (FRAP) experiments were performed using a Zeiss Invert880 microscope system (Zeiss) and the bleaching module of Zeiss ZEN software. Photobleaching of G3BP1−mEGFP was achieved using a 488-nm laser at a scan speed = 1 (pixel dwell 131.07 ms). Images were captured at 20 or 30-s intervals for at least 20 min. The photobleaching started after three time points that were used as the basal intensity reference.

**Confocal microscopy of GUVs.** Confocal Leica SP8 microscope equipped with a 63×, 1.2 NA water-immersion objective and a 40×1.3 NA oil immersion objective (Leica) was used for imaging. FITC and ATTO 647N-DOPE were excited using the 488 nm and 633 nm laser lines, respectively and signal was collected between 500–600 nm for FITC and 650–720 nm for ATTO. Time-lapse sequences (*xyt*) of individual GUVs were acquired at 650 ms per frame using the bidirectional acquisition mode.

**High-content live-cell imaging.** Thirty thousand iPSDM were seeded into a ViewPlate glass bottom 96 well plate (PerkinElmer) and treated with LLOMe or infected with Mtb as described above. The plate was sealed with parafilm and placed in a pre-heated (37 °C) Opera Phenix microscope with a 40× or 60× water-immersion lens (PerkinElmer) with 5% $CO_2$. Capture settings were: LysoTracker Red was excited with the 561 nm laser at 10% power with 100 ms exposure. MitoTracker Deep Red FM (M22426, Thermo Fischer), iABP probe and Mtb E2-Crimson were excited with the 640 nm laser at 10% power with 100 ms exposure. DAPI was excited with the 405 nm laser at 20% power with 100 ms exposure. At least 20 fields per well were imaged in all the experiments. Images were acquired at 1,020 × 1,020 pixels using Harmony 4.9 high-content imaging and analysis software (PerkinElmer).

**Superresolution live-cell imaging.** iPSDMs transfected with GAL-3−RFP, G3BP1−mEGFP or LAMP1−RFP were treated with LLOMe (1 mM) and imaged on a VT-iSIM superresolution imaging system (Visitech International), using an Olympus IX83 microscope, 150×/1.45 Apochromat objective (UAPON150XOTIRF), ASI motorized stage with piezo Z, and 2× Prime BSI Express scientific CMOS cameras (Teledyne Photometrics). Cells were always in the stage incubator at 37 °C and 5% $CO_2$. Simultaneous GFP and mCherry imaging was done using 488 nm and 560 nm laser excitation and ET525/50 m and ET600/50 m emission filters (Chroma), respectively. *Z*-stacks (100 nm *z*-step) were acquired at the intervals indicated in the figure legends. The microscope was controlled with CellSens software (Olympus). Image processing and deconvolution was done using Huygens Essential software (Scientific Volume Imaging). For 3D imaging, spatial deconvolution and 3D surface-rendering reconstruction, *z*-stack slices were defined each 200 nm (if confocal microscope was used) or 100 nm (if VT-iSIM microscope was used) and images were processed using Huygens Essential Software (Scientific Volume Imaging).

**LysoTracker recovery assay.** Forty thousand cells (background indicated in the figure legends) were seeded into a ViewPlate glass bottom 96 well plate (PerkinElmer). Cells were loaded with the nuclear dye Nuclear Green LCS1 (Abcam) at a dilution of 1:5,000 and 25 nM LysoTracker DND-99 (Thermo Fisher Scientific) for 20 min. The cells were washed twice in X-VIVO15, and the medium was replaced with X-VIVO15 containing 25 nM LysoTracker. The cells were imaged every 1 min at 37 °C, 5% $CO_2$ using an Opera Phenix microscope (PerkinElmer). First, a baseline was established by imaging 3 time points, followed by the addition of LLOMe to a final concentration of 1 mM. After 2 time points, the cells were washed 3 times with X-VIVO15 and the medium was replaced with X-VIVO15 containing 25 nM LysoTracker, and lysosomal recovery was followed for 20 min. Analysis was done as indicated below (see 'Imaging analysis').

**10K-dextran release assay.** Wild-type and G3BP-DKO U2OS cells were seeded into a PerkinElmer Cell Carrier Ultra 96 well plate (PerkinElmer) and incubated with 25 μg ml$^{-1}$ 10K-dextran conjugated to Alexa Fluor 647 (D22914; Thermo Fisher Scientific) in DMEM for 12–16 h, and then in fresh DMEM for 2–4 h (chase) to allow dextran to reach the lysosomes[3]. After that, cells were treated with LLOMe (2 mM, 2 min), cells were then washed within a min and imaged for 40 min after in fresh medium. Images were acquired at 1 min intervals.

**Immunofluorescence of lung sections.** Tissue sections (see 'Tissue sectioning') on SuperFrost Plus Adhesion slides (Fisher Scientific, 11950657) were unfrozen at room temperature and and permeabilized with a 0.5% solution of Tween-20/PBS for 20 min. After that, samples were washed twice with PBS and stained with the corresponding primary antibodies (dilution 1:100) for 2 h. The coverslips were then washed twice with PBS and incubated with the corresponding Alexa Fluor (Thermo Fischer) secondary antibodies (dilution 1:700) for 1 h. After two washes with PBS, the samples were mounted using Dako fluorescence mounting medium (S3023).

**Poly(A) RNA in situ protocol.** iPSDM cultured in μ-Slide 18-Well Glass Bottom plates (Ibidi) were fixed in 4% PFA for 10 min, PFA aspirated, and 100% cold methanol added to each well for 10 min. Methanol was replaced with 70% ethanol and incubated for 10 min. The ethanol was aspirated and 1 M Tris, pH 8.0 added to each well for 5 min. After Tris removal, hybridization buffer was added containing the dilution of 5′-labelled Cy3-Oligo-dT(20) stock (Integrated DNA Technologies) for a final concentration of 1 ng μl$^{-1}$. Hybridization was carried out at 37 °C for 2 h. After hybridization, samples were washed once with 4× SSC and then once with 2× SSC (all DEPC-treated). Incubation with primary antibodies was in 2× SSC + 0.1% Triton X-100 for 2 h, washed three times with 2× SSC, and then incubated with secondary antibodies for 1 h at room temperature. Hybridization buffer composition: 1 mg ml$^{-1}$ Yeast tRNA

(AM7119 Thermo Fischer), 0.005% BSA, 10% dextran sulfate (D8906-5G, Merck), 25% Formamide (17899, Thermo Fischer) 20× SSC + DEPC water so that final buffer volume is in 2× SSC.

## Electron microscopy

**Fixation.** Samples were fixed by adding a mixture of 8% PFA in 200 mM HEPES buffer to culture medium (v/v) and incubated at room temperature for 15 min then replaced with 4% PFA in 100 mM HEPES for 30 min before imaging by confocal microscope. After imaging by confocal, samples were transferred to 1% glutaraldehyde in 100 mM HEPES buffer.

**Resin embedding.** Fluorescently imaged samples were processed for correlative light and electron microscopy in a Biowave Pro (Pelco) with use of microwave energy and vacuum. Cells were twice washed in HEPES (Sigma-Aldrich H0887) at 250 W for 40 s, post-fixed using a mixture of 2% osmium tetroxide (Taab O011) 1.5% potassium ferricyanide (Taab, P018) (v/v) at equal ratio for 14 min at 100 W power (with/without vacuum 20 inch Hg at 2-min intervals). Samples were washed with distilled water twice on the bench and twice again in the Biowave 250 W for 40 s. Samples were stained with 1% aqueous uranyl acetate (Agar scientific AGR1260A) in distilled water (w/v) for 14 min at 100 W power (with/without vacuum 20 inch Hg at 2-min intervals) then washed using the same settings as before. Samples were dehydrated using a step-wise acetone series of 50, 75, 90 and 100%, then washed 4 times in absolute acetone at 250 W for 40 s per step. Samples were infiltrated with a dilution series of 25, 50, 75, 100% Durcupan ACM (Sigma-Aldrich 44610) (v/v) resin to propylene oxide. Each step was for 3 min at 250 W power (with or without vacuum 20 inch Hg at 30 s intervals). Samples were then cured for a minimum of 48 h at 60 °C.

**Sample trimming and image acquisition.** Referring to grid coordinates, the sample block was trimmed, coarsely by a razor blade then finely trimmed using a 35° ultrasonic, oscillating diamond knife (DiATOME, Switzerland) set at a cutting speed of 0.6 mm s$^{-1}$, a frequency set by automatic mode and a voltage of 6.0 V, on a ultramicrotome EM UC7 (Leica Microsystems, Germany) to remove all excess resin surrounding the region of interest (ROI). Ribbons were cut to a thickness of 65 nm and Images were acquired using a JEOL JEM-1400 series 120 kV transmission electron microscope.

**Correlative light and electron microscopy image alignment.** Fluorescent images were converted to tiff file format and liner adjustments made to brightness and contrast using FIJI (version 2.9.0/1.53t). Fluorescent images were aligned to serialEM micrographs (TrakEM2) using BigWarp_fiji_7.0.7 plugin. No less than 10 independent fiducials were chosen per alignment for 3D image registration. When the fiducial registration error was greater than the predicted registration error, a non-rigid transformation (a nonlinear transformation based on spline interpolation, after an initial rigid transformation) was applied.

## Imaging analysis

**Live-cell imaging analysis of G3BP1$^+$ and GAL-3$^+$ events.** An event was considered positive when the fluorescence signal was at least three times greater than the mean background (determined previously to the addition of LLOMe). Intensities were quantified selecting the corresponding ROI in FIJI/ImageJ. G3BP1$^+$ and GAL-3$^+$ events were manually tracked over time and the time registered when the first positive event was detected. At least 20 events per experiment were annotated. The percentage of 'capping events' was determined considering the amount of double-positive G3BP1$^+$/GAL-3$^+$ events that presented a polarized G3BP1 fluorescence signal distribution in comparison with the distribution observed for GAL-3$^+$ vesicles. To this end, GAL-3$^+$ areas were segmented and the mean intensity of G3BP1 in those areas determined. A G3BP1$^+$ event was considered polarized (in 'cap pattern') when the G3BP1 mean intensity, of a ROI

in proximity of a GAL-3$^+$ event, was at least two times greater than the G3BP1 mean intensity corresponding to the segmented (overlapping) GAL-3$^+$ area. For fluorescence intensity or puncta area over time analysis, the corresponding values after segmentation were plotted using the geom_smooth function in R Studio (method = "loess") (see source data for Fig. 1).

**Spatial point pattern analysis.** To determine whether the distances between points in group A (G3BP1$^+$) and points in group B (GAL-3$^+$) were significantly different from the distances between points in group A and randomly generated points, we performed a spatial point pattern analysis using the spatstat package in R (version 3.0–6). First, a point pattern object was created from the $x$ and $y$ coordinates of the data using the ppp function. The lambda value, which represents the intensity of the point pattern for group A, was then calculated as the number of points in group A divided by the area of the point pattern. Next, a random point pattern with the same window as the original point pattern was generated using the runifpoint function. The distances between points in group A and points in group B were then calculated using the nncross function, and the distances between points in group A and the randomly generated points were calculated in a similar way. The number of simulated distances that were less than or equal to the observed distances was counted and used to calculate a $P$ value using a Monte Carlo simulation. A total of 1,000 simulations were performed to estimate the $P$ value. Finally, a boxplot was created to visually compare the distribution of observed distances between points in group A and points in group B to the distribution of distances between points in group A and randomly generated points.

**Stress granules and GAL-3 puncta analysis.** Analysis was done in Fiji/ImageJ using the sequence Image>Adjust>Threshold and then puncta or area in the segmented image was determined using the menu command Analyze>Analyze particles. Size was restricted to particles greater than 0.1 um and the circularity restricted to values between 0.4 and 1.

**Co-localization analysis.** VT-iSIM images were analysed using the EzColocalization plugin[63] on FIJI/ImageJ and Spearman's rank correlation coefficient (SRCC) values used for quantification. Analysis was done on a single $z$-stack section of 150 nm.

**High-content imaging and evaluation of lysosomal activity and content.** iPSDM were incubated with 1 µM solution of the iABP Smart Cathepsin Imaging Probe (40200-100, Vergent Bioscience) for 3 h at 37 °C and 5% CO$_2$. Cells were single-cell segmented based on nuclear staining and lysosomes segmented using the Find Spots and Morphology Properties modules of Harmony 4.9 software. After that lysosomal intensity values (650–760 em) and cellular morphology parameters were quantified.

**LysoTracker recovery analysis.** iPSDM were segmented using the NucGreen nuclear signal and Find Nuclei module or based on cellular segmentation after applying the Inverse Grey palette and Find Cell module of Harmony 4.9 software. LysoTracker puncta were identified using the Spots building block (local maximum). The spot intensity per cell and timepoint was normalized to the average of relative spot intensity before LLOMe addition. For each experiment, at least 300 cells were analysed per condition.

**Mtb replication analysis.** Images of Mtb-infected iPSDM were acquired on an Opera Phenix microscope using a 40× objective with at least 20 fields of view per well (with three wells per condition) and analyzed in Harmony 4.9. Cells were segmented based on DAPI, excluding any cells touching the edge of the imaged area. Bacteria were detected using the Find Spots building block of Harmony. The total bacterial area in each cell was then determined. Data was exported and analysed in R Studio to calculate the mean Mtb area per cell for each condition at

each timepoint, with all three wells pooled. At least three independent experiments were done per condition and timepoint.

## Mouse aerosol infection with Mtb H37Rv

Six- to eight-week-old, female C3HeB/FeJ mice were infected with either wild-type Mtb Wasabi Hyg[R] or Mtb ΔRD1 E2-Crimson Kan[R]. Sample size was determined in accordance with ARRIVE guidelines and previous studies[64]. Five animals per group were used per time of infection. Females were used for safety and space allocation restrictions as infected mice were contained in BSL3. All mice were maintained in BSL3 cages, at 22 ± 2 °C and a relative humidity of 55 ± 10%. No randomization or blinding was applied for this study. For low dose aerosol infection experiments all bacteria were used at mid-exponential phase and a Glascol aerosol generator was calibrated to deliver approximately 100 colony-forming units (CFU) of wild-type Mtb Wasabi Hyg[R]/lung and 500 CFU of Mtb ΔRD1 E2-Crimson Kan[R]/lung. Fifty-six days after infection, lungs were perfused in 4% PFA for fluorescence microscopy analysis. Lungs were equilibrated in 0.1 M HEPEs buffer (pH 7.4) with 0.2 M sucrose for 1 h before being transferred to silicon moulds containing OCT medium (Agar scientific, AGR1180). Moulds containing OCT and lungs were then transferred to dry ice and frozen in preparation for sectioning. Sections were cut using a Leica CM30505S Cryostat (CT-18 °C, OT-20 °C), to a size of 8 µm. Sections were collected on SuperFrost Plus Adhesion slides (Fisher Scientific, 11950657) and stored at −80 °C before further processing.

## Statistical analysis

Statistical analysis was performed using GraphPad Prism 10 software or R Studio 2023.03.0 (R version 4.2.2). High-content imaging analysis and mean values were obtained using R 4.2.2 or Harmony 4.9 software. The number of biological replicates, the statistical analysis performed, and post hoc tests used are mentioned in the figure legends. The statistical significance of data is denoted on graphs by informing the $P$ value or asterisks, where $*P < 0.05$, $**P < 0.01$, $***P < 0.001$; or NS, not significant. Graphs were plotted in GraphPad Prism software or using R Studio 2023.03.0 (R version 4.2.2). RNA-sequencing data were obtained from a previous study[42] and plotted using Morpheus (https://software.broadinstitute.org/morpheus/). Schematics were created with BioRender.com.

## Ethics

This study involved the use of KOLF2 human iPS cells, Public Health England Culture Collections, catalogue number 77650100, and the use of WTC human mEGFP-tagged G3BP1 iPS cells (Coriell Institute, AICS-0082 cl.1). The use of human cells is covered and approved by the Ethical Committee and regulated by the Francis Crick Institute Biological Safety Code of Practice in the project registered at the Crick (Project HTA17) framed under Human Tissue Authority Licence number 12650.

## Reporting summary

Further information on research design is available in the Nature Portfolio Reporting Summary linked to this article.

## Data availability

The data needed to evaluate the conclusions of the study are present in the manuscript or in the supplementary materials. Source data for gels and blots are provided as Supplementary information (Supplementary Fig. 1). Source data are provided with this paper.

## Code availability

Custom analysis codes were used to extract pore lifetime and solution exchange measurements from simulations. All analysis code used is available on a public repository at https://doi.org/10.15479/ AT:ISTA:14472. This includes analysis code as well as relevant examples of simulation scripts used in this work and instructions on how to run them. Details on the simulations setup and analysis implementation can be found in the Methods section.

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

**Acknowledgements** We thank the Human Embryonic Stem Cell Unit, Advanced Light Microscopy and High-throughput Screening facilities at the Crick for their support in various aspects of the work. We thank the laboratory of P. Anderson for providing the G3BP-DKO U2OS cells. The authors thank N. Chen for providing the purified glycinin protein; Z. Zhao for providing the microfluidic chip wafers; and M. Amaral and F. Frey for helpful discussions and valuable input regarding analysis methods. This work was supported by the Francis Crick Institute (to M.G.G.), which receives its core funding from Cancer Research UK (FC001092), the UK Medical Research Council (FC001092) and the Wellcome Trust (FC001092). This project has received funding from the European Research Council (ERC) under the European Union's Horizon 2020 research and innovation programme (grant agreement no. 772022 to M.G.G.). C.B. has received funding from the European Respiratory Society and the European Union's H2020 research and innovation programme under the Marie Sklodowska-Curie grant agreement no. 713406. A.M. acknowledges support from Alexander von Humboldt Foundation and C.V.-C. acknowledges funding by the Royal Society and the European Research Council under the European Union's Horizon 2020 Research and Innovation Programme (grant no. 802960 to A.S.). All simulations were carried out on the high-performance computing cluster at the Institute of Science and Technology Austria. For the purpose of Open Access, the author has applied a CC BY public copyright licence to any Author Accepted Manuscript version arising from this submission.

**Author contributions** C.B. performed and analysed all the light microscopy imaging studies done in cellulo, immunoblots and replication experiments with Mtb. A.M. performed and analysed all the in vitro experiments using GUVs with support from T.M.F. and R.D. C.V.-C. performed and analysed the simulation studies with support from A.Š. E.P. contributed with the nucleofection experiments. N.A. contributed with the maintenance of induced pluripotent stem cells and generation of iPSDM. B.A. contributed with the imaging studies. C.B., A.M. and C.V.-C. generated the figures. A.R. performed the mouse aerosol infection with Mtb. A.F. performed the electron microscopy studies. C.B., M.G.G., A.Š. and R.D contributed with the project supervision. C.B. and M.G.G. designed the study and wrote the original draft. C.B., A.M., C.V.-C., B.A., E.P., N.A., A.F., A.R., T.M.F, A.Š., R.D. and M.G.G. contributed with manuscript review and editing.

**Funding** Open Access funding provided by The Francis Crick Institute.

**Competing interests** The authors declare no competing interests.

**Additional information**
**Correspondence and requests for materials** should be addressed to Claudio Bussi or Maximiliano G. Gutierrez.

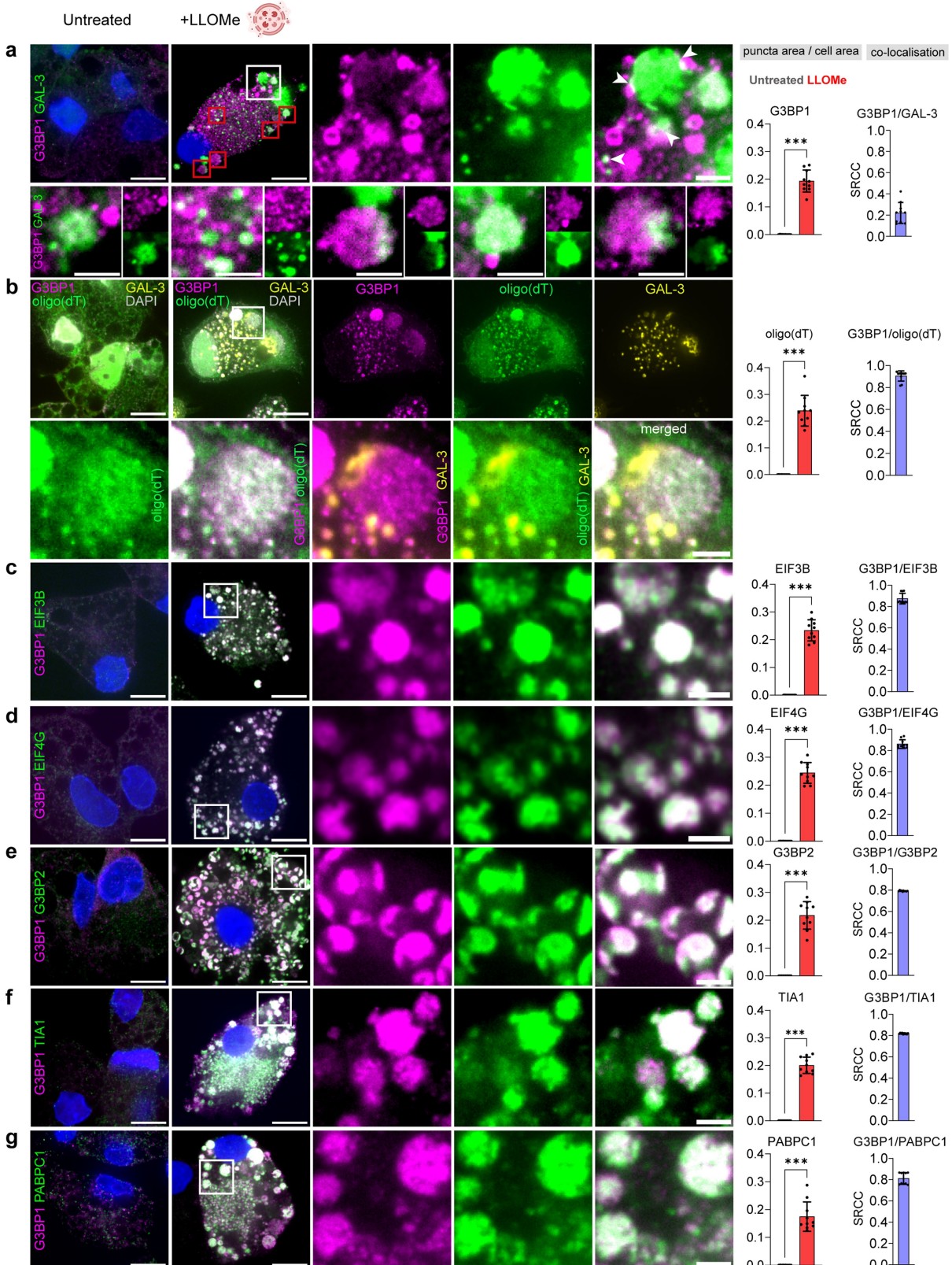

**Extended Data Fig. 1 | Lysosomal damage induces stress granule formation in human macrophages. a**-**g**, iPSDM were left untreated or treated with LLOMe (1 mM, 30 min) and stained for G3BP1 and GAL-3 (a), bottom panel shows different zoom-in areas (red squares); poly(A) (b); EIF3B (c); EIF4G (d); G3BP2 (e); TIA1 (f) and PABPC1 (g). The bar plots on the right show the corresponding quantification of the indicated marker area normalised to cell area, and the corresponding Spearman's rank correlation coefficient (SRCC). Data represent the mean ± SEM of at least 10 cells from one out of three independent experiments ($n$ = 30 cells examined over 3 independent experiments). Images show only one z-stack section of 150 nm. Scale bar: 10 μm, zoom-in: 2 μm. Two-tailed t-test, ***$P$ < 0.001.

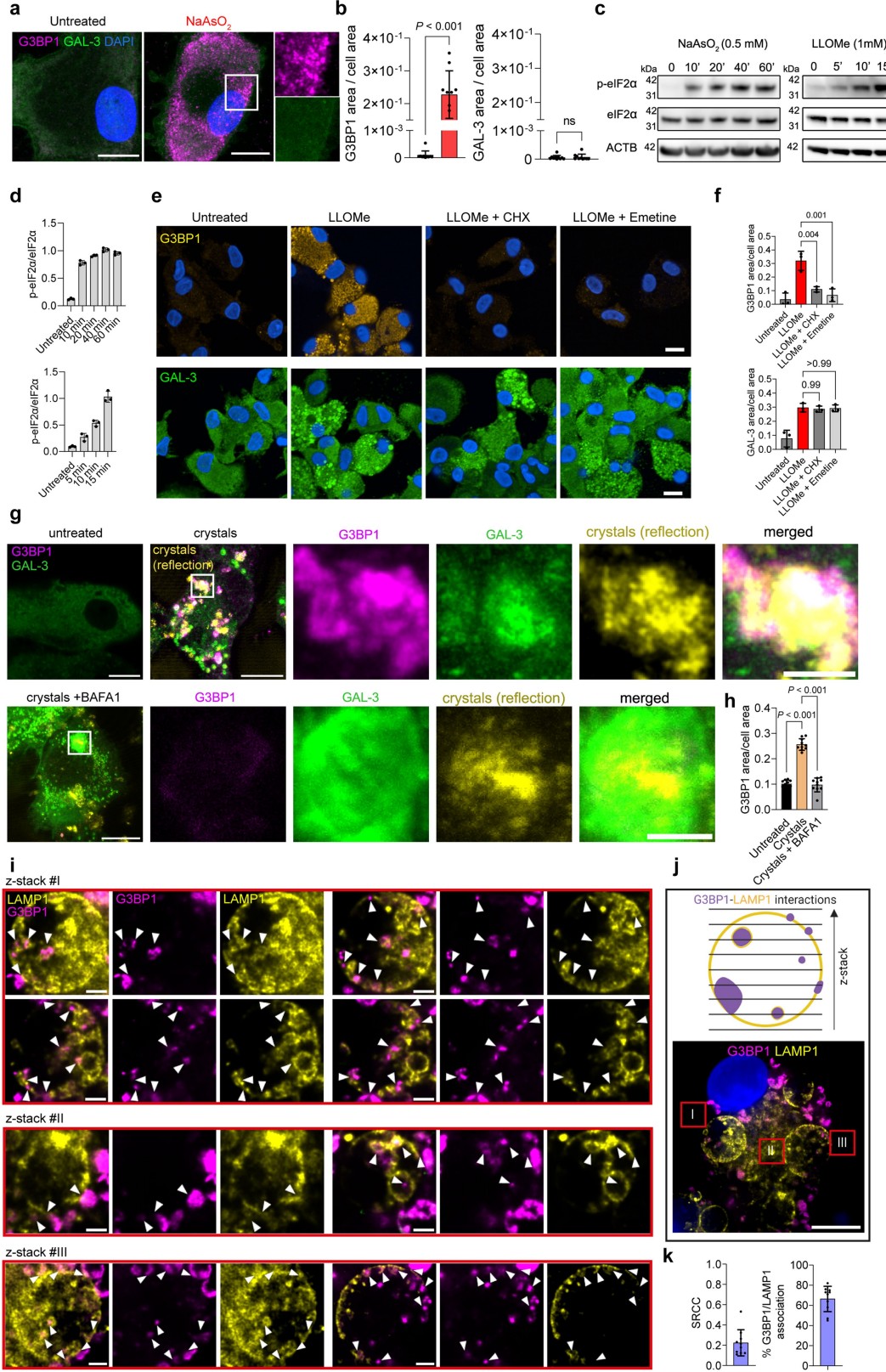

**Extended Data Fig. 2 |** See next page for caption.

**Extended Data Fig. 2 | Supplementary data related to Fig. 1. a**, Representative images of iPSDM left untreated or treated with $NaAsO_2$ (0.5 mM, 1 h) and stained for G3BP1 (magenta) and GAL-3 (green). **b**, Bar plots show quantification of G3BP1, and GAL-3 puncta area normalised to cell area of iPSDM treated as in (a). Data represent the mean ± SEM of at least 10 cells from one out of three independent experiments ($n$ = 30 cells examined over three independent experiments, two-tailed t-test). **c**, immunoblot for phospho- (p) and total levels of eIF2a in iPSDM left untreated or treated with $NaAsO_2$ (0.5 mM) or LLOMe (1 mM) at the indicated time points. ACTB was used as a loading control. **d**, shows eIF2a protein levels relative to ACTB, Bar plot represents mean values ± SEM ($n$ = three independent experiments). **e**, Representative images of iPSDM left untreated or treated with LLOMe (1 mM, 20 min) and incubated in the presence or absence of 10 μg/ml cycloheximide (CHX) or 10 μg/ml emetine; and stained for G3BP1 (top panel) and GAL-3 (bottom panel). **f**, Bar plots show high-content imaging quantification of G3BP1, and GAL-3 puncta area normalised to cell area of iPSDM treated as in (e). Data represent the mean ± SEM ($n$ = three independent experiments). $P$-value was calculated using one-way ANOVA, Tukey's multiple comparisons test. **g**, iPSDM were incubated in the presence or absence of BAFA1 (100 nM, 1 h) and left untreated or treated with silica crystals (200 μg/mL, 3 h), and stained for GB3P1 and GAL-3. Silica crystals were imaged using confocal reflection microscopy. **h**, Bar plot shows the quantification of G3BP1 puncta area normalised to cell area. Data represent the mean ± SEM of at least 10 cells from one out of three independent experiments ($n$ = 30 cells examined over 3 independent experiments) $P$-value was calculated using one-way ANOVA, Tukey's multiple comparisons test. **i,j**, Fixed image analysis of G3BP1-LAMP1 (lysosome) interactions (i) in iPSDM stimulated with LLOMe (1 mM, 30 min). #1–3 show three different z-stack sequences corresponding to the cell regions indicated in (j) (red squares). Arrowheads illustrate different membrane and intralumenal G3BP1-LAMP1 interactions. **k**, Bar plots show G3BP1/LAMP1 SRCC and the percentage of events where G3BP1 associates with LAMP-1[+] structures. Data represent the mean ± SEM of at least 10 cells from one out of 3 independent experiments ($n$ = 30 cells examined over 3 independent experiments). Scale bar: 10 μm, zoom-in: 2 μm. For gel source data, see Supplementary Fig. 1.

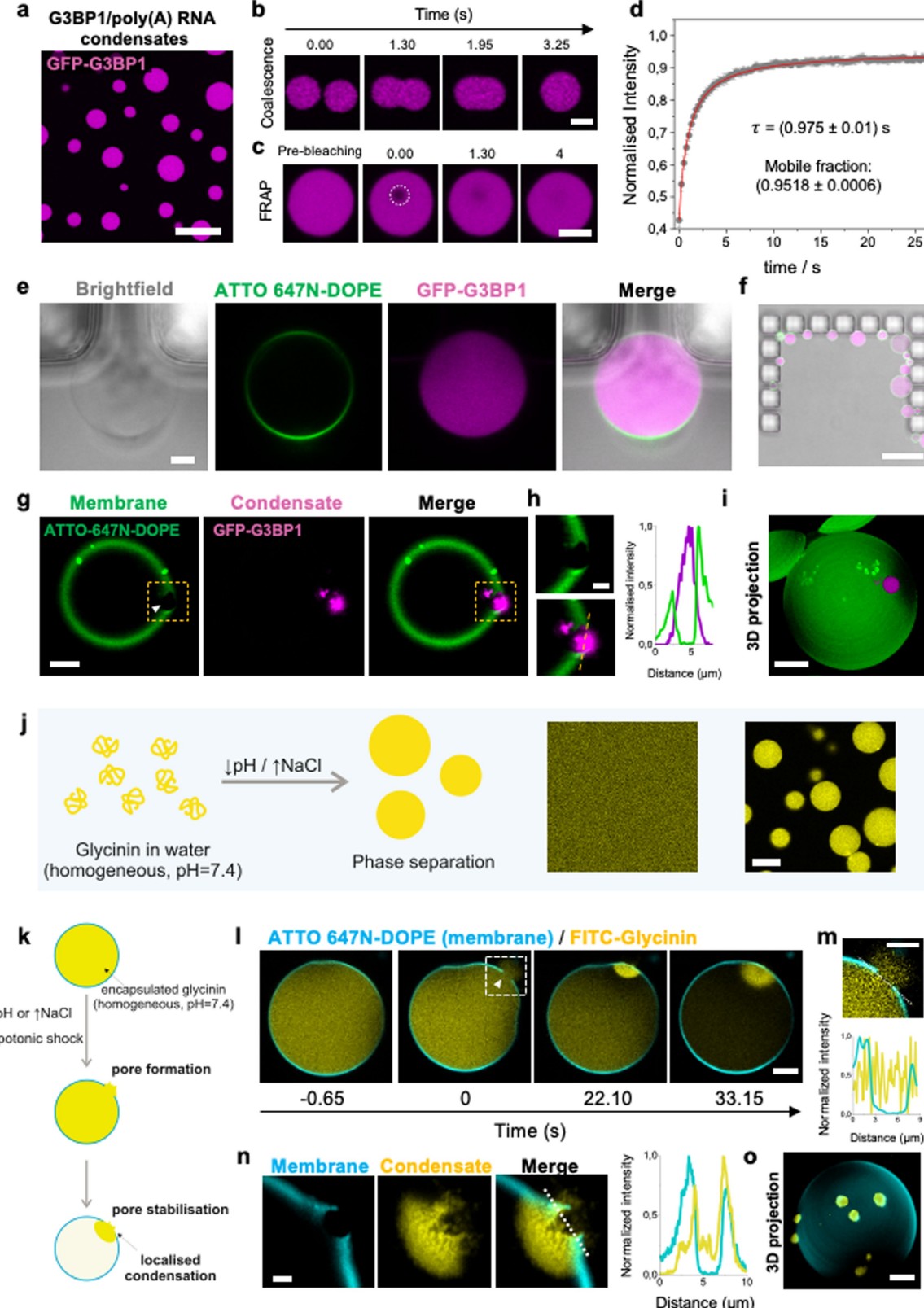

**a** G3BP1/poly(A) RNA condensates

GFP-G3BP1

**b** Coalescence

Time (s)

0.00  1.30  1.95  3.25

**c** FRAP

Pre-bleaching  0.00  1.30  4

**d**

$\tau = (0.975 \pm 0.01)$ s

Mobile fraction: $(0.9518 \pm 0.0006)$

Normalised Intensity

time / s

**e** Brightfield   ATTO 647N-DOPE   GFP-G3BP1   Merge

**f**

**g** Membrane   Condensate   Merge

ATTO-647N-DOPE   GFP-G3BP1

**h**

Normalised intensity

Distance (µm)

**i** 3D projection

**j** Glycinin in water (homogeneous, pH=7.4)   ↓pH / ↑NaCl   Phase separation

**k** encapsulated glycinin (homogeneous, pH=7.4)

↓pH or ↑NaCl hypotonic shock

pore formation

pore stabilisation

localised condensation

**l** ATTO 647N-DOPE (membrane) / FITC-Glycinin

−0.65  0  22.10  33.15

Time (s)

**m**

Normalized intensity

Distance (µm)

**n** Membrane   Condensate   Merge

Normalized intensity

Distance (µm)

**o** 3D projection

**Extended Data Fig. 3** | See next page for caption.

**Extended Data Fig. 3 | Supplementary Data related to Fig. 2. a**, Stress granules reconstitution *in-vitro*[3]. Confocal fluorescence image of GFP-G3BP1/poly(A)-RNA condensates at pH = 5. Scale bar: 20 μm. **b**, Coalescence of GFP-G3BP1/poly(A)-RNA condensates occur within seconds. Scale bar: 5 μm. **c-d**, Fluorescence recovery after photobleaching (FRAP) shows a fast fluorescent recovery. **c**, Confocal images showing the condensate bleaching and the fluorescence recovery in a few seconds. Scale bar: 5 μm. **d**, Analysis of the FRAP data for G3BP1/RNA condensates, mean values are plotted as grey dots ± SD (*n* = 10 events analysed over 3 independent experiments) and the fit curve as a red line. **e**, Bright field and confocal sections of a GUV (green, DOPC:DOPS 9:1, with 0.1 mol% of the dye ATTO-647N-DOPE) encapsulating GFP-G3BP1 (magenta, 10 μM in 50 mM HEPES pH = 7.5, 100 mM NaCl and 4 mM $MgCl_2$) pressed against the posts of a microfluidic system. Scale bar: 5 μm. **f**, merged bright-field and confocal fluorescence image of vesicles in one of the traps of the microfluidic device used for GUV trapping and complete exchange of the external medium. Each microfluidic chip has 8 channels containing, 17 traps per channel. Scale bar: 100 μm. **g**, Confocal fluorescence image of a damaged vesicle (green) stabilized by G3BP1/RNA condensate (magenta) formed at the pore after inducing membrane damage by exchanging the external solution for a hypotonic solution at lower pH containing poly(A)-RNA (20 mM HEPES, pH = 5, 100 mM NaCl, 4 mM $MgCl_2$, 200 ng/μL poly(A)-RNA). The white arrowhead points at the membrane pore. Scale bar: 5 μm. **h**, zoomed images of the rectangular dashed regions in g, showing the membrane discontinuity and the G3BP1/RNA condensate plugging the pore. The intensity profile shows the condensate wetting at the pore rim. Scale bar: 2 μm. See Supplementary video 10. **i**, 3D reconstruction of the vesicle shown in g. Scale bar: 10 μm. **j**, Sketch showing glycinin as highly soluble and forming homogeneous solutions in water at neutral pH but undergoing phase-separation when the pH is decreased (below pH ≈ 6.5) or the NaCl concentration is increased (above [NaCl] ≈ 40 mM). The confocal fluorescence images on the right show a homogeneous solution of FITC-glycinin in water (pH = 7.4) and the phase separated protein-rich droplets in the presence of 150 mM NaCl at the same pH. Scale bar: 10 μm. **k**, Sketch showing the experimental design for glycinin experiments: phase separation at the pore site can be triggered by lowering the pH or increasing the NaCl concentration. **l-o**, Confocal fluorescence images showing the pore formation and sealing by local phase separation of glycinin at the damaged zone. Vesicles are made of DOPC containing 0.1 mol% of the dye ATTO 647N-DOPE, encapsulating a homogeneous solution of glycinin 10 mg/mL containing a 4 mol% of FITC-glycinin (15 mM NaCl, 10 mM sucrose, pH = 7.4). The hypotonic buffer for triggering phase separation consisted in deionized water with pH = 4.8 adjusted by addition of HCl 2 M. **l**, Time-lapse sequence showing the GUV poration and subsequent condensate formation at the damaged zone (see Supplementary Video 11). Time stamp zero indicates the moment of pore formation. The glycinin condensate forms quickly at the pore site. The vesicle is slightly deformed as it is pressed by the flow against the microfluidic posts. Scale bar is 10 μm. **m**, zoomed image and intensity profile of the squared dashed region in l, showing the membrane discontinuity at the pore region and the encapsulated content leaking to the external solution. Scale bar: 5 μm. **n**, Confocal section of the ruptured membrane and the condensate patch of the vesicle system shown in l. The intensity profile at the dashed white line shows the membrane wetting by the condensate. Scale bar: 5 μm. **o**, Pore plugging can be reached via a different phase separation trigger: in this case the encapsulated glycinin solution contained 300 mM sucrose at pH 7.4 and phase separation was promoted via a hypotonic shock and increasing the salinity of the external buffer while keeping the pH constant (100 mM NaCl, pH = 7.4). The image corresponds to a 3D projection of a vesicle showing several pores plugged by glycinin condensates. Scale bar: 10 μm.

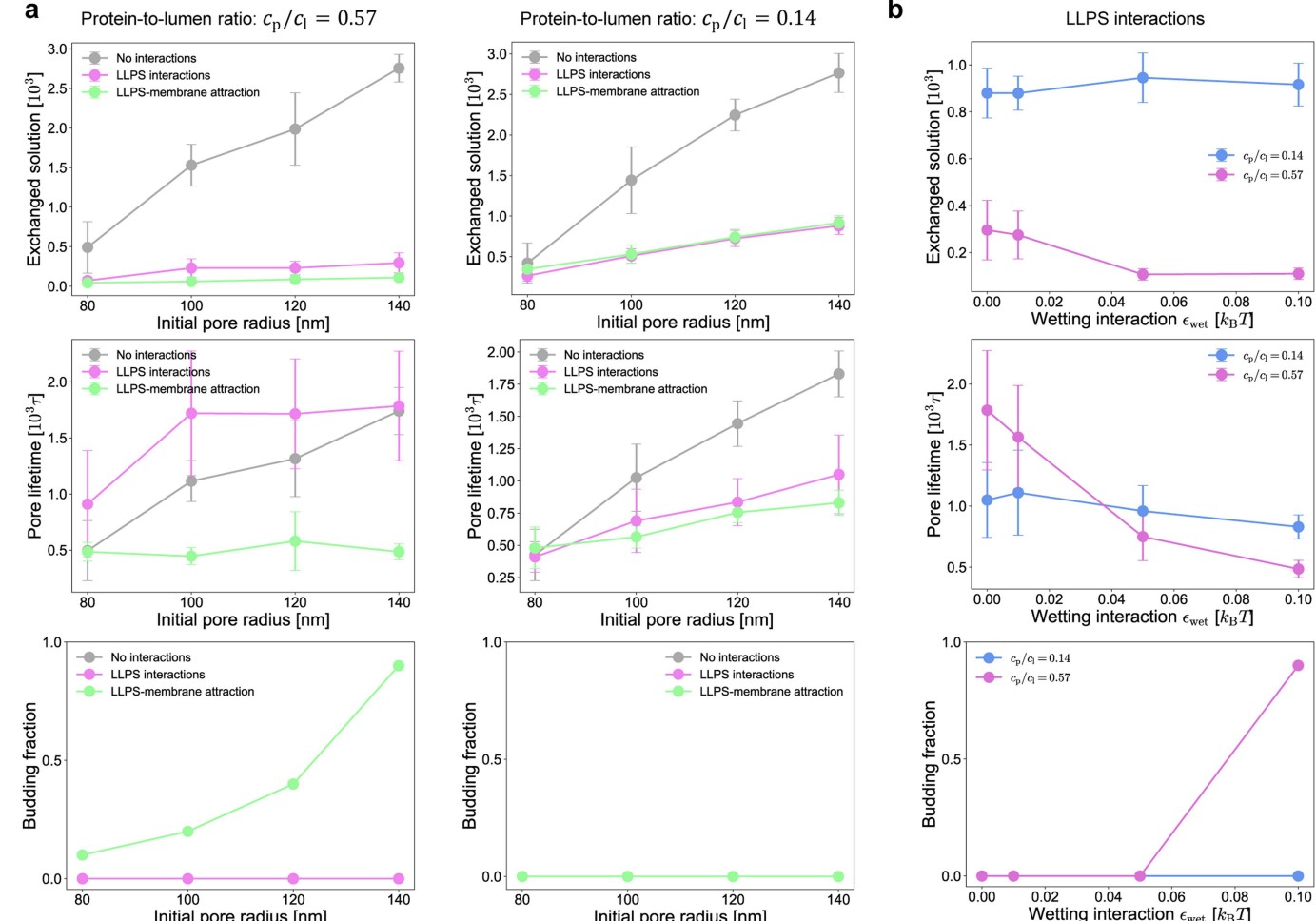

**Extended Data Fig. 4 | Plugging and engulfment depends on wetting and protein concentration. a**, Amount of exchanged solution, pore lifetime and budding vesicle fraction for the three main cases of interest (see Fig. 2i) at high (left) and low (right) protein-to-lumen concentration ratio ($c_p/c_l = 0.57$ and $c_p/c_l = 0.14$ respectively) as a function of initial pore radius. **b**, Amount of exchanged solution, pore lifetime and budding vesicle fraction as a function of the wetting interaction strength for LLPS protein-solute interactions for high and low protein-to-lumen ratio (initial pore radius $r_p = 140$ nm). At low protein concentrations the plugging effect is inhibited, and wetting has little effect. At high protein concentrations wetting promotes pore closure and can drive engulfment and budding of the droplet. Dots are the average over 10 replicas and error bars represent the standard deviation; $\tau$ is the MD unit of time.

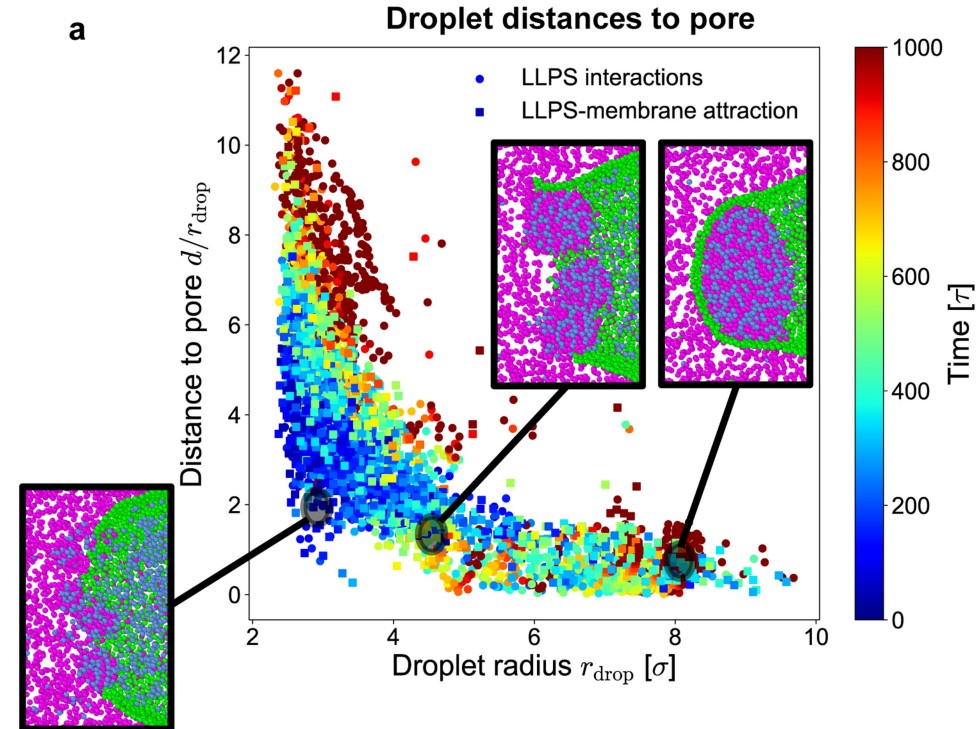

**a**

**Droplet distances to pore**

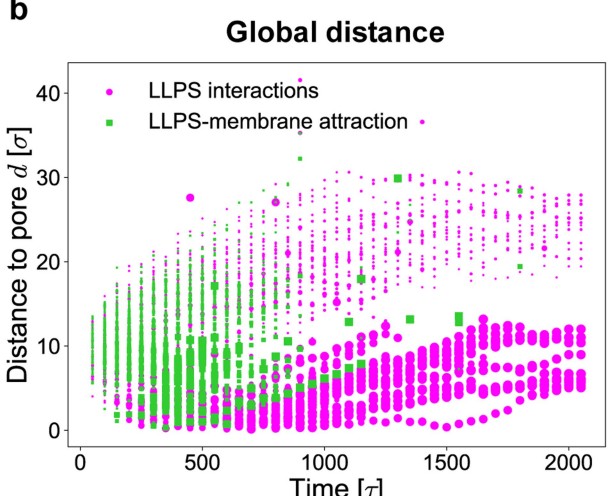
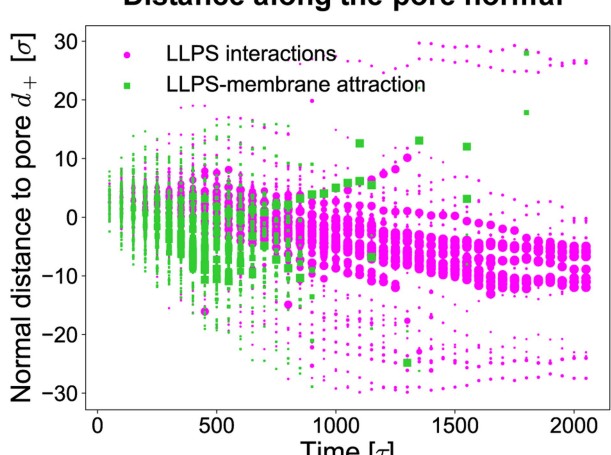

**b**

**Global distance**

**Distance along the pore normal**

**Extended Data Fig. 5 | Spatial distribution analysis related to Fig. 2.**
**a**, Droplets nucleate near the pore (blue points) – more than 99% of the condensate mass at $t < 200\,\tau$ is close to the pore ($d/r\_$drop$\leq 5$) and then, as time goes on, the system proceeds following one of two trajectory directions: i) The droplets diffuse away from the pore and remain small (vertical trajectory) – both for wetting and no-wetting interactions less than 5% of the condensate mass at $t > 1000\,\tau$ is observed away from the pore ($d/r\_$drop$>5$); ii) They grow and stay close to the pore (horizontal trajectory) – depending on the interactions,

we observe: ~96% of the condensate mass at $t > 1000\,\tau$ remains close to the pore ($d/r\_$drop$\leq 5$) without wetting interactions; ~98% of the condensate mass at $t > 1000\,\tau$ remains close to the pore ($d/r\_$drop$\leq 5$) with wetting interactions.
**b**, Points have size proportional to the droplet size (size (in points^2) = droplet mass/100. Only small droplets diffuse away from the pore as time goes on, the larger droplets remain close to the pore where they grow over time (valid for both wetting and no-wetting interactions).

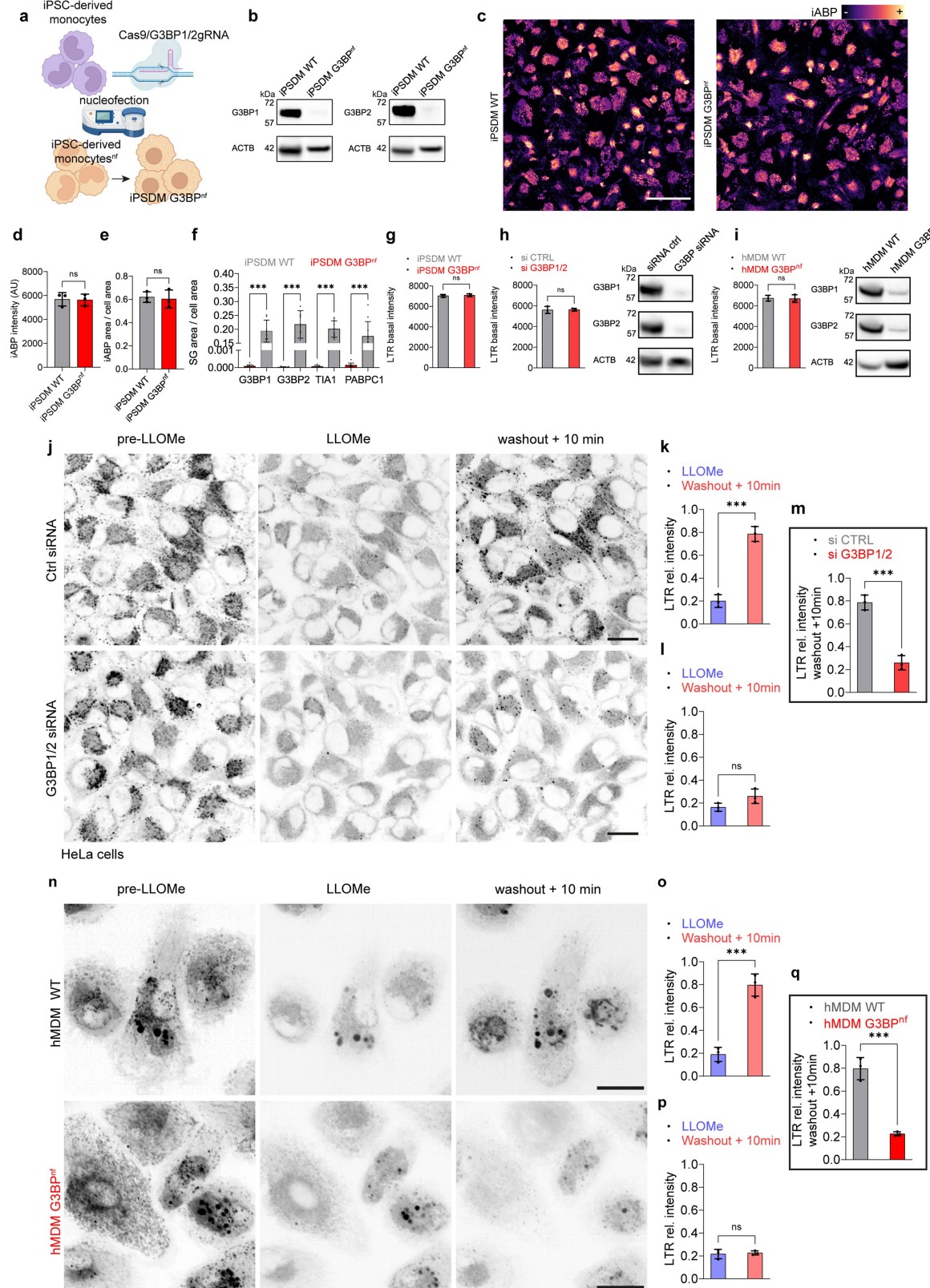

**Extended Data Fig. 6** | See next page for caption.

**Extended Data Fig. 6 | Supplementary data related to Fig. 3. a**, Scheme showing the experimental workflow targeting G3BP1 and G3BP2 using CRISPR/Cas9 system delivered as a ribonucleoprotein (RNP) complex by nucleofection. **b**, immunoblot showing G3BP1 and G3BP2 protein levels in iPSDM WT and iPSDM G3BP[nf], ACTB was used as a loading control (n = 3). **c**, representative images of proteolytic activity (magma colormap) in iPSDM WT and iPSDM G3BP[nf] incubated with the pan-cathepsin activity-based probe (iABP). Scale bar: 20 μm. **d**,**e**, Bar plots show iABP intensity quantification (d) and lysosomal area per cell evaluation (e) in iPSDM WT and iPSDM G3BP[nf], $n = 3$ independent experiments. **f**, high-content imaging SG evaluation in iPSDM WT and iPSDM G3BP[nf] left untreated or treated with LLOMe (1 mM, 30 min). Bar plots represent mean values ± SEM of one out of three independent experiments ($n = 3$), ***$P < 0.001$, two-way ANOVA, Šídák's multiple comparisons test. **g**, LysoTracker basal intensity quantification of iPSDM WT and iPSDM G3BP[nf], $n = 3$ independent experiments. **h-i**, LysoTracker basal intensity quantification and immunoblot showing G3BP1 and G3BP2 protein levels in HeLa WT and HeLa G3BP1/2 knockdown (KD) cells (h), and in hMDM WT and hMDM G3BP[nf] (i), ACTB was used as a loading control, $n = 3$ independent experiments. **j**, Image sequence of iPSDM WT and iPSDM G3BP[nf] incubated with LysoTracker (seen as black puncta) before adding LLOMe (left panel), 2 min after (central panel) and 10 min after washout (left panel). Scale bar: 10 μm. **k-l**, Show quantification of LTR puncta intensity relative to basal values (pre-LLOMe) for the indicated conditions in HeLa WT (k) and HeLa KD (l), $n = 3$ independent experiments, ***$P < 0.001$, two-tailed $t$-test. **m**, Shows the LTR intensity at 10 min after washout in HeLa WT compared to HeLa KD, $n = 3$ independent experiments. ***$P < 0.001$, two-tailed $t$-test. **n**, Image sequence of hMDM WT and hMDM G3BP[nf] incubated with LysoTracker (seen as black puncta) before adding LLOMe (left panel), 2 min after (central panel) and 10 min after washout (left panel). Scale bar: 10 μm. **o-p**, Show quantification of LTR puncta intensity relative to basal values (pre-LLOMe) for the indicated conditions in hMDM WT (o) and hMDM G3BP[nf] (p), $n = 3$ independent experiments. ***$P < 0.001$, two-tailed $t$-test. **q**, Shows the LTR intensity at 10 min after washout in hMDM WT compared to hMDM G3BP[nf], $n = 3$ independent experiments. ***$P < 0.001$, two-tailed $t$-test. Bar plots show mean ± SEM. For gel source data, see Supplementary Fig. 1.

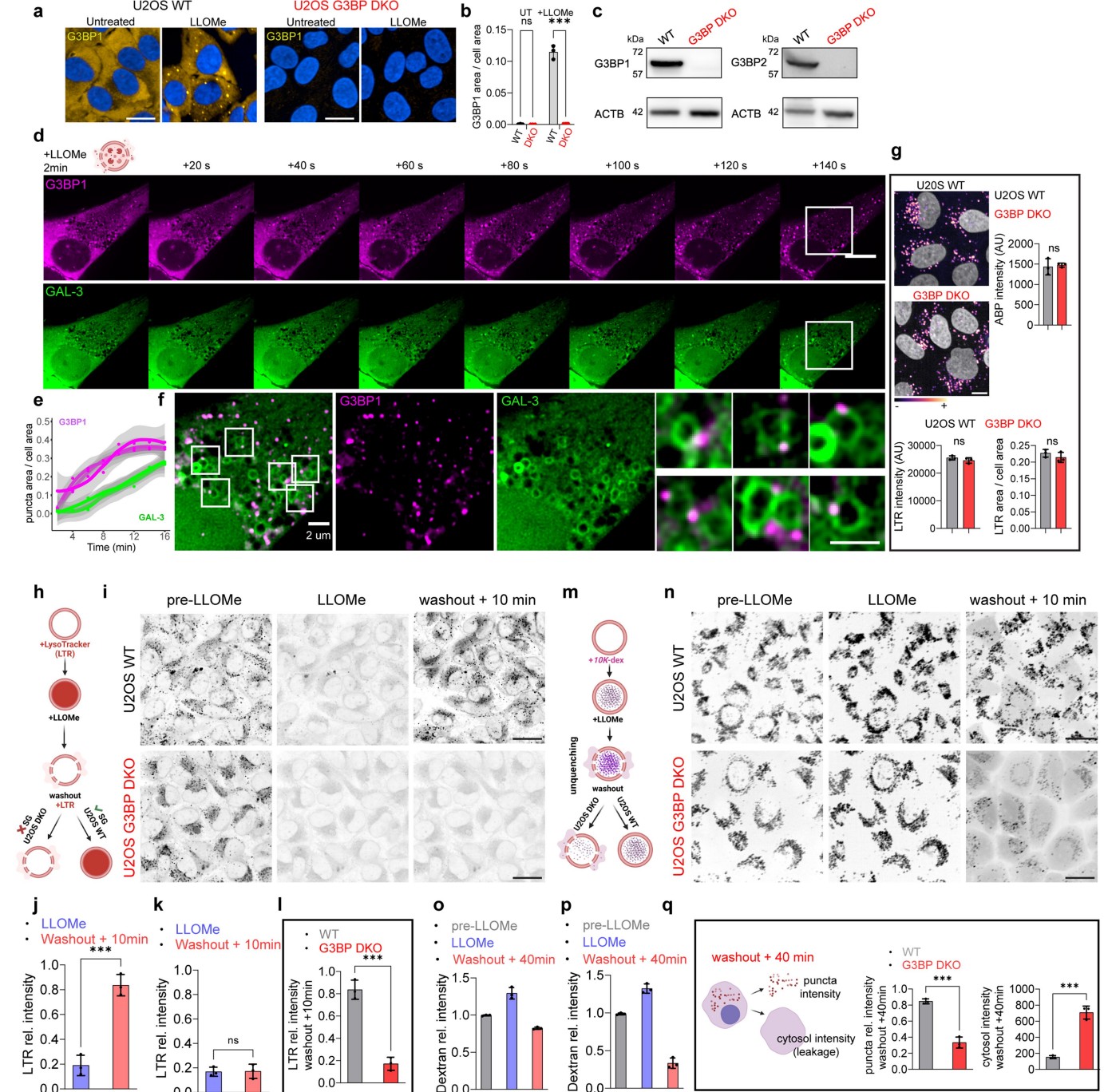

**Extended Data Fig. 7** | See next page for caption.

**Extended Data Fig. 7 | SG dynamics and functional characterisation using U2OS WT and U2OS G3BP double knockout cells. a-b**, Representative images (a) and high content image quantification (b) of U2OS WT and U2OS G3BP DKO cells left untreated or treated with LLOMe (2 mM, 30 min) and stained for G3BP1. Bar plots represent mean values ± SEM of one out of three independent experiments (*n* = 3), ***P* < 0.001, two-way ANOVA, Šídák's multiple comparisons test. **c**, immunoblot showing G3BP1 and G3BP2 protein levels in U2OS WT and U2OS G3BP DKO, ACTB was used as a loading control (n = 3). **d**, Live-cell imaging sequence (20 s time frame) of U2OS WT cells expressing G3BP1-GFP (magenta) and GAL-3 RFP (green) after 2 min of LLOMe treatment (2 mM). **e**, Shows the quantification of G3BP1 and GAL-3 puncta area over time, indicating that G3BP1-positive signal precedes GAL-3. The curve shows the mean and standard error (shadow areas), n = 3. **f**, Shows the zoom-in area indicated in (d) (white square), and different areas are further magnified (left panel) highlighting G3BP1⁺ granules interacting with damaged lysosomes (GAL-3⁺). **g**, Representative images of proteolytic activity (magma colormap) in U2OS WT and U2OS G3BP DKO cells incubated with the pan-cathepsin activity-based probe (iABP). Scale bar: 10 µm. Bar plots show iABP intensity quantification (top right), basal LysoTracker puncta intensity (bottom left) and lysosomal area (quantified as LTR area) normalised to cell area (bottom right). **h**, Scheme illustrating the lysosomal recovery assay using LysoTracker (LTR). **i**, image sequence of U2OS WT and U2OS G3BP DKO cells incubated with LysoTracker (seen as black puncta) before adding LLOMe (left panel), 2 min after (central panel) and 10 min after washout (left panel). Scale bar: 10 µm. See also Supplementary Video 15. **j-k**, Show quantification of LTR puncta intensity relative to basal values (pre-LLOMe) for the indicated conditions in U2OS WT (j) and U2OS G3BP DKO (k). *n* = 3 independent experiments. ***P* < 0.001, two-tailed *t*-test. **l**, Shows the LTR intensity at 10 min after washout in U2OS WT compared to U2OS G3BP DKO. *n* = 3 independent experiments. ***P* < 0.001, two-tailed *t*-test. **m**, Scheme illustrating the lysosomal recovery assay using a dextran-chase assay. Lysosomes loaded with 10 K dextran particles will be damaged using LLOMe and left recover for 40 min after washout (note that, unlike LTR, dextran particles will be unquenched after LLOMe addition). **n**, Image sequence of U2OS WT and U2OS G3BP DKO cells incubated with 10 K dextran (seen as black puncta) before adding LLOMe (left panel), 2 min after (central panel) and 40 min after washout (left panel). Scale bar: 10 µm. **o-p**, Show quantification of dextran puncta intensity relative to basal values (pre-LLOMe) for the indicated conditions in U2OS WT (o) and U2OS G3BP DKO (p). *n* = 3 independent experiments, ***P* < 0.001, one-way ANOVA, Dunnett's multiple comparisons test. **q**, Shows the dextran puncta intensity and the cytosolic intensity (leakage) at 40 min after washout in U2OS WT compared to U2OS G3BP DKO. Note that since lysosomes do not recover in U2OS G3BP DKO cells, the leakage of dextran particles continues over time resulting in increased fluorescence signal in the cytosol. Data represent the mean ± SEM. *n* = 3 independent experiments. ***P* < 0.001, two-tailed *t*-test. Scale bar: 10 µm. For gel source data, see Supplementary Fig. 1.

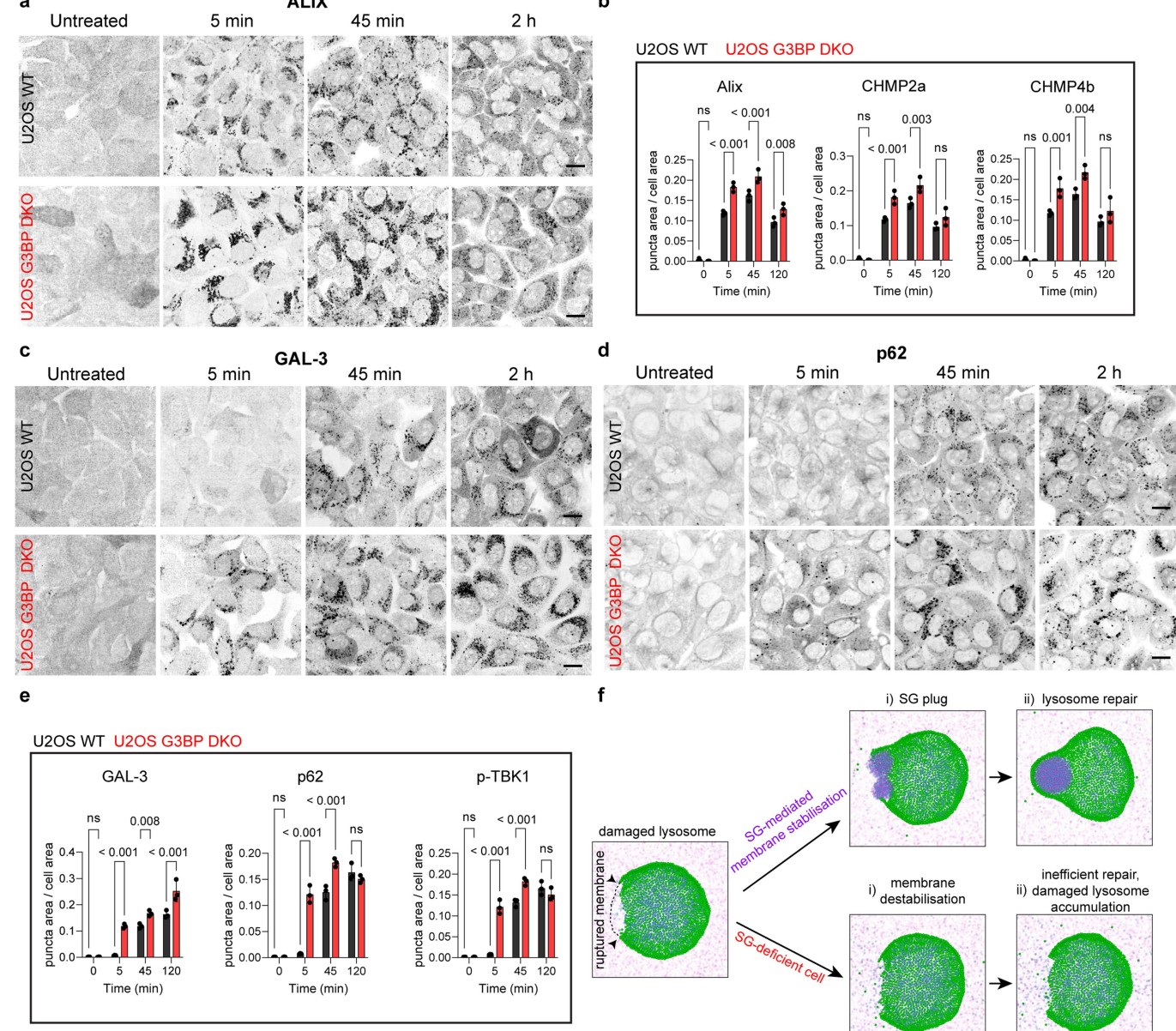

**Extended Data Fig. 8 | Evaluation of ESCRT and autophagy-related markers in U2OS WT and U2OS G3BP double knockout cells after lysosomal damage.** **a**, Representative images of U2OS WT and U2OS G3BP DKO cells left untreated or treated with LLOMe (2 mM) for 5 min, 45 min and 2 h, and stained for the CHMP4A-interacting protein ALIX. **b**, shows the high-content image quantification for the indicated markers in U2OS WT and U2OS G3BP DKO cells treated as in (a). $n \geq 900$ cells examined over three independent experiments. *P*-value was calculated using a two-way ANOVA, Šídák's multiple comparisons test. **c-d**, Representative images of U2OS WT and U2OS G3BP DKO cells left untreated or treated with LLOMe (2 mM) for 5 min, 45 min and 2 h, and stained for GAL-3 (c) and p62 (d). **e**, Shows the quantification for the indicated markers in U2OS WT and U2OS G3BP DKO cells treated as in (a). $n \geq 900$ cells examined over three independent experiments. *P*-value was calculated using a two-way ANOVA, Šídák's multiple comparisons test. **f**, Scheme using simulation snapshots -as shown in Fig. 2- illustrating the main outcomes after inducing lysosomal damage in U2OS WT and U2OS G3BP DKO cells. Data represent the mean ± SEM ($n$ = 3 independent experiments). Scale bar: 10 μm.

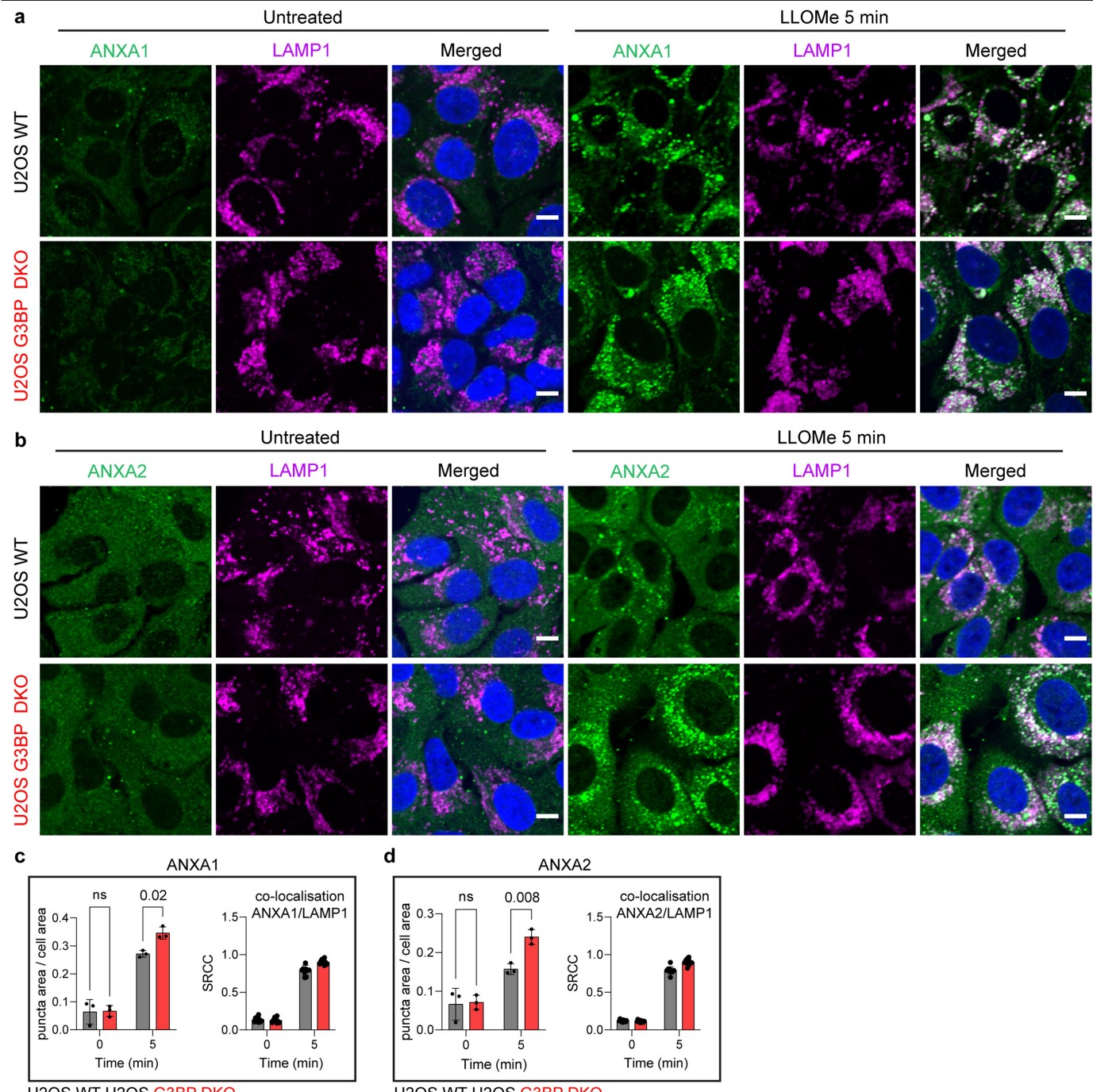

**Extended Data Fig. 9 | Evaluation of ANXA1 and ANXA2 in U2OS WT and U2OS G3BP double knockout cells after lysosomal damage. a,b**, Representative images of U2OS WT and U2OS G3BP DKO cells left untreated or treated with LLOMe (2 mM) for 5 min and stained for LAMP1 and the lysosomal repair-related proteins ANXA1 (a) and ANXA2 (b). **c,d**, High-content image quantification ($n \geq 300$ cells) of ANXA1 (c) and ANXA2 (d) puncta area per cell of cells treated as in (a,b). The graph on the right shows the corresponding co-localisation results with the lysosomal marker LAMP1. Data represent the mean ± SEM. $n \geq 900$ cells examined over three independent experiments. *P*-value was calculated using a two-way ANOVA, Šídák's multiple comparisons test. Scale bar: 10 μm.

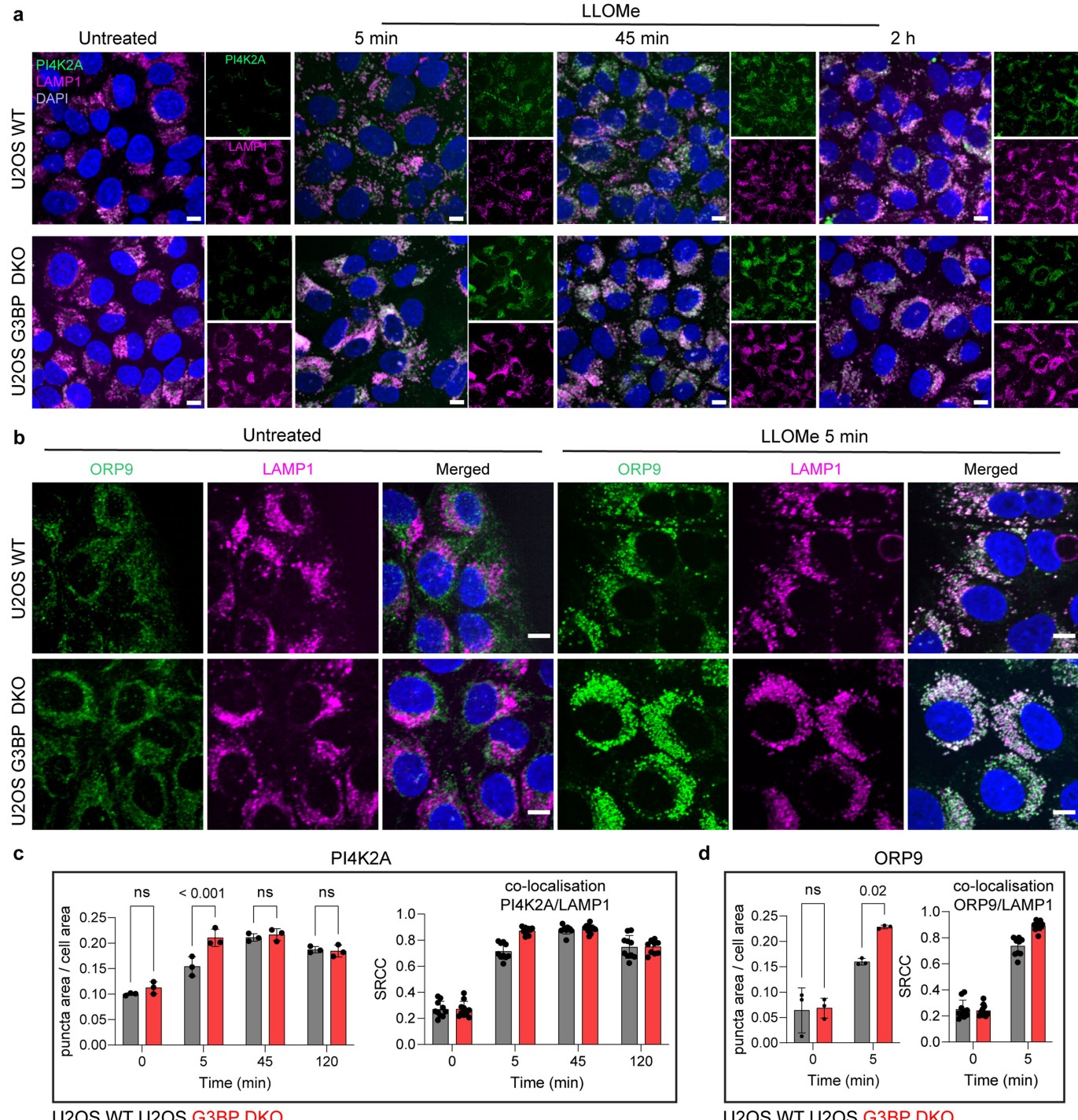

**Extended Data Fig. 10 | Evaluation of the phosphoinositide-initiated membrane tethering and lipid transport pathway (PITT)-related markers PI4K2A and ORP9 in U2OS WT and U2OS G3BP double knockout cells after lysosomal damage. a, b,** Representative images of U2OS WT and U2OS G3BP DKO cells left untreated or treated with 2 mM LLOMe for 5 min, 45 min and 2 h, and stained for PI4K2A (a), ORP9 (b) and LAMP1(magenta). Note that quickly after LLOMe treatment PI4K2A and ORP9 localise to lysosomes. **c,d,** Shows high-content image quantification (n ≥ 300 cells) of PI4K2A (c) and ORP9 (d) puncta area per cell of cells treated as in (a,b). The graph on the right shows the corresponding co-localisation results at the indicated time points with the lysosomal marker LAMP1. Data represent the mean ± SEM. *n* ≥ 900 cells examined over three independent experiments. *P*-value was calculated using a two-way ANOVA, Šídák's multiple comparisons test. Scale bar: 10 μm.

|---|---|

# Reporting Summary

## Statistics

For all statistical analyses, confirm that the following items are present in the figure legend, table legend, main text, or Methods section.

| n/a | Confirmed | |
|---|---|---|
| ☐ | ☒ | The exact sample size (*n*) for each experimental group/condition, given as a discrete number and unit of measurement |
| ☐ | ☒ | A statement on whether measurements were taken from distinct samples or whether the same sample was measured repeatedly |
| ☐ | ☒ | The statistical test(s) used AND whether they are one- or two-sided *Only common tests should be described solely by name; describe more complex techniques in the Methods section.* |
| ☒ | ☐ | A description of all covariates tested |
| ☐ | ☒ | A description of any assumptions or corrections, such as tests of normality and adjustment for multiple comparisons |
| ☐ | ☒ | A full description of the statistical parameters including central tendency (e.g. means) or other basic estimates (e.g. regression coefficient) AND variation (e.g. standard deviation) or associated estimates of uncertainty (e.g. confidence intervals) |
| ☐ | ☒ | For null hypothesis testing, the test statistic (e.g. *F*, *t*, *r*) with confidence intervals, effect sizes, degrees of freedom and *P* value noted *Give P values as exact values whenever suitable.* |
| ☒ | ☐ | For Bayesian analysis, information on the choice of priors and Markov chain Monte Carlo settings |
| ☒ | ☐ | For hierarchical and complex designs, identification of the appropriate level for tests and full reporting of outcomes |
| ☒ | ☐ | Estimates of effect sizes (e.g. Cohen's *d*, Pearson's *r*), indicating how they were calculated |

*Our web collection on statistics for biologists contains articles on many of the points above.*

## Software and code

Policy information about availability of computer code

| Data collection | VT-iSIM images were acquired using Olympus cellSens software. High-content images were acquired using Harmony 4.9 (Perkin Elmer). Western Blot membranes were obtained using an Amersham Imager 680 instrument (GE Life Sciences). |
|---|---|
| Data analysis | Image processing of confocal, VT-iSIM microscopy images and western blots: FIJI/ImageJ (version 2.1.0/1.53t). Image processing and deconvolution of VT-iSIM images were done using Huygens Essential software (Scientific Volume Imaging B.V, Netherlands, v 21.1). Statistical analysis was performed using Graph Pad Prism 10 software or R Studio 2023.03.0 (R 4.2.2). High-content imaging analysis was done using Harmony 4.9 (Perkin Elmer) and mean values were obtained using  Harmony 4.9 or R 4.2.2. The number of biological replicates and the statistical analysis performed and post hoc tests used can be found in the figure legends. RNA-seq heatmaps were done using Morpheus (https://software.broadinstitute.org/morpheus/). Spatial point pattern analysis was done using spatstat package in R (version 3.0-6) as specified in the methods section. Molecular dynamics analysis are specified in detail in the methods section. Custom code is available at https://github.com/Saric-Group and https://github.com/cvanhille/SGporecondensation |

For manuscripts utilizing custom algorithms or software that are central to the research but not yet described in published literature, software must be made available to editors and reviewers. We strongly encourage code deposition in a community repository (e.g. GitHub). See the Nature Portfolio guidelines for submitting code & software for further information.

## Data

Policy information about availability of data

All manuscripts must include a data availability statement. This statement should provide the following information, where applicable:

- Accession codes, unique identifiers, or web links for publicly available datasets
- A description of any restrictions on data availability
- For clinical datasets or third party data, please ensure that the statement adheres to our policy

Data availability: The data needed to evaluate the conclusions of the study are present in the manuscript or in the supplementary materials. Source data for the main and Extended Data figures are provided with this paper. Source data for gels and blots are provided as Supplementary information (Supplementary Figure 1). Source Data are provided for Figs. 1 to 4 and Extended Data 1 to 10. Code availability: Custom analysis codes were used to extract pore lifetime and solution exchange measurements from simulations. All analysis code used is available on a public GitHub repository (see data analysis and methods section).

## Human research participants

Policy information about studies involving human research participants and Sex and Gender in Research.

| Reporting on sex and gender | N/A |
|---|---|
| Population characteristics | N/A |
| Recruitment | N/A |
| Ethics oversight | N/A |

Note that full information on the approval of the study protocol must also be provided in the manuscript.

# Field-specific reporting

Please select the one below that is the best fit for your research. If you are not sure, read the appropriate sections before making your selection.

☒ Life sciences ☐ Behavioural & social sciences ☐ Ecological, evolutionary & environmental sciences

For a reference copy of the document with all sections, see nature.com/documents/nr-reporting-summary-flat.pdf

# Life sciences study design

All studies must disclose on these points even when the disclosure is negative.

| Sample size | No statistical method was used to predetermine sample size. Standard considerations based on expected variations from previous experiments [10.1126/science.aat9689, 10.15252/embj.2020104494] were applied to determine the necessary repeats to ensure reproducibility and statistical significance. The corresponding number of events that was analysed is indicated in the Figure legend or Methods section. |
|---|---|
| Data exclusions | No data were excluded from analysis. |
| Replication | We have indicated the number of independent experiments performed in the figure legends and additional information in the methods section. |
| Randomization | No randomization was performed for this study. Images were automatically acquired for the data analysis by high-content imaging. For super-resolution imaging the experimental setup included clearly defined conditions. To avoid bias, same software with identical settings between conditions was used for quantifications. |
| Blinding | No blinding was performed for this study. Blinding was not possible as all samples were analysed pairwise or multiple compared. |

# Reporting for specific materials, systems and methods

We require information from authors about some types of materials, experimental systems and methods used in many studies. Here, indicate whether each material, system or method listed is relevant to your study. If you are not sure if a list item applies to your research, read the appropriate section before selecting a response.

## Materials & experimental systems

| n/a | Involved in the study |
|---|---|
| ☐ | ☒ Antibodies |
| ☐ | ☒ Eukaryotic cell lines |
| ☒ | ☐ Palaeontology and archaeology |
| ☐ | ☒ Animals and other organisms |
| ☒ | ☐ Clinical data |
| ☒ | ☐ Dual use research of concern |

## Methods

| n/a | Involved in the study |
|---|---|
| ☒ | ☐ ChIP-seq |
| ☒ | ☐ Flow cytometry |
| ☒ | ☐ MRI-based neuroimaging |

# Antibodies

| Antibodies used | Antibodies were: anti-G3BP1 (13057-2-AP) or alternatively, anti-G3BP1 (66486-1-Ig), anti-G3BP2 (16276-1-AP) anti-TIA1 (12133-2-AP), anti-PABPC1 (10970-1-AP), anti-ALIX (12422-1-AP), anti-CHMP2a (10477-1-AP), anti-CHMP4b (13683-1-AP), anti-Annexin A1 (66344-1-Ig), anti-Annexin A2 (66035-1-Ig), anti-EIF3B (10319-1-AP), and anti-EIF4G1 (15704-1-AP) from Proteintech. Anti-Galectin-3 (125410) and anti-Lamp1 (121610) from Biolegend. Anti-p62 (GTX111393) from GeneTex. Anti-PI4K2a (B-5, sc390026), anti-ORP9 (A-7, sc398961), and anti-G3BP1-546 (sc-365338 AF546) from Santa Cruz. Anti phospho-eIF2α (Ser51) (9721), anti- eIF2α (9722) and anti-β-Actin (8H10D10, 12262), and anti-phospho TBK1 (5483T) from Cell Signalling Technology; and HRP-conjugated anti-mouse (W4021) and anti-rabbit (W4011) antibodies from Promega. |
|---|---|
| Validation | All the antibodies purchased have been validated in multiple previous studies accessible at the manufacturer's website. <br> anti-G3BP1 (13057-2-AP), https://www.ptglab.com/products/G3BP1-Antibody-13057-2-AP.htm <br> anti-G3BP1 (66486-1-Ig), https://www.ptglab.com/products/G3BP-Antibody-66486-1-Ig.htm <br> anti-G3BP1 Alexa Fluor 546 (sc-365338 AF546) https://datasheets.scbt.com/sc-365338.pdf <br> anti-G3BP2 (16276-1-AP), https://www.ptglab.com/products/G3BP2-Antibody-16276-1-AP.htm <br> anti-TIA1 (12133-2-AP), https://www.ptglab.com/products/TIA1-Antibody-12133-2-AP.htm <br> anti-PABPC1 (10970-1-AP), https://www.ptglab.com/products/PABPC1,PABP-Antibody-10970-1-AP.htm <br> anti-ALIX (12422-1-AP), https://www.ptglab.com/products/PDCD6IP-Antibody-12422-1-AP.htm <br> anti-CHMP2a (10477-1-AP), https://www.ptglab.com/products/CHMP2A-Antibody-10477-1-AP.htm <br> anti-CHMP4b (13683-1-AP), https://www.ptglab.com/products/CHMP4B-Antibody-13683-1-AP.htm <br> anti-Annexin A1 (66344-1-Ig), https://www.ptglab.com/products/ANXA1-Antibody-66344-1-Ig.htm <br> anti-Annexin A2 (66035-1-Ig), https://www.ptglab.com/products/ANXA2-Antibody-66035-1-Ig.htm <br> anti-EIF3B (10319-1-AP), https://www.ptglab.com/products/EIF3B-Antibody-10319-1-AP.htm <br> anti-EIF4G1 (15704-1-AP), https://www.ptglab.com/products/EIF4G1-Antibody-15704-1-AP.htm <br> Anti-Galectin-3 (125410), https://www.biolegend.com/fr-lu/products/alexa-fluor-488-anti-mouse-human-mac-2-galectin-3-antibody-7084?GroupID=BLG2786 <br> anti-Lamp1 (121610), https://www.biolegend.com/de-at/products/alexa-fluor-647-anti-mouse-cd107a-lamp-1-antibody-3589?GroupID=BLG4966 <br> Anti-p62 (GTX111393), https://www.genetex.com/Product/Detail/SQSTM1-P62-antibody/GTX111393 <br> anti-phospho TBK1 (5483T), https://www.cellsignal.com/products/primary-antibodies/phospho-tbk1-nak-ser172-d52c2-xp-rabbit-mab/5483?_requestid=3918912&gclid=Cj0KCQjwpc-oBhCGARIsAH6ote9PvI-e1E0t2t2P7d4LM9ghlrwW7TRzfpvqyiFjyIYyxoaU8XrxsQsaAsFyEALw_wcB&gclsrc=aw.ds&_requestid=4404764 <br> Anti-PI4K2a (B-5, sc390026), https://www.scbt.com/p/pi-4-kinase-ii-alpha-antibody-b-5 <br> anti-ORP9 (A-7, sc398961), https://www.scbt.com/p/orp-9-antibody-a-7 <br> phospho-eIF2α (Ser51) (9721), https://www.cellsignal.com/products/primary-antibodies/phospho-eif2a-ser51-antibody/9721 <br> anti- eIF2α (9722), https://www.cellsignal.com/products/primary-antibodies/eif2a-antibody/9722 <br> anti-β-Actin (8H10D10, 12262), https://www.cellsignal.com/products/antibody-conjugates/b-actin-8h10d10-mouse-mab-hrp-conjugate/12262#:~:text=Specificity%20%2F%20Sensitivity-,%CE%B2%2DActin%20(8H10D10)%20Mouse%20mAb%20(HRP%20Conjugate),react%20with%20cytoplasmic%20%CE%B3%2Dactin. <br> anti-mouse (W4021), https://www.promega.co.uk/products/protein-detection/primary-and-secondary-antibodies/anti_mouse-igg-h-and-l-hrp-conjugate/?catNum=W4021 <br> anti-rabbit (W4011), https://www.promega.co.uk/products/protein-detection/primary-and-secondary-antibodies/anti-rabbit-igg-h-and-l-hrp-conjugate/?catNum=W4011 |

# Eukaryotic cell lines

Policy information about cell lines and Sex and Gender in Research

| Cell line source(s) | KOLF2 human iPSCs, Public Health England Culture Collections, Cat#77650100. <br> The use of human cells is covered and approved by the Ethical Committee and regulated by the Francis Crick Institute Biological Safety Code of Practice in the project registered at the Crick (Project HTA17) framed under Human Tissue Authority Licence number 12650 <br> HeLa cells: Cell Services, The Francis Crick Institute. <br> U2OS WT and G3BP DKO cells: Paul Anderson Laboratory (Harvard University), 10.1083/jcb.201508028 <br> mEGFP-G3BP1 human iPSCs, Coriell Institute (AICS-0082-001) |
|---|---|
| Authentication | Authentication results for human mEGFP-G3BP1 iPSCs can be accessed at https://catalog.coriell.org/0/Sections/Search/Sample_Detail.aspx?Ref=AICS-0082-001&Product=CC Authentication results for KOLF2 human iPSCs can be accessed at the respective source's website. https://www.phe-culturecollections.org.uk/products/celllines/generalcell/search.jsp KOLF2 |

hiPSC are routinely authenticated at the lab by flow cytometry. Cell line authentication was initially performed by ATCC. Further authentication (U2OS, HeLa) was performed by microscopy at our lab and at The Francis Crick Cell Services unit.

Mycoplasma contamination

All cells tested negative for mycoplasma contamination.

Commonly misidentified lines
(See ICLAC register)

No ICLAC cell lines were used in this study.

# Animals and other research organisms

Policy information about studies involving animals; ARRIVE guidelines recommended for reporting animal research, and Sex and Gender in Research

Laboratory animals

Six- to eight-week-old, C3HeB/FeJ mice were used in this study. All mice were maintained in BSL3 cages, at 22°C ± 2°C and a relative humidity of 55 ± 10%.

Wild animals

No wild animals were used in this study.

Reporting on sex

ARRIVE guidelines and previous studies (10.1016/j.chom.2017.04.004) were followed to define animal cohorts. 5 animals per group were used per time of infection. Females were used for safety and space allocation restrictions as infected mice were contained in BSL3.

Field-collected samples

No field collected samples were used in this study.

Ethics oversight

All protocols for breeding and experiments were approved by the Home Office (U.K.) under project license P4D8F6075 and performed in accordance with the Animal Scientific Procedures Act, 1986.

Note that full information on the approval of the study protocol must also be provided in the manuscript.

