## [Peer Review File · Nature]

Manuscript Title: Stress granules plug and stabilise damaged endolysosomal membranes

Reviewer Comments & Author Rebuttals

Reviewer Reports on the Initial Version:

Referees' comments:

Referee #1 (Remarks to the Author):

Bussi et al describe the assembly of stress granule (SG)-like particles at sites of LLOMe-induced endolysosomal damage in iPS-derived macrophages. These granules do not appear in cells lacking G3BP1/2, proteins essential for the nucleation of SGs. In these cells, lysosomal integrity quantified by LysoTracker is reduced compared to WT cells suggesting that SG-like particles function to "plug" holes in damaged lysosomes. In a murine model of Mtb infection, SG-like particles are found in association with Mtb-containing lysosomes. This association is not seen in an Mtb mutant (Δ RD1) that does not induce lysosomal damage, consistent with the hypothesis that SG particles prevent lysosomal damage. Mtb growth is reduced in iPS-derived macrophages lacking G3BP1/2 consistent with the importance of endolysosomal damage for Mtb proliferation. Taken together, these results suggest that SG-like particles contribute to the repair of endomembrane damage in a way that may influence the proliferation of Mtb. The authors should address the following points:

1. The authors show that SG proteins TIA1 and PABPC1 co-localize with G3BP1/2 at sites of endomembrane damage. Several SG proteins are known to pre-associate in the cytoplasm prior to condensing into SGs in response to stress. It is therefore not clear whether the described particles bona fide SGs. It will be important to show that poly(A) mRNA is included in these endolysosomal sites.
2. LLOMe-induced G3BP particles are observed after 2 minutes (Fig. 1e), but LLOMe-induced phosphorylation of eIF2a is not observed until 20 minutes (Fig. E3c). This suggests that phospho-eIF2a is not responsible for the condensation event. Thus accumulation of untranslated mRNAs, the precursor to SG assembly, is not likely to be involved in the condensation event. This suggests that these G3BP particles may not be SGs.
3. In my view, the in vitro data using glycinin-containing giant unilamellar vesicles is a distraction and does not shed light on the phenomenon being studied in cells.
4. In the text, the experiments shown in Fig. 4d-f are said to use "human macrophages", but the figure legend says these are iPS-derived macrophages. This needs to be clarified. If they are not human macrophages, it is important that this phenomenon is confirmed in human macrophages.
5. The use of nucleofection to effect crispr-mediated G3BP knock down is a good start, but these results should be confirmed in cells with stable knock out of G3BP1/2.
6. The finding that G3BP-depletion reduces Mtb proliferation in both WT and DRD1 iPS macrophages (Fig. 4K) suggests that G3BP has effects on Mtb proliferation independent of lysosomal membrane stabilization.

Paul Anderson

Referee #2 (Remarks to the Author):

The manuscript by Bussi et al reports the interesting observation that damage to endolysosomal membranes leads to stress granule formation at the pores, which are plugged and then repaired. They make the case through in vitro experiments and simulations that the location of condensate formation is caused by the local mixing of the intra- and extraluminal content. Further the simulations suggest that wetting of the condensate on the membrane is important for the repair,

and this is in agreement with the final configurations of budded vesicles inside the repaired endolysosomes. Finally, the manuscript makes the case that this mechanism is important during infection with *Mycobacterium tuberculosis*, which results in endomembrane damage. This manuscript demonstrates a novel function of stress granules. Furthermore, it demonstrates that the fluid-like properties of the stress granules is important for this function, i.e. wetting of the membrane and therefore repair. The manuscript therefore fills a considerable gap in our knowledge on SG function. This is an exciting advancement given that SGs are amongst the best understood biomolecular condensates in terms of their formation (References 1-3 in the manuscript), but that their function is much less well understood. The current manuscript may not only provide a SG functional assay, but also makes a case for why the formation via phase separation is important for SG function. I expect that this manuscript will have wide-reaching impact in the SG, phase separation and membrane remodeling fields and support its publication in Nature after my comments below have been addressed.

Major comment:

1. The in vitro experiments do not reconstitute the SGs; rather they make use of a protein inside the vesicles that phase separates upon pore formation and exposure to acidic pH (on the outside). How does this relate to the natural system, where the endolysosome interior presumably has the low pH, and cytosolic components will get into contact with this milieu after damage? Does G3BP or other important stress granule components phase separate under such conditions? Can stress granule condensates wet the membrane? I do not think that this is necessarily a generic property. The authors need to justify usage of their model system and/or repeat some of their experiments with SG proteins.

Minor comments:

1. "Fluorescence recovery after photobleaching (FRAP) is used to test the mobility of the molecular components inside the condensates and verify their liquid character^{28,29}." References 28 and 29 are cautionary manuscripts against the overinterpretation of FRAP data. FRAP cannot unequivocally show that a liquid phase is present. Can the authors address this differently?
2. The text and the figure captions should be carefully edited.

Referee #3 (Remarks to the Author):

Summary of the key results:

The authors present work in which they argue stress granules preferential form at sites of endolysosomal damage. They combine in vitro, in cellula and in silico experiments to make a reasonable case that G3BP1-positive granules form at the sites of endolysosomal damage and offer protection against both biological and chemical disruption to lysosomes.

Originality and significance:

The work is largely novel, although a major missing piece of literature is a citation/mention of the work of Lai et al. PNAS 2012, who make analogous observations for septal pore-clogging proteins that gel during wound healing in *N. crassa*. I do not think this previous paper weakens the importance or novelty of this paper - if anything, it strengthens the case that this approach of plugging using condensates has evolved multiple times.

Data & methodology:

I found the figures very difficult to read. Bar charts should have labels under each bar. The protein components being labeled should be CLEARLY stated in the figure, and I would encourage the authors to provide context for what the figures are showing, either by using dashed white outlines to highlight cellular structures and/or using schematics to summarize the experiments.

Appropriate use of statistics and treatment of uncertainties

In general, yes, although some things could be explicitly quantified, as discussed below

Conclusions:

The conclusions are mostly robust and reasonable, although please see the listed suggestions.

Clarity and context:

In general the paper was well-written, although I felt many times that assumed knowledge regarding experiments, proteins, or re-agents was made. It's critical the authors explain their logic and what certain proteins, markers, or compounds mean and how they work.

Suggested improvements:

To be up front, I am generally supportive of publication but there are a number of issues I think should be addressed.

Major

The paper title states, "Stress granules plug and stabilise damaged endomembranes", but the evidence in the paper is exclusively for endolysosomal damage. This biophysically appears to agree with prior work demonstrating a pH-dependence for G3BP1 phase separation. With this in mind, unless the authors can show that SGs stabilize other types of endomembranes, I would encourage them to change this title to "Stress granules plug and stabilise damaged endolysosomal membranes" - there is, as far as I can see, no a priori expectation or evidence that SGs would form at other endomembrane damage site unless explicit recruitment were provided.

I found many (most?) of the figures often hard to make sense of. Components were not labeled, labels were in unclear places, it was unclear what we were looking at - I would encourage the authors to consider schematizing experiments so we, as a reader, can understand what the figures are reporting. Even the captions often lacked critical details. In short, the figures need a lot of work (aesthetically/graphically - the science looks good!)

The abstract claims "These biomolecular condensates act as a plug that stabilises damaged membranes and restricts the interchange of luminal contents" -I think this is a reasonable inference given the data, but without a direct demonstration that SG formation prevents the exchange of lysosomal contents I would suggest this should not be claimed in the abstract.

I find the temporal evolution of the system reported in Fig. 1 slightly unintuitive. The model introduces verbally in the abstract implies SG form and plug a damaged site, THEN ESCRT-dependent repair processes occur. However, Fig.1 appears to show the concomitant arrival of galectin-3 and G3BP1 at the site of damage, where galectin-3 has previously been reported to coordinate ESCRT-dependent repair (Jia et al., Dev Cell 2020). With this in mind, I think if the argument is that G3BP1 precedes galectin-3 arrival at sites of damage, this should be quantified (which could be done by plotting the integrated G3BP1 signal and integrated galectin-3 signal vs. time at an individual lysosome and comparing the trace kinetics). If G3BP1 truly arrives first, then I think the original model is fine. If galectin-3 arrives first, then I think the authors need to perform an experiment in a +/- GAL-3 background to confirm that G3BP1 recruitment is not dependent on galectin-3, OR alter the model/language to make clear they cannot rule out galectin-3-dependent recruitment. I realize this is annoying, but I think it's important to be sure here, otherwise, this really appealing model will become lore without the relative temporal order having been directly addressed.

The first section states, "Stress granules rapidly condensate in the proximity of damaged endolysosomes", but I don't actually see quantification to support this statement. I think this would require showing that the formation of SGs is closer to lysoendosomal damage than expected

by random chance (i.e. if SGs were to randomly nucleate uniformly across the cytosol, do we see the average distance to lysosomal sites is closer than expected by random chance). This is a central claim to the paper, and I think needs to be explicitly quantified (which can probably be done easily from existing data).

The authors state, "70% of these events G3BP1-positive granules formed a plug pattern closely associated with the GAL-3-positive damaged endolysosome", but surely the converse analysis is more relevant (i.e. what percentage of GAL-3 positive damaged endolysosomes have G3BP1-positive granules). If this is very low, it speaks to EITHER this not being a core mechanism OR that LLOMe triggers uniform and widespread lysosomal damage leading to G3BP1 exhaustion from the cytosol. If this second explanation is the case, the authors may consider examining LLOMe-dependent lysosomal damage with G3BP1 driven via transient transfection to crank up the protein levels really high. The "advantage" that transient transfection would offer here is non-uniformity because the key thing the authors want to see is cells where prior to LLOMe-damage no G3BP1-positive granules exist, and after LLOMe-damage many/most galectin-3 positive sites co-localize with G3BP1 plugs. The challenge here is above a certain cellular concentration, G3BP1 will start to form condensates regardless of stress.

The authors propose that pH changes could induce G3BP1 condensation and cite prior work on other proteins, but G3BP1 itself has explicitly been shown to undergo RNA-independent condensation at low pH (Fig 4B Guillen-Boixet et al 2020). This is potentially an important piece of the authors' mechanism in that canonically, G3BP1 condensation requires RNA for SG formation, yet endosomal damage does not necessarily coincide with RNA release, so why would G3BP1 condense? The answer is likely explained by the pH dependence previously described. Seems like a massive missed opportunity not to make this connection as it offers a clear example as to WHY condensation would be relevant here - two entirely orthogonal physicochemical inputs that can trigger analogous physical outputs. If this is the case, however, it does suggest this mechanism is probably only going to be limited to lysosomal damage?

In the paragraph "Distinct types of interactions between G3BP1-positive SG and LAMP1-positive endolysosomal..." - I honestly did not follow the point this was trying to make. What's LAMP-1? Is this further enhancing the message? The figures (both main text and extended data) again were again hard to understand. Maybe there was an important point made here, but it was lost on me.

In Fig. 2, what components are labeled in magenta vs. green? The authors compare examples when condensates either form or don't form, but how this is achieved is not described in the main text or the figure capture. This information makes it very difficult to evaluate the data, although I accept the general point appears to be well-made. Also, what is 'blocking' the pore? Something is there, but I don't know what it is because the figure doesn't explain what is labeled.

It seems, from the field of view in 3c, that there are FAR fewer lysozymes in the G3BP-nf cells pre-LLOMe than post. The intensity may be similar, but surely the number of lysozymes is an important factor here? I'm not a lysosomal biologist, so I defer to the field standard here, but visually speaking, if you'd shown me the iPSDM-WT pre-LLOMe vs. iPSDM-G3BP-nf pre-LLOMe I would have said they were definitely different, and I don't think they're supposed to be given the interpretation of the data. This feels like it needs to be explicitly addressed, or I worry readers may think this is masking the real behavior, weakening the impact of the paper.

I would like to see some speculation on if/how SGs recruit ESCRT machinery, or if their ONLY role is to stabilize.

Minor

In the abstract the authors state "...stress granules (SG) rapidly condensate in the proximity..." - "condensate" should be "condense"

"...membrane-less and membrane-bound condensates likely correspond to different functional states" - I'm unconvinced ref: 12 is relevant here?

When referencing the importance of G3BP1 in SG formation, I would encourage the authors to cite the original work by Nancy Kedersha Kedersha et al. JCB 2016.

The sentence "Cytosol- biomolecular condensate interactions have been reported to play a role in cellular organisation" is pretty vapid... I'm not sure cytosolic condensates have any choice other than to interact with the cytosol, where they exist.

I would define endomembrane damage explicitly - don't give a reader the opportunity to be confused about a core term in the paper.

"Because of the membrane biophysical properties," - this is incredibly non-specific (basically akin to 'because membranes are membranes') I would explicitly explain why endomembrane damage makes organelles unstable.

The significance/role of Galectin-3 is never explained; without this information, the significance of G3BP1-positive granules near GAL-3 positive endolysosomes is unclear.

The authors should explain how LLOMe works (one short sentence).

For FRAP analysis (Extended data 3E) the authors should show normalized FRAP (i.e. 100 to 0 at $t=0$) - this reports on the actual recovery from the reference frame of the condensate, otherwise your reporting a convolution of absolute recruitment into condensate vs. kinetics which makes it hard to figure out a reasonable $t=1/2$. I also don't understand how extended data 3e has led to the assessment that you get 60% recovery after 10 minutes when the lowest recovery is ever at is ~45%. Hence the importance of using a normalized curve and reporting $t=1/2$ and the immobile fraction.

FRAP does not verify liquidity, it simply reports of re-arrangement time of a specific labeled component. This is not a big issue, but the property being examined by FRAP should be corrected. Also the 10 min recovery for FRAP is pretty slow, it might be worth the authors comparing this value to numbers in the literature and asking if perhaps interaction with the endolysosomal damaged is rewiring SG dynamics...

I would break panel 3A (G3BP1^{nf} vs. WT) into two panels because it took me many reads to understand what was going on here. I also found d/e INCREDIBLY difficult to follow, because iPSDM and WT look like the bar names.

Stating that ZNFX1 has "has been previously identified in SGs" does the reference a disservice; I had expected this just to be a proteomics paper but the reference explicitly makes the point that ZNFX1-mutations lead to SG deficiencies. This is strong (albeit anecdotal) evidence in the authors'

favor.

Fig. 3 caption, LTR is not defined (suspect it's LysoTracker Red, please use full name).

Referee #4 (Remarks to the Author):

In this article, Bussi et al propose a new mechanism by which damage to the lysosomal endomembrane is repaired. Through the use of cell culture, *in vivo* and *in silico* analysis they find that lysosomal membrane damage triggers formation of stress granules at the site of the membrane damage. They find this depends on the pH and ion gradient across the endomembrane and suggest that the SG condensate that forms functions as a plug to stabilize the damaged membrane. They test the physiological role of this process by implicating this process in curtailing WT Mtb infection.

This work reproduces many of the findings described by a recent study (Reference #46) that identified SG forms at the lysosome membrane that is damaged by LLOMe and Mtb infection. This diminishes the novelty and also impacts on the interpretation of the findings reported here. Further, while authors propose a SG plug model for repair and show SGs form on Galectin-3 labeled lysosome, they do not present any data to address how (and if) this links with the ubiquitous lysosome membrane repair machinery involving ESCRT-III recruitment by Galectin-3 and other proteins. Their claim is that SG forms a plug to stabilize the lysosomal membrane and facilitate its repair. This is complicated by the literature showing that SG formation occurs via biochemical process triggered by cell stress and phosphorylation of eIFa, which impacts on the mTOR for lysosomal and cell homeostasis (Reference #43, 46). These and the specific issues below dampen overall enthusiasm for this study as it stands.

1. G3BP1 binding to lysosome is known to occur even when SG formation is inhibited (Reference #43, 46). So, if ion homeostasis alters SG plug formation at the injured lysosomal membrane by LLPS, authors should examine plug formation in presence of cycloheximide or when eIFa phosphorylation is inhibited.
2. Related to the above experiment, it is unclear how a process that requires new protein synthesis occurs rapidly enough to plug the lysosome membrane leak within seconds as shown by *in silico* modeling studies. This also raises the question regarding the kinetics of this process mediated by eIF2 phosphorylation, which does not even initiate until 20 minutes post LLOMe treatment (extended data 3), while SG formation peaks by 5-7 minutes (figure 1).
3. The claimed biomolecular condensate function of SG plug is based on *in vitro* modeling of this using the plant protein glycinin. If and how this relates to endogenous mammalian proteins in the SGs is unclear and this preliminary observation should be extended by reconstituting SG-mediated plugging of membrane leak using G3BP1 and other mammalian SG components (Freibaum et al JCB 2021; 220(3): e202009079).
4. To extend the *in vivo* relevance of the findings, authors allude to the inherited deficiency in patients of the SG protein ZNFX1. To support this speculation there is a need to examine if this protein is in the G3BP1 containing SG plugs that form at the damaged membrane or if its absence impacts the SG plug formation/lysosome membrane repair.
5. In figure 3, acidification of lysosomes is used as a readout of lysosomal membrane integrity. However, lack of G3BP1 will impair mTOR and other stress response of cells independent of SG condensate formation which in turn will impair the ability of these stressed cells to handle ion and pH homeostasis. Thus, this does not offer unambiguous evidence for the claim presented. A direct readout such as leakage of larger dyes / dextran etc. from lysosomes should be examined for

assessing lysosomal membrane integrity.

6. Data shown in figure 4 identifies that the Δ RD1-infected cells do not form G3BP1 condensates as this strain does not lead to lysosomal damage. Nonetheless, iPSDM G3BP^{nf} cells show similar fold increase in infection by Δ RD1 as WT bacteria (Fig. 4k). This argues in favor of a non-SG plug formation role of G3BP1 in affecting Mtb infection, and against author's conclusion that "...blockade of SG formation, critically affects the outcome of Mtb infection in macrophages."

7. The data presented here has been examined in macrophages, but would be expected to be relevant to other cell types as well and should be tested.

8. Minor - In extended data 3, PABPC1 and G3BP1 were used interchangeably for the control (untreated) and experimental (BAF1 treated) cells, it is unclear why this was done, but it should be corrected.

Author Rebuttals to Initial Comments:

Point by point response to reviewers.

Referee #1:

Bussi et al describe the assembly of stress granule (SG)-like particles at sites of LLOMe-induced endolysosomal damage in iPS-derived macrophages. These granules do not appear in cells lacking G3BP1/2, proteins essential for the nucleation of SGs. In these cells, lysosomal integrity quantified by LysoTracker is reduced compared to WT cells suggesting that SG-like particles function to “plug” holes in damaged lysosomes. In a murine model of Mtb infection, SG-like particles are found in association with Mtb-containing lysosomes. This association is not seen in an Mtb mutant (DRD1) that does not induce lysosomal damage, consistent with the hypothesis that SG particles prevent lysosomal damage. Mtb growth is reduced in iPS-derived macrophages lacking G3BP1/2 consistent with the importance of endolysosomal damage for Mtb proliferation. Taken together, these results suggest that SG-like particles contribute to the repair of endomembrane damage in a way that may influence the proliferation of Mtb. The authors should address the following points:

1. The authors show that SG proteins TIA1 and PABPC1 co-localize with G3BP1/2 at sites of endomembrane damage. Several SG proteins are known to pre-associate in the cytoplasm prior to condensing into SGs in response to stress. It is therefore not clear whether the described particles bona fide SGs. It will be important to show that poly(A) mRNA is included in these endolysosomal sites.

We thank the reviewer for highlighting this point. We have now performed these experiments and imaged poly(A) mRNA. We show that poly(A)-RNA colocalises with G3BP1-positive granules and displayed similar interactions with GAL-3-positive lysosomal damage sites (Extended Data Figure 1). We have also expanded the characterisation of SG markers and found that the canonical components EIF3B and EIF4G are also co-localising with G3BP1-positive granules (Extended Data Figure 1). In line with these results, we observed by high-content imaging analysis that cycloheximide and emetine treatment (10.1083/jcb.151.6.1257) blocked SG formation in LLOMe-treated cells.

2. LLOMe-induced G3BP particles are observed after 2 minutes (Fig. 1e), but LLOMe-induced phosphorylation of eIF2a is not observed until 20 minutes (Fig. E3c). This suggests that phospho-eIF2a is not responsible for the condensation event. Thus accumulation of untranslated mRNAs, the precursor to SG assembly, is not likely to be involved in the condensation event. This suggests that these G3BP particles may not be SGs.

We thank the reviewer for bringing this to our attention. Regarding the Western blot studies, the concentration of LLOMe was inadvertently omitted in the legend of the figure. The concentration was 250 μ M, not 1 mM as in the rest of the experiments. In fact, we used this strategy because 1 mM was inducing a very rapid phosphorylation of eIF2 α , and the aim of the experiment was to test if lysosomal damage could induce eIF2 α phosphorylation. However, we did not consider matching the imaging kinetics. We apologise for any confusion caused by this oversight. We now realise that this discrepancy may have led to confusion, and we have now included the data with LLOMe 1mM (that will induce lysosomal damage faster than 250 μ m). Notably, as depicted in Extended Data 2, exposure to 1mM LLOMe induces a rapid increase in p-eIF2 α levels within minutes, consistent with our imaging data. Moreover, the new additional data argue that the G3BP1-positive granules we observe are bona-fide SGs.

3. In my view, the *in vitro* data using glycinin-containing giant unilamellar vesicles is a distraction and does not shed light on the phenomenon being studied in cells.

We thank the reviewer for this comment. We believe that the *in vitro* experiments are relevant not only to support the observations in cells, but also because we are proposing a new function for biomolecular condensates. *In vitro* (and *in silico*) experiments not only show that membrane pore plugging by condensates can occur, but also provide insight about some key factors that help stabilising the membrane, like the wetting affinity between the membrane and the condensate. To address this point, we have performed additional experiments with condensates formed by G3BP1 and poly(A) RNA following the suggestions of the other reviewers (see new Figure 2 and Extended Data 3) which strengthens the conclusions that:

- Membrane damage can trigger localised phase separation.

- The phase-separated condensate can wet the membrane allowing for pore stabilisation.

We consider that these results strengthen the main hypothesis of this work, and also represent a key finding for the biomolecular condensate field, opening new directions for the research of condensate-membrane damage interactions. The fact that the results can also be reproduced with a plant protein, suggests that this could be a conserved mechanism as pointed out by reviewer 3, and now is discussed in the main text.

4. In the text, the experiments shown in Fig. 4d-f are said to use “human macrophages”, but the figure legend says these are iPS-derived macrophages. This needs to be clarified. If they are not human macrophages, it is important that this phenomenon is confirmed in human macrophages.

We appreciate the comment from the reviewer and would like to provide clarification regarding the use of induced pluripotent stem cell-derived macrophages (iPSDM) in our study. We would like to highlight that the iPSDM used in our research are indeed human macrophages, generated through a differentiation process similar to blood-derived monocytes (10.1242/jcs.252973, 10.1038/s41467-022-30557-4). In the previous version of the manuscript, we presented G3BP1+/PABPC1+ granules in blood-derived macrophages following lysosomal damage (Extended Data 2). However, due to space constraints, we were unable to include a full panel as we did with iPSDM. Nonetheless, we obtained similar results (data not shown) and decided to include a single panel demonstrating G3BP1+/PABPC1+ granules. To address potential confusion, we have excluded that panel in this revised version (new Extended Data 1), and instead incorporated new data showing that G3BP-nucleofected blood-derived macrophages exhibit a comparable impairment in lysosomal repair to human iPSDM (new Extended Data 6).

5. The use of nucleofection to effect crispr-mediated G3BP knock down is a good start, but these results should be confirmed in cells with stable knock out of G3BP1/2.

We thank the reviewer for this constructive suggestion. We have now incorporated several new datasets (new Extended Data 7-10) done in U2OS WT and U2OS G3BP1/2 double KO cells (10.1083/jcb.201508028). In agreement with our previous results, we observed a severe impairment in the lysosomal recovery capacity and lysosomal repair of U2OS G3BP DKO cells after inducing lysosomal damage (see also Supp Video 15).

6. The finding that G3BP-depletion reduces Mtb proliferation in both WT and DRD1 iPS macrophages (Fig. 4K) suggests that G3BP has effects on Mtb proliferation independent of lysosomal membrane stabilization.

We thank the reviewer for this important consideration. *Mycobacterium tuberculosis* lacking RD1 (Mtb Δ RD1) is restricted in its ability to induce endosomal damage. However, the bacterium still has the lipid PDIM, which enhances macrophage phagosomal permeabilization and membrane damage (10.1111/cmi.12726). This means that in long-term experiments, macrophages infected with Mtb Δ RD1 will still exhibit some degree of endolysosomal damage. These characteristics explain why we observed the main differences in Mtb Δ RD1 growth between WT and G3BP1/2^{mf}- macrophages after 72 hours of culture but not at earlier time points (in contrast with Mtb WT). We have now included the growth curve with the earlier time points, where the growth difference between Mtb WT and Mtb Δ RD1 are better illustrated and evident already after 48 hours of infection (new Figure 4). Due to space constraints and the focus of this work, we did not further expand on this finding. However, we acknowledge this point and, in line with the most recent findings, we have revised the text to state that Mtb Δ RD1 is a strain that is "severely restricted" in its ability to induce endomembrane damage, but not "unable." We have also updated the text mentioning the role of the bacterial lipids.

Referee #2:

The manuscript by Bussi et al reports the interesting observation that damage to endolysosomal membranes leads to stress granule formation at the pores, which are plugged and then repaired. They make the case through in vitro experiments and simulations that the location of condensate formation is caused by the local mixing of the intra- and extraluminal content. Further the simulations suggest that wetting of the condensate on the membrane is important for the repair, and this is in agreement with the final configurations of budded vesicles inside the repaired endolysosomes. Finally, the manuscript

makes the case that this mechanism is important during infection with *Mycobacterium tuberculosis*, which results in endomembrane damage.

This manuscript demonstrates a novel function of stress granules. Furthermore, it demonstrates that the fluid-like properties of the stress granules is important for this function, i.e. wetting of the membrane and therefore repair. The manuscript therefore fills a considerable gap in our knowledge on SG function. This is an exciting advancement given that SGs are amongst the best understood biomolecular condensates in terms of their formation (References 1-3 in the manuscript), but that their function is much less well understood. The current manuscript may not only provide a SG functional assay, but also makes a case for why the formation via phase separation is important for SG function. I expect that this manuscript will have wide-reaching impact in the SG, phase separation and membrane remodeling fields and support its publication in *Nature* after my comments below have been addressed.

We are grateful for the reviewer's positive and constructive comments. We sincerely believe these comments helped us to improve the strength and clarity of our manuscript.

1. The *in vitro* experiments do not reconstitute the SGs; rather they make use of a protein inside the vesicles that phase separates upon pore formation and exposure to acidic pH (on the outside). How does this relate to the natural system, where the endolysosome interior presumably has the low pH, and cytosolic components will get into contact with this milieu after damage? Does G3BP or other important stress granule components phase separate under such conditions? Can stress granule condensates wet the membrane? I do not think that this is necessarily a generic property. The authors need to justify usage of their model system and/or repeat some of their experiments with SG proteins.

We thank the reviewer for the comments and suggestions. We would like to address first the question regarding the experimental system: we do not see a reason as to why the directionality would affect the results, i.e., exchanging the inside components for the outside components. The trigger of phase separation is provided by the mixing of the components at the pore generated in the membrane, i.e. the protein will phase separate when it encounters a higher proton concentration (lower pH), or a higher NaCl concentration as we showed for glycinin, or a higher proton concentration and RNA in the case of G3BP1 (see below). The system was built with the protein on the inside because the protein concentration is the main limiting factor; otherwise very large amounts of proteins would be needed to flush through the microfluidic system to exchange the external solution. In addition, having the fluorescent protein on the inside facilitates the observation of the pore and condensate formation; while if it was on the outside, the predominant fluorescent signal in the field of view would make the experiment less clear. Finally, because of the large size of the vesicles, the membrane can be considered as practically flat and the curvature is not expected to play a role. We have now clarified this and justified the experimental model in the methods section of the manuscript.

In the previous version of the manuscript, we chose glycinin as a model protein to trigger phase separation upon damage, because the phase diagrams for this protein in different conditions such as pH, temperature and NaCl concentration are known (10.1021/acsmacrolett.0c00709), and the interaction of glycinin condensates with membranes has been recently reported in detail (10.1038/s41467-023-37955-2). This makes glycinin a robust model protein to explore different conditions for triggering phase separation, as we showed in the previous version of the manuscript by changing the pH or the NaCl concentration. Additionally, glycinin can be easily purified in large quantities, which also facilitates these experiments requiring substantial amounts of protein to load the vesicles. However, we understand and agree with the reviewer's concern regarding the extrapolation of the results obtained with glycinin to stress granules, and for that reason we have performed new experiments as follows. We reconstituted stress granules *in vitro* following Guillén-Boixet et al. *Cell* 2020, 10.1016/j.cell.2020.03.049 (see Extended Figure 3). We updated Figure 2 showing that G3BP1 condensation triggered by poly(A) RNA at pH 5 can plug and stabilize damaged membranes. We believe these results clearly strengthen the hypothesis of our work, and in addition suggest that damage stabilization by condensates might be a shared property for condensates formed at pore sites and interacting with membranes. Note that irrespective of the composition of the condensates (protein, polymers, etc.), wetting transitions at membranes can occur by modifying the milieu conditions (10.1038/s41467-023-37955-2). Thus, from a physico-chemical point of view, if a condensate is formed at the site of pore formation and has a strong wetting affinity to the membrane, it might be able to plug and stabilize the damage, as we showed previously for glycinin and now also for G3BP1/RNA. We believe this is a key finding for the stress

granule field, but also has a general relevance for the field of biomolecular condensates, since we assign a new function for biomolecular condensates interacting with damaged membranes. We have now discussed this in the manuscript, and we believe the work has improved thanks to the reviewer comments and suggestions.

2. “Fluorescence recovery after photobleaching (FRAP) is used to test the mobility of the molecular components inside the condensates and verify their liquid character^{28,29}.” References 28 and 29 are cautionary manuscripts against the overinterpretation of FRAP data. FRAP cannot unequivocally show that a liquid phase is present. Can the authors address this differently?

We completely agree with the reviewer and apologise if our claim seemed to be an overstatement. We conducted the FRAP experiments because they are widely used in the SG literature, and our aim was to expand the characterization of the SGs observed *in cellulo*. However, as also mentioned by Reviewer 3, we cannot exclude that the G3BP1 granules we observe after damage are fully membrane-less, given the rapid formation of these structures at membrane damage sites. In addition, even G3BP1 and G3BP2 have been shown to be associated to the lysosomal membrane protein LAMP1 at homeostatic conditions (independently of SG formation) (10.1016/j.cell.2020.12.024, 10.1016/j.neuron.2023.05.033)

In line with these results, our FRAP analysis show a slower recovery than the observed for G3BP1 condensates *in vitro* (or *in cellulo* after other type of stress, 10.1016/j.cell.2020.03.049) and as suggested by reviewer 3, we believe this is due to the dynamic interactions with lysosomal membranes (see reply to Reviewer 3). Overall, and considering the additional new data (see reply to Reviewer 1) we realised that adding the FRAP studies here would not necessarily reinforce the message of our work and it might shift the aim of this study. We therefore decided to include them for review purpose only (see Reply to Reviewer 3).

On the other hand, as part of our new *in vitro* studies (Fig 2 and Extended Data 3), we have reconstituted G3BP1/poly(A)-RNA granules *in vitro* and characterise them using FRAP showing quick and almost full recovery. To confirm the fluidity, we also show droplet coalescence completed within seconds, which confirms the liquid-like properties for these condensates as previously reported (10.1016/j.cell.2020.03.049).

In addition, we also observed G3BP1 foci coalescing with one another and undergoing shape changes *in cellulo* (see figure below).

The image (for review purpose only) shows a zoom-in area from Supp. Video 8 (Fig. 1h) where the dynamics of G3BP1 puncta (after inducing lysosomal damage) illustrate fusion and shape changes events suggesting liquid-like properties. Scale bar:2 μ m

The text and the figure captions should be carefully edited.

Thanks for raising this point. We have now carefully gone through the text and the captions and have proofread the text.

Referee #3:

Summary of the key results:

The authors present work in which they argue stress granules preferential form at sites of endolysosomal damage. They combine *in vitro*, *in cellulo* and *in silico* experiments to make a reasonable case that

G3BP1-positive granules form at the sites of endolysosomal damage and offer protection against both biological and chemical disruption to lysosomes.

The work is largely novel, although a major missing piece of literature is a citation/mention of the work of Lai et al. PNAS 2012, who make analogous observations for septal pore-clogging proteins that gel during wound healing in *N. crassa*. I do not think this previous paper weakens the importance or novelty of this paper - if anything, it strengthens the case that this approach of plugging using condensates has evolved multiple times.

Thank you for bringing up this exciting point, we have now included this work as part of our manuscript's discussion. Although we acknowledge the significance of their previous paper, we believe that our research brings both novelty and valuable mechanistic insights, while describing plugging as a general mechanism for ruptured membrane stabilisation.

I found the figures very difficult to read. Bar charts should have labels under each bar. The protein components being labeled should be CLEARLY stated in the figure, and I would encourage the authors to provide context for what the figures are showing, either by using dashed white outlines to highlight cellular structures and/or using schematics to summarize the experiments.

We thank the reviewer for this observation. We have re-organised many of the figures and revised the labels and added more schematics summarising the experiments. We hope the figures are now clearer.

To be up front, I am generally supportive of publication but there are a number of issues I think should be addressed.

Thank you very much for the supportive statements.

The paper title states, "Stress granules plug and stabilise damaged endomembranes", but the evidence in the paper is exclusively for endolysosomal damage. This biophysically appears to agree with prior work demonstrating a pH-dependence for G3BP1 phase separation. With this in mind, unless the authors can show that SGs stabilize other types of endomembranes, I would encourage them to change this title to "Stress granules plug and stabilise damaged endolysosomal membranes" - there is, as far as I can see, no a priori expectation or evidence that SGs would form at other endomembrane damage site unless explicit recruitment were provided.

We agree with the reviewer that "endolysosomal" is a more accurate term for our work, and we have now changed the title to "Stress granules plug and stabilise damaged endolysosomal membranes".

I found many (most?) of the figures often hard to make sense of. Components were not labeled, labels were in unclear places, it was unclear what we were looking at - I would encourage the authors to consider schematizing experiments so we, as a reader, can understand what the figures are reporting. Even the captions often lacked critical details. In short, the figures need a lot of work (aesthetically/graphically - the science looks good!)

Thanks for this constructive feedback. We now proofread the figures, added sketches and tried to clarify specific points and make them clearer and appropriately labelled.

The abstract claims "These biomolecular condensates act as a plug that stabilises damaged membranes and restricts the interchange of luminal contents" -I think this is a reasonable inference given the data, but without a direct demonstration that SG formation prevents the exchange of lysosomal contents I would suggest this should not be claimed in the abstract.

Thanks for highlighting this important point. The *in silico* data actually show that plugging prevents interchange of luminal contents (Figure 2, and Extended Data 4). In addition, we believe this is also exemplified *in cellulo* by the lysotracker leakage and recovery assay, where in the absence of SG formation, the luminal content is lost and the lysosomal population does not recover. We have now added new data using U2OS G3BP double KO cells and a dextran chase assay that provide additional evidence regarding this point. After damage, we can see that G3BP double KO cells displayed a severe impairment in their lysosomal repair capacity and the luminal content (dextran particles) leaks over time, which results in an increased cytoplasmic fluorescence (leaked content) (Extended Data 7m-q). Finally, the *in vitro* studies support the notion of the condensates plugging micron-sized pores because

in the absence of the condensate plug, the giant vesicles would not retain their spherical shape maintaining the Laplace pressure across the membrane.

I find the temporal evolution of the system reported in Fig. 1 slightly unintuitive. The model introduced verbally in the abstract implies SG form and plug a damaged site, THEN ESCRT-dependent repair processes occur. However, Fig.1 appears to show the concomitant arrival of galectin-3 and G3BP1 at the site of damage, where galectin-3 has previously been reported to coordinate ESCRT-dependent repair (Jia et al., Dev Cell 2020). With this in mind, I think if the argument is that G3BP1 precedes galectin-3 arrival at sites of damage, this should be quantified (which could be done by plotting the integrated G3BP1 signal and integrated galectin-3 signal vs. time at an individual lysosome and comparing the trace kinetics). If G3BP1 truly arrives first, then I think the original model is fine. If galectin-3 arrives first, then I think the authors need to perform an experiment in a *-/-* GAL-3 background to confirm that G3BP1 recruitment is not dependent on galectin-3, OR alter the model/language to make clear they cannot rule out galectin-3-dependent recruitment. I realize this is annoying, but I think it's important to be sure here, otherwise, this really appealing model will become lore without the relative temporal order having been directly addressed.

This is a very important point, and we fully agree with the reviewer. To answer this point in macrophages, we have now included new data where we increase the temporal resolution of our live-cell imaging experiments to approx. 1s. As suggested, we plotted the G3BP1 and GAL-3 intensities over time and we observed that G3BP1 puncta appears before GAL-3 (new Figure 1h,i). We would also like to note that SG would not necessarily need to be recruited to the damage site, in fact we believe they are formed at the site of damage given that the core G3BP proteins have already been shown to associate with the lysosomal protein LAMP1 at basal (in the absence of stress) conditions. In line with these results, we also observed similar kinetics using U2OS cells (new Extended Data 6d-f)

The first section states, “Stress granules rapidly condensate in the proximity of damaged endolysosomes”, but I don't actually see quantification to support this statement. I think this would require showing that the formation of SGs is closer to lysoendosomal damage than expected by random chance (i.e. if SGs were to randomly nucleate uniformly across the cytosol, do we see the average distance to lysoendosomal sites is closer than expected by random chance). This is a central claim to the paper, and I think needs to be explicitly quantified (which can probably be done easily from existing data).

We completely agree this is a critical claim and we thank the reviewer for the constructive comment. We have now added new datasets using spatial pattern analysis (*in cellulo*) and spatial density distribution (*in silico*) and we confirm that SG formation at damage sites is closer than expected by random chance (new Fig. 1e and Extended Data 5).

The authors state, “70% of these events G3BP1-positive granules formed a plug pattern closely associated with the GAL-3-positive damaged endolysosome”, but surely the converse analysis is more relevant (i.e. what percentage of GAL-3 positive damaged endolysosomes have G3BP1-positive granules). If this is very low, it speaks to EITHER this not being a core mechanism OR that LLOMe triggers uniform and widespread lysosomal damage leading to G3BP1 exhaustion from the cytosol. If this second explanation is the case, the authors may consider examining LLOMe-dependent lysosomal damage with G3BP1 driven via transient transfection to crank up the protein levels really high. The “advantage” that transient transfection would offer here is non-uniformity because the key thing the authors want to see is cells where prior to LLOMe-damage no G3BP1-positive granules exist, and after LLOMe-damage many/most galectin-3 positive sites co-localize with G3BP1 plugs. The challenge here is above a certain cellular concentration, G3BP1 will start to form condensates regardless of stress. We thank the reviewer for raising this point. We have now included the percentage of GAL-3 positive damaged endolysosomes that are positive for G3BP1-positive granules. As shown in Fig. 1d we found that most of the GAL-3 positive puncta are positive for G3BP1 (more than 90%).

The authors propose that pH changes could induce G3BP1 condensation and cite prior work on other proteins, but G3BP1 itself has explicitly been shown to undergo RNA-independent condensation at low pH (Fig 4B Guillen-Boixet et al 2020). This is potentially an important piece of the authors' mechanism

in that canonically, G3BP1 condensation requires RNA for SG formation, yet endosomal damage does not necessarily coincide with RNA release, so why would G3BP1 condense? The answer is likely explained by the pH dependence previously described. Seems like a massive missed opportunity not to make this connection as it offers a clear example as to WHY condensation would be relevant here - two entirely orthogonal physicochemical inputs that can trigger analogous physical outputs. If this is the case, however, it does suggest this mechanism is probably only going to be limited to lysosomal damage?

We appreciate the reviewer for bringing up this exciting observation. We fully agree with the reviewer's statement that pH serves as the primary trigger in this context, and our results align with this hypothesis. By blocking lysosomal acidification using the vATPase inhibitor Bafilomycin A1 and inducing lysosomal damage through silica crystals (which physically damage the membrane without requiring proteolytic processing), we do not observe SG formation (Extended Data 2g,h). However, we still detect GAL-3+ damaged lysosomes in this experimental setup. Furthermore, our *in vitro* findings demonstrate that a sudden decrease in pH alone is sufficient to initiate the phase separation. We have now mentioned the connection to Guillen-Boixet et al 2020 in the manuscript.

While we also concur with the reviewer's perspective regarding the restriction of this particular mechanism to lysosomal damage, we cannot rule out the possibility that other signals, such as a sudden change in specific ions, might also trigger the formation of these condensates following other types of membrane damage. We believe that this represents an intriguing avenue for further research that our study has opened. We now show that condensate plugging can stabilise damage membranes for condensates formed by very different proteins, namely G3BP1 and glycinin, and by different triggers for phase separation, like lowering the pH or changing the ionic concentration. This strongly suggests that the mechanism of localised phase separation and damage stabilisation can be induced by different triggers and might be a general behaviour for condensates forming at pores and wetting the membrane.

In the paragraph “Distinct types of interactions between G3BP1-positive SG and LAMP1-positive endolysosomal...” - I honestly did not follow the point this was trying to make. What's LAMP-1? Is this further enhancing the message? The figures (both main text and extended data) again were again hard to understand. Maybe there was an important point made here, but it was lost on me. We thank the reviewer for this comment. LAMP1 is a commonly used endolysosomal marker (together with LAMP2 constitute 50% of all lysosomal membrane proteins). The idea of the figure is to illustrate different types of condensate-membrane interactions across a cellular z-stack given that not all the range of interactions is covered in a single z-section (of approximate 150 nm). By showing the images from different regions in z (axial direction) we can appreciate better the range of interactions and identify G3BP1 granules at the surface of the lysosome but also inner granules and some of them “trapped” inside inner vesicles. These structures also match with the observed simulations where condensate content is not only wetting the membrane but also engulfed during vesicle sealing. We have now added a brief scheme to make the figure clearer.

In Fig. 2, what components are labeled in magenta vs. green? The authors compare examples when condensates either form or don't form, but how this is achieved is not described in the main text or the figure capture. This information make it very difficult to evaluate the data, although I accept the general point appears to be well-made. Also, what is ‘blocking’ the pore? Something is there, but I don't know what it is because the figure doesn't explain what is labeled.

We apologise for the lack of clarity and information in this Figure, we have now updated it and make it clearer both in the Figure's legend and in the main text that the condensate blocks the pore.

It seems, from the field of view in 3c, that there are FAR fewer lysozymes in the G3BP-nf cells pre-LLOMe than post. The intensity may be similar, but surely the number of lysozymes is an important factor here? I'm not an lysosomal biologist, so I defer to the field standard here, but visually speaking, if you'd shown me the iPSDM-WT pre-LLOMe vs. iPSDM-G3BP-nf pre-LLOMe I would have said they were definitely different, and I don't think they're supposed to be given the interpretation of the data. This feels like it needs to be explicitly addressed, or I worry readers may think this is masking the real behavior, weakening the impact of the paper.

We thank the reviewer for raising this important observation. Unlike many commonly used cell lines, macrophages present a high degree of lysosomal morphology heterogeneity even at the intracellular level (see attached example). We have analysed the basal proteolytic activity and lysosomal content of macrophages WT and nucleofected for G3BP1/2 and we did not find differences (Extended Data 6c-e), and similar results were obtained using U2OS WT and G3BP double KO cells (Extended Data 7g), as well as knockdown HeLa cells and nucleofected blood-derived macrophages (Extended Data 6). However, and considering the broad audience for this journal, we do agree with this reviewer that that figure could lead to misinterpretations. Given the focus of this manuscript and the space restrictions to further explain these phenotypes, we decided to select regions from different fields of view that present

more comparable lysosomal morphology (new Figure 3). We also matched the figure style with the rest of the lysosomal recovery assays shown in the manuscript for clarity and consistency.

Figure showing untreated iPSDM (basal conditions) incubated with LysoTracker red (seen as black puncta). Note the lysosomal heterogeneity both at the morphological and distribution level. Scale bar: 10 μ m

I would like to see some speculation on if/how SGs recruit ESCRT machinery, or if their ONLY role is to stabilize.

We greatly appreciate the reviewer for suggesting this exciting experiment, which we believe has broadened the scope of our work. In response to this suggestion, we have conducted additional experiments using U2OS WT and G3BP double KO cells. The results reveal that the ESCRT and autophagy-related machinery, including GAL-3, p62, and p-TBK1, are recruited and accumulate more rapidly in U2OS G3BP DKO cells. These findings support the notion of stress granules playing an upstream role in stabilising membranes. Furthermore, they demonstrate that if these condensates fail to form, resulting in unstable ruptured membranes, even with recruitment of the repair machinery, lysosomes still exhibit leakage, leading to the accumulation of damaged lysosomes (see Extended Data 7-10). Additionally, we have explored ESCRT-independent repair pathways as reported recently, and our results align with similar observations (see Extended Data 9-10). Moreover, we believe that an intriguing research area that emerges from our study is the potential role of condensates as “reaction hubs” that facilitate or enhance the activity of the endolysosomal repair machinery.

In the abstract the authors state “...stress granules (SG) rapidly condensate in the proximity...” - “condensate” should be “condense”

We have mentioned it correctly now.

“...membrane-less and membrane-bound condensates likely correspond to different functional states” – I’m unconvinced ref: 12 is relevant here?

We thank the reviewer for this comment. We have tried to select, the manuscripts that have explored in detail membrane-condensate interactions and help to better understand the context of our sentence. We have included Li et al. because it was, to the best of our knowledge, the first report on condensates wetting transitions in membranes, which paved the way for understanding and quantifying the interaction between condensates and membranes.

When referencing the importance of G3BP1 in SG formation, I would encourage the authors to cite the original work by Nancy Kedersha Kedersha et al. JCB 2016.

We agree with this reviewer and we unintentionally omitted this important manuscript, which we have now added it.

The sentence “Cytosol– biomolecular condensate interactions have been reported to play a role in cellular organisation” is pretty vapid... I’m not sure cytosolic condensates have any choice other than to interact with the cytosol, where they exist.

We agree with the reviewer, for space restrictions, we could not further expand on this subject, but we meant to highlight the crosstalk between condensates and membrane-bound organelle interactions. We changed the word “cytosol” for “membrane-bound organelle” now to make clearer this point.

I would define endomembrane damage explicitly - don’t give a reader the opportunity to be confused about a core term in the paper.

We thank the reviewer for bringing this important point, we have defined it now in the introduction of the manuscript as “loss of membrane integrity, either through rupture or poration”

“Because of the membrane biophysical properties,” - this is incredibly non-specific (basically akin to ‘because membranes are membranes’) I would explicitly explain why endomembrane damage makes organelles unstable.

We thank the reviewer for noticing this, and we are sorry we have overlooked it. Now the phrase reads: “Upon membrane damage, pores form making the vesicles unstable and prone to collapse if the ruptured area cannot be sealed (10.1002/advs.202004068).”

The significance/role of Galectin-3 is never explained; without this information, the significance of G3BP1-positive granules near GAL-3 positive endolysosomes is unclear.

We thank the reviewer for this comment, we have explained now why we used GAL-3 and defined it as a “cytosolic lectin that binds to exposed glycans on damaged lysosomes, serving as a marker of endomembrane damage” 10.1080/15548627.2015.1063871

The authors should explain how LLOMe works (one short sentence).

Thank you for this comment, we have mentioned it now.

For FRAP analysis (Extended data 3E) the authors should show normalized FRAP (i.e. 100 to 0 at $t=0$) - this reports on the actual recovery from the reference frame of the condensate, otherwise your reporting a convolution of absolute recruitment into condensate vs. kinetics which makes it hard to figure out a reasonable $t=1/2$. I also don’t understand how extended data 3e has led to the assessment that you get 60% recovery after 10 minutes when the lowest recovery is ever at is ~45%. Hence the importance of using a normalized curve and reporting $t=1/2$ and the immobile fraction. FRAP does not verify liquidity, it simply reports of re-arrangement time of a specific labeled component. This is not a big issue, but the property being examined by FRAP should be corrected. Also the 10 min recovery for FRAP is pretty slow, it might be worth the authors comparing this value to numbers in the literature and asking if perhaps interaction with the endolysosomal damaged is rewiring SG dynamics...

We thank the reviewer for this constructive comment. We have re-plotted our data using normalised values and indeed observed a slow FRAP recovery *in cellulo* compared to similar observations in the literature (10.1016/j.cell.2020.03.049). As suggested by the reviewer, we believe this is due to the dynamic interactions between damaged membranes and SG, which may not occur or be less relevant under other stress conditions. Additionally, even if we do not observe a positive signal for a membrane damage marker, we cannot exclude the possibility that in some cases, the condensate is already engulfed or associated with GAL-3-negative (“healthy”) lysosomes. Furthermore, even under basal conditions, G3BP1/2 interactions with the lysosomal protein LAMP1 might also play a role (10.1016/j.cell.2020.12.024, 10.1016/j.neuron.2023.05.033). As we mentioned to reviewer 1, considering the new data we provide now and the main focus of this work, we have decided to include these results solely for review purposes. This is to avoid any misunderstanding or overstatements regarding the fluid-like properties of SG.

A

B

A. a representative FRAP experiment sequence is shown where iPSDM expressing G3BP1-GFP and GAL-3-RFP were treated with LLOMe (1mM, 15 min) and G3BP1 condensates subjected to FRAP analysis. Scale bars: 10 μ m (main) and 2 μ m (magnified). B. Smooth curve representing the mean FRAP curve with 95% confidence interval, the immobile fraction (IF) and half-time recovery ($t_{1/2}$) values are depicted ($n=3$).

I would break panel 3A (G3BP1nf vs. WT) into two panels because it took me many reads to understand what was going on here. I also found d/e INCREDIBLY difficult to follow, because iPSDM and WT look like the bar names.

Thank you very much for raising this point. We have now separated the panel in two and also rearranged the figure for clarity and consistency with the new figures where we also evaluate lysosomal recovery in G3BP-deficient/KO cells.

Stating that ZNFX1 has “has been previously identified in SGs” does the reference a disservice; I had expected this just to be a proteomics paper but the reference explicitly makes the point that ZNFX1-mutations lead to SG deficiencies. This is strong (albeit anecdotal) evidence in the authors’ favor.

Thank you very much for this supportive comment. We agree the mechanism we provide here could be relevant for several diseases where different agents trigger lysosomal membrane rupture. We have done additional IF studies and we found that ZNFX1 co-localises with G3BP1 in *M. tuberculosis*-infected macrophages and it also associates to membrane damage sites (new Fig. 4).

Fig. 3 caption, LTR is not defined (suspect it's LysoTracker Red, please use full name). We have added now a scheme to illustrate the lysosomal recovery assay and define the LysoTracker abbreviation. We also make it clearer in the figure’s legend.

Referee #4:

In this article, Bussi et al propose a new mechanism by which damage to the lysosomal endomembrane is repaired. Through the use of cell culture, in vivo and in silico analysis they find that lysosomal membrane damage triggers formation of stress granules at the site of the membrane damage. They find this depends on the pH and ion gradient across the endomembrane and suggest that the SG condensate that forms functions as a plug to stabilize the damaged membrane. They test the physiological role of this process by implicating this process in curtailing WT Mtb infection.

This work reproduces many of the findings described by a recent study (Reference #46) that identified SG forms at the lysosome membrane that is damaged by LLOMe and Mtb infection. This diminishes the novelty and also impacts on the interpretation of the findings reported here.

We thank the reviewer for appreciating the new mechanism of membrane stabilisation and repair we are proposing here. We respectfully disagree with the reviewer that ref46 (Jia et al., 2022) diminished the novelty of our work.

First, we would like to emphasize that the main finding of our manuscript *is not* on *SG formation* after lysosomal damage *but on addressing* a major gap in the field that is **1-** the understanding of how damaged endomembranes are stabilised, and **2-** reporting a biological function for SG in this context. Therefore, the novelty of our manuscript does not rely on what triggers SG formation (e.g., lysosomal damage) but in assigning a function for SG that can be extended to other biomolecular condensates as positively highlighted by other reviewers.

Regarding the work of Jia et al., the authors neither mention nor hypothesise a role for SG in their study, which remains a leading question in the cell biology of biomolecular condensates (10.1016/j.molcel.2022.05.014). Furthermore, the authors do not provide sufficient evidence to support the claim that SG forms at lysosomal membrane damage sites, as their study lacks high-resolution analysis to support this conclusion. The assertion that SG *interacts or associates* with damaged lysosomes, but does not form at the damage sites, is primarily based on biochemical (IP) studies and is not complemented with quantitative spatiotemporal imaging studies, including co-localisation analysis of SG with endomembrane damage markers. It is important to note that the authors mainly use the terms "associated" and occasionally "recruited" when referring to the IP studies, but they do not explicitly state that "SG form at the damage sites". Significantly, the authors themselves acknowledge this apparent limitation of the study and clearly state: "...By confocal fluorescence microscopy, the majority of G3BP1-positive SGs formed during lysosomal damage were either independent of lysosomes or at best juxtaposed to lysosomes." Moreover, it is worth noting that the graphical abstract of reference 46 correctly illustrates SG separated from lysosomes, as the authors could not conclude that SG interacts with lysosomes.

In conclusion, the study did not report a function for SG, and the lack of in-depth quantitative dynamics studies makes it difficult to conclude if the induction of these granules is spatially linked to the endomembrane damage response or it is a secondary effect of other downstream pathways triggered after lysosomal damage. We have now clearly stated this point in our manuscript.

Further, while authors propose a SG plug model for repair and show SGs form on Galectin-3 labeled lysosome, they do not present any data to address how (and if) this links with the ubiquitous lysosome membrane repair machinery involving ESCRT-III recruitment by Galectin-3 and other proteins.

This is an important observation; we thank the reviewer as these comments inspired us to perform a whole new set of experiments that we believe have significantly expanded the scope of our study (Extended Data 7-10, Supplementary Video 15). Building upon Reviewer 1's suggestion to validate the functional aspects in stable G3BP1/2 KO cells, we have conducted additional experiments using U2OS WT and G3BP DKO cells to address this concern. Our findings indicate that in LLOMe-treated G3BP DKO cells, as compared to WT cells:

- 1- There is increased recruitment of both ESCRT-dependent and independent repair machinery.
- 2- There is faster recruitment of Galectin-3 (GAL-3) and lysophagy adaptors.
- 3- The lack of a membrane damage stabilising mechanism mediated by SG leads to the accumulation of damaged lysosomes.

Our *in vitro*, *in cellulo*, and *in silico* data support a model wherein the absence of an SG plug, which serves to stabilise the ruptured membrane, leads to enhanced membrane leakage (attributable to vesicle instability and increase in pore size) and exposure of lysosomal luminal components. This, in turn, results in heightened recruitment of the repair machinery and the accumulation of damaged lysosomes. Furthermore, our observations in G3BP1/2 DKO cells, where we observed increased recruitment of a repair machinery but a lack of lysosomal recovery following damage, suggest an additional role for SG beyond their stabilising function. We envision a scenario in which SG also serve as reaction hubs, facilitating the concentration and efficient functioning of the membrane repair machinery components. This represents an exciting future research direction that we are currently exploring and it is beyond the scope of this study.

Their claim is that SG forms a plug to stabilize the lysosomal membrane and facilitate its repair. This is complicated by the literature showing that SG formation occurs via biochemical process triggered by cell stress and phosphorylation of eIFa, which impacts on the mTOR for lysosomal and cell homeostasis (Reference #43, 46). These and the specific issues below dampen overall enthusiasm for this study as it stands.

We respectfully disagree with this reviewer's comment. We show that SG are rapidly formed after lysosomal damage and this has been reported to be a rapid process overall (10.1016/j.bbamcr.2020.118876, 10.1016/j.cell.2020.03.049). Moreover, previous evidence show that G3BP1/2 directly bind LAMP1 (10.1016/j.cell.2020.12.024, 10.1016/j.neuron.2023.05.033). Considering G3BP1/2 are the main nucleating factors mediating SG formation, our data argue for a model where SG are rapidly formed after acute changes (induced by lysosomal damage) in the presence of well-known phase-separation triggers, such as a decrease in the pH, which aligns with a previous study of G3BP1 condensates in vitro 10.1016/j.cell.2020.03.049 (also emphasised by reviewer 3), and with our in cellulo, in vitro and in silico data.

G3BP1 binding to lysosome is known to occur even when SG formation is inhibited (Reference #43, 46). So, if ion homeostasis alters SG plug formation at the injured lysosomal membrane by LLPS, authors should examine plug formation in presence of cycloheximide or when eIFa phosphorylation is inhibited.

We thank the reviewer for this comment. We do not see any conflicting results between our proposed mechanism (that is SG formation dependent) and the fact that G3BP1 is bound to LAMP1 (as shown in ref43-10.1016/j.cell.2020.12.024-, more recently in 10.1016/j.neuron.2023.05.033 and suggested by Fig1G in ref46-Jia et al., 2022-) independently of SG formation. In fact, this further supports our proposed model where SG formation occurs rapidly and locally because the recruitment of the core components (G3BP1/2) to the lysosomes is not necessarily required (they are already bound to lysosomes). We are unsure we understand what the reviewer means by altered SG formation. Nonetheless, we have now done experiments with cycloheximide and emetine and, in agreement with our results and previous observations (10.1083/jcb.151.6.1257, 10.1083/jcb.200502088), we found that these compounds block SG formation (Extended Data 2).

Related to the above experiment, it is unclear how a process that requires new protein synthesis occurs rapidly enough to plug the lysosome membrane leak within seconds as shown by in silico modeling studies. This also raises the question regarding the kinetics of this process mediated by eIF2 phosphorylation, which does not even initiate until 20 minutes post LLOMe treatment (extended data 3), while SG formation peaks by 5-7 minutes (figure 1).

We thank the reviewer for this comment. Phase separation or SG condensate formation is not a process requiring new protein synthesis but driven by multivalent macromolecular interactions (<https://doi.org/10.1016/j.cell.2020.03.056>, 10.1038/s41392-021-00678-1, 10.1038/nrm.2017.7). Regarding the Western blot studies, the concentration of LLOMe was inadvertently omitted in the figure's legend. The concentration was 250 μ M, not 1 mM as in the rest of the experiments. In fact, we used this strategy because 1 mM was inducing a very rapid phosphorylation of eIF2 α , and the aim of our experiments was to test if lysosomal damage can induce p-eIF2 α . However, we did not consider matching the imaging kinetics. We apologise for any confusion caused by this oversight. We now realise that this discrepancy may have led to confusion, and we have now included the data with LLOMe 1mM (that will induce lysosomal damage faster than 250 μ m). Notably, as depicted in Extended Data 2, exposure to 1mM LLOMe induces a rapid increase in p-eIF2 α levels within minutes, consistent with our imaging data.

The claimed biomolecular condensate function of SG plug is based on in vitro modeling of this using the plant protein glycinin. If and how this relates to endogenous mammalian proteins in the SGs is unclear and this preliminary observation should be extended by reconstituting SG-mediated plugging of membrane leak using G3BP1 and other mammalian SG components (Freibaum et al JCB 2021; 220(3): e202009079).

We thank the reviewer for this suggestion. The approach used in Freibaum et al., combines cellular lysates with purified protein to form G3BP1 granules. Unfortunately, this model introduces several variables (unknown components present in the lysate, detergent contamination from the lysis buffer which would rupture the vesicles, etc.) that make it incompatible with a controlled and standardised in vitro GUV system. Therefore, following the reviewer suggestion, we have reconstituted SG in vitro following Guillén-Boixet et al. Cell 2020, (10.1016/j.cell.2020.03.049) and conducted a new set of experiments that we understand has greatly benefited our study. We have now updated Figure 2 (and

Extended Data 3) showing that G3BP1 triggered condensation by poly(A) RNA at pH 5 can plug and stabilise damaged membranes. We believe these results substantially strengthen the hypothesis of our work, and further suggest that damage stabilisation by condensates might be a shared property for condensates formed at pore sites. We have now discussed this further in the manuscript, and we believe this point is now clearer thanks to the reviewer feedback.

In the previous version of this work, we considered convenient to use glycinin because it is a robust model protein for phase separation since its phase diagram is known for different conditions including NaCl concentration, pH and temperature (10.1021/acsmacrolett.0c00709). In addition, glycinin interaction with membranes has been reported in detail (10.1038/s41467-023-37955-2). This allowed us to explore different triggers for phase separation upon damage as we showed before. It is important to note that, it has been demonstrated that irrespective of the chemical composition of condensates (i.e. mammalian vs plant proteins, polymers, nucleic acids), they can undergo membrane wetting transitions depending on the milieu conditions (10.1038/s41467-023-37955-2); i.e. the interaction with the membrane can be easily tuned. Then, from a biophysical point of view, we will expect that the phenomenon of plugging and stabilisation of the membrane is general for any condensate forming at the site of damage, and that can interact with (wet) the membrane. We have now briefly discussed this in the main text, and we believe this expands the scope of our work and opens interesting new areas that require further research.

To extend the *in vivo* relevance of the findings, authors allude to the inherited deficiency in patients of the SG protein ZNFX1. To support this speculation there is a need to examine if this protein is in the G3BP1 containing SG plugs that form at the damaged membrane or if its absence impacts the SG plug formation/lysosome membrane repair.

We thank the reviewer for raising this important point. We have now added new data showing that ZNFX1 is present in SG induced after lysosomal damage, and we also found ZNFX1-positive granules in *Mtb* WT but not *Mtb* Δ RD1- infected iPSDM (new Fig. 4).

In figure 3, acidification of lysosomes is used as a readout of lysosomal membrane integrity. However, lack of G3BP1 will impair mTOR and other stress response of cells independent of SG condensate formation which in turn will impair the ability of these stressed cells to handle ion and pH homeostasis. Thus, this does not offer unambiguous evidence for the claim presented. A direct readout such as leakage of larger dyes / dextran etc. from lysosomes should be examined for assessing lysosomal membrane integrity.

We thank the reviewer for this comment, and we agree this is an important point. In the original version of this manuscript, we had already shown that the lysosomal volume and lysosomal proteolytic activity of G3BP^{nf} iPSDM was not different from iPSDM WT, suggesting no differences in basal lysosomal function for G3BP^{nf} iPSDM. In agreement with these results, we have now quantified the basal LysoTracker fluorescence levels in iPSDM WT and G3BP^{nf} iPSDM and found no difference (Extended Data 6g). We also do not expect the data from Prentzell et al., (previous ref43) will necessarily be extensive to any biological model since these processes are highly dependent on specific protein-protein interactions (10.1016/j.cell.2017.12.032). In fact, it has been recently shown that mTORC1-TFEB signalling pathway is unaffected in G3BP2-depleted human neurons (10.1016/j.neuron.2023.05.033), and lysosomal recovery after damage has also been observed to be independent of mTORC1 activation (10.1038/s41556-023-01125-9). Nonetheless, we agree this is a critical point in our manuscript and we now present lysotracker and dextran-chase experiment in U2OS WT and G3BP DKO cells (Extended Data 7i-q, Supplementary Video 15). In accordance with our previous observations, U2OS G3BP DKO cells present a severe impairment to recover the lysosomal population besides no difference at basal proteolytic activity, lysosomal volume or basal LysoTracker levels (Extended Data 7g, Supplementary Video 15).

Data shown in figure 4 identifies that the Δ RD1-infected cells do not form G3BP1 condensates as this strain does not lead to lysosomal damage. Nonetheless, iPSDM G3BP^{nf} cells show similar fold increase in infection by Δ RD1 as WT bacteria (Fig. 4k). This argues in favor of a non-SG plug formation role of G3BP1 in affecting *Mtb* infection, and against author's conclusion that "...blockade of SG formation, critically affects the outcome of *Mtb* infection in macrophages."

We thank the reviewer for raising this important observation, also highlighted by reviewer 1. There is compelling evidence that ESX-1 (encoded in the RD1 region) is not the only factor affecting phagosome membrane damage and it is more evident now that a fraction of Mtb RD1 mutant could induce membrane damage (at very low levels and later time points) and that host factors (10.1016/j.chom.2017.04.004) or Mtb factors, such as PDIMs (10.1111/cmi.12726), could affect this. In fact, our Mtb RD1 mutant strain is PDIM positive. We agree that is important to discuss this point as it was not included in the original manuscript for space constrains. We have now added a short discussion and make clearer the difference with Mtb WT in the new Fig. 4. (See also reply to reviewer 1 regarding this point).

The data presented here has been examined in macrophages, but would be expected to be relevant to other cell types as well and should be tested.

We thank the reviewer for this observation. We have now evaluated lysosomal recovery after membrane damage in U2OS G3BP DKO cells and in knockdown HeLa cells (siRNA G3BP1/2), as well as in human blood-derived macrophages (new Extended Data 6-10).

In extended data 3, PABPC1 and G3BP1 were used interchangeably for the control (untreated) and experimental (BAF1 treated) cells, it is unclear why this was done, but it should be corrected.

We thank the reviewer for raising this observation. We inadvertently showed a panel of iPSDM treated with LLOMe, and we have now corrected with the corresponding panel using silica crystals. In the original panel we used PABPC1 because it is another well characterised SG marker, however we understand this might lead to confusion and for clarity and the purpose of the figure we selected G3BP1 and GAL-3 now. We have also added a channel using reflection microscopy to visualise the internalisation of the crystals and the surrounding damaged membranes.

Reviewer Reports on the First Revision:

Referees' comments:

Referee #1 (Remarks to the Author):

The authors have adequately addressed my concerns. The addition of new data makes me more confident in the conclusions.

Referee #2 (Remarks to the Author):

The manuscript by Bussi et al reports the interesting observation that damage to endolysosomal membranes leads to stress granule formation at the pores, which are plugged and then repaired. They make the case through in vitro experiments and simulations that the location of condensate formation is caused by the local mixing of the intra- and extraluminal content. Further the simulations suggest that wetting of the condensate on the membrane is important for the repair, and this is in agreement with the final configurations of budded vesicles inside the repaired endolysosomes. Finally, the manuscript makes the case that this mechanism is important during infection with *Mycobacterium tuberculosis*, which results in endomembrane damage.

This manuscript demonstrates a novel function of stress granules. Furthermore, it demonstrates that the fluid-like properties of the stress granules is important for this function, i.e. wetting of the membrane and therefore repair. The manuscript therefore fills a considerable gap in our knowledge on SG function. This is an exciting advancement given that SGs are amongst the best understood biomolecular condensates in terms of their formation (References 1-3 in the manuscript), but that their function is much less well understood. The current manuscript may not only provide a SG functional assay, but also makes a case for why the formation via phase separation is important for SG function. I expect that this manuscript will have wide-reaching impact in the SG, phase separation and membrane remodeling fields. The authors have satisfactorily addressed my comments, and I am impressed with the number of additional experiments they are performed during the revision, which further support their model. I support publication in Nature.

Referee #3 (Remarks to the Author):

The authors have done an excellent job adding new data, re-wording the text, and updating the figures.

While there are of course more things the authors could do and that I (or another review) may suggest, at this point I don't see any of those things changing the impact or importance of this manuscript and would recommend publication as is.

Referee #4 (Remarks to the Author):

I appreciate the efforts authors have put in revising the manuscript and addressing prior comments, which has improved the presentation, making it easier to follow the results and offering greater support for the claims. Some of the issues have remained and are listed below.

- Claims by the authors that the study by Jia et al does not show induction of SGs at the lysosome is by membrane damage is contradicted by statements in that study such as "quantitative proteomics analysis detected increased association of SG proteins with damaged lysosomes" and "NUFIP2 and G3BP1 were on the surface and not sequestered within the lumen of the lysosomes."

Thus, it is best to give credit where it is due.

- The data in revised manuscript shows over 70% of the G3BP1/GAL-3-positive events exhibit a plug pattern and 90% of Gal3+ vesicles are G3BP1+. However, to support the claim of specificity of these events at the lysosomal membrane, authors should also note how many of the total SGs formed following lysosomal injury occur at the Gal3+ sites - it appears in figure 1, that only a small proportion of total G3BP1+ events are also Gal3+, which is aligned with the study by Jia et al, showing LLOMe and other lysosomal damaging agent trigger widespread SG formation including on the lysosomes.
- Images in figure 1h do not match with the associated plot in figure 1i, as the images show timepoints from 0-30 s, while the plot only shows timepoints beyond 70 s. Related to the issue of kinetics, the slope and other characteristics of the plot showing G3BP1 accumulation kinetics in figure 1i is different from the plot shown in figure 1k.
- With the plug engulfment seen by molecular simulation in figure 2, it is worth clarifying if the internal accumulation of G3BP1 seen in figure 1 is due to ILVs or internalized plug.
- In silico and in vitro SG plug formation by Glycinin and now G3BP1, show that phase separation causes plug formation independent of eIF2a while the in cellulo data shows SG plug formation is prevented in absence of eIF2a pathway. Authors should expand on this seeming contradiction to clarify the underlying mechanism for SG formation following lysosome injury. Perhaps including a schematic that describes the sequence of events and their timing from injury to repair in these assays and how they relate to the cellular response machinery is needed.
- Statement in the introduction that the mechanism to stabilize lysosome is elusive is not accurate, as the known pathways for lysosomal membrane repair including lipids (e.g., sphingomyelin, cholesterol) and proteins (e.g., ESCRT, Gal3) also stabilize membranes in addition to repairing lysosomal. (<https://doi.org/10.1016/j.ejcb.2017.01.002>, [10.4161/cc.9.12.12052](https://doi.org/10.1038/s41594-020-0404-x), <https://doi.org/10.1038/s41594-020-0404-x>).
- The figures and labeling have improved from original submission, but issues remain –
- In CLEM images (figure 3h), the disruption sites appear to be drawn on the image. But there is no note about it in the legends, and it also obscures the data.
- The extended data 9c and 9d, the mixed font color is used to label the "U2OS G3BP DKO".

Referee #5 (Remarks to the Author):

This is a very interesting series of findings around endomembrane damage during Mtb infection.

Many of the Mtb findings are strong, specifically the findings that these stress granules associate with Mtb in an RD1 dependent manner.

The bacterial growth differences are however a bit modest and I agree with reviewer 1 that it appears that G3BP has an overall impact on Mtb growth and it appears that the G3BP impact is stronger than the RD1 impact. The use of bacterial area to quantify is a nice way to utilize their microscopy based assay however, it is quite hard to translate what appears to be maybe a 1.25-1.6 fold difference in area to a difference in bacterial burden. These data don't seem necessary to really make the strong point that is the major observation that Mtb is in association with stress granules. [Please note that in their confidential comments, the reviewer suggests removal of Fig. 4m from the manuscript.]

Author Rebuttals to First Revision:

Point by point reply

Referee #1

The authors have adequately addressed my concerns. The addition of new data makes me more confident in the conclusions.

Referee #2

The manuscript by Bussi et al reports the interesting observation that damage to endolysosomal membranes leads to stress granule formation at the pores, which are plugged and then repaired. They make the case through in vitro experiments and simulations that the location of condensate formation is caused by the local mixing of the intra- and extraluminal content. Further the simulations suggest that wetting of the condensate on the membrane is important for the repair, and this is in agreement with the final configurations of budded vesicles inside the repaired endolysosomes. Finally, the manuscript makes the case that this mechanism is important during infection with *Mycobacterium tuberculosis*, which results in endomembrane damage.

This manuscript demonstrates a novel function of stress granules. Furthermore, it demonstrates that the fluid-like properties of the stress granules is important for this function, i.e. wetting of the membrane and therefore repair. The manuscript therefore fills a considerable gap in our knowledge on SG function. This is an exciting advancement given that SGs are amongst the best understood biomolecular condensates in terms of their formation (References 1-3 in the manuscript), but that their function is much less well understood. The current manuscript may not only provide a SG functional assay, but also makes a case for why the formation via phase separation is important for SG function. I expect that this manuscript will have wide-reaching impact in the SG, phase separation and membrane remodeling fields. The authors have satisfactorily addressed my comments, and I am impressed with the number of additional experiments they are performed during the revision, which further support their model. I support publication in Nature.

Referee #3

The authors have done an excellent job adding new data, re-wording the text, and updating the figures. While there are of course more things the authors could do and that I (or another review) may suggest, at this point I don't see any of those things changing the impact or importance of this manuscript and would recommend publication as is.

We are thankful for these reviewer's insightful and constructive feedback, which has enabled us to refine and strengthen our manuscript.

Referee #4

I appreciate the efforts authors have put in revising the manuscript and addressing prior comments, which has improved the presentation, making it easier to follow the results and offering greater support for the claims. Some of the issues have remained and are listed below.

Thank you for your supportive comment. We are grateful for your feedback, which helped us to strengthen the conclusions of our manuscript.

Claims by the authors that the study by Jia et al does not show induction of SGs at the lysosome is by membrane damage is contradicted by statements in that study such as “quantitative proteomics analysis detected increased association of SG proteins with damaged lysosomes” and “NUFIP2 and G3BP1 were on the surface and not sequestered within the lumen of the lysosomes.” Thus, it is best to give credit where it is due.

We thank the reviewer for raising this point. We did not intend to undermine the previous work of Jia *et al.*, and do not believe that our text at any point miscredits the work of Jia *et al.*, We would like to note that the authors do not use the term “form at”, instead they used the term “recruited” or “were on the surface”.

We believe the lack of quantitative spatiotemporal studies do not allow the authors to conclude the spatial relationship between SG and lysosomes as they also acknowledged in their manuscript. We have made every effort to be clear and accurate in our presentation of the data, and we believe that our findings provide new insights into the biological function of SGs in the context of endolysosomal damage. We consider this is reflected in our manuscript as we wrote: “Emerging evidence indicates that G3BP proteins can associate with the lysosomal membrane under homeostatic conditions (19, 20). In addition to the broad range of stimuli triggering SG formation, lysosomal damage induces SG formation (21, 22). However, it remains unclear whether this response is primarily caused by the damaged membrane itself or whether SG formation is spatially restricted to the site of damage. Importantly, the biological function of SG in the context of endolysosomal membrane damage (i.e., loss of membrane integrity through rupture or poration) has yet to be elucidated”.

The data in revised manuscript shows over 70% of the G3BP1/GAL-3-positive events exhibit a plug pattern and 90% of Gal3+ vesicles are G3BP1+. However, to support the claim of specificity of these events at the lysosomal membrane, authors should also note how many of the total SGs formed following lysosomal injury occur at the Gal3+ sites - it appears in figure 1, that only a small proportion of total G3BP1+ events are also Gal3+, which is aligned with the study by Jia et al, showing LLOMe and other lysosomal damaging agent trigger widespread SG formation including on the lysosomes.

We acknowledge Reviewer 4's comment that Figure 1 may not be entirely clear. Because of space limitations, the number of zoom-in examples we can show, and the fluorescence levels is limited. These images were chosen to avoid saturating the image, may have made it difficult to discern the percentage of SGs at GAL3+ sites.

We observed that approximately 90% of the SGs are tightly associated with GAL3+ structures during all the time points studied. This observation differs significantly from the findings of Jia et al., who did not detect many interactions between SGs and GAL3+ structures. In Figure 1, we quantified the number of GAL3+ events that are also positive for G3BP1 (rather than vice versa), as this is the specific interaction that was requested by Reviewer 3 (please see reply to reviewer 3, first round of revision). To further clarify our observations, we illustrate here some examples (for review's purpose only):

As shown in this figure, and highlighted in the zoom-in example, almost all G3BP1-positive granules (magenta) are in the proximity to GAL-3-positive (green) damage sites. We have also included several supplementary videos illustrating these G3BP1/GAL-3 interactions.

Images in figure 1h do not match with the associated plot in figure 1i, as the images show timepoints from

0-30 s, while the plot only shows timepoints beyond 70 s. Related to the issue of kinetics, the slope and other characteristics of the plot showing G3BP1 accumulation kinetics in figure 1i is different from the plot shown in figure 1k.

We thank the reviewer for this comment. As indicated, the timepoints illustrated in Figure 1h correspond to images after 2 minutes of LLOMe treatment. We accidentally mislabelled the x-axis from Figure 1i as starting after 1 minute (instead of 2 minutes), and we have now corrected it. Although the 5-second intervals are quantified, the time-lapse has a temporal resolution of 1 second, which differs from the 20s interval sequence shown in Fig. 1k. The temporal resolution difference explains why the slope is more pronounced in Fig. 1k.

With the plug engulfment seen by molecular simulation in figure 2, it is worth clarifying if the internal accumulation of G3BP1 seen in figure 1 is due to ILVs or internalized plug.

We thank the reviewer for raising this important point. We believe that future work on condensate-membrane-bound organelle interactions will shed light on this intriguing concept. *In vitro* studies have shown that membrane wetting can lead to condensate engulfment, depending on the membrane-condensate interaction for different condensates and membrane compositions (10.1021/jacs.2c04096, 10.1038/s41467-023-37955-2), without the involvement of active processes.

Our data indicate that both phenomena may be occurring. Our live-cell imaging data and Extended data 2i-k suggest that condensates are internalised via ILVs after wetting. However, if extensive damage occurs, such as if the pore size is too large or if several pores occur simultaneously, phase separation could occur, completely clogging the lysosomes, as appears to be the case in Fig. 1f,j. This would prevent additional damage by containing the hydrolytic enzymes and preventing them from entering the cytoplasm, for example. Due to space constraints, we did not expand on this discussion, but we believe that further studies are needed.

In silico and *in vitro* SG plug formation by Glycinin and now G3BP1, show that phase separation causes plug formation independent of eIF2a while the *in cellulo* data shows SG plug formation is prevented in absence of eIF2a pathway. Authors should expand on this seeming contradiction to clarify the underlying mechanism for SG formation following lysosome injury. Perhaps including a schematic that describes the sequence of events and their timing from injury to repair in these assays and how they relate to the cellular response machinery is needed.

We thank the reviewer for their comment. The main aim of the *in vitro* (and *in silico*) experiments was to demonstrate that condensate formation can occur at the pore site and stabilise the damaged vesicles. To this end, we used a minimal model system: giant vesicles composed of one or two lipid components and condensates made of G3BP1/poly(A) RNA or Glycinin. Our results show that membrane damage can trigger condensate formation at the pore site, either by reducing the pH for G3BP1 condensates or by changing the salinity for Glycinin. These findings strongly support the new function for condensates that we are proposing: condensates can nucleate at the pore site, stabilising damaged membranes. Moreover, we have shown that there can be multiple triggers for the onset of phase separation upon poration. From a biophysical perspective, our data suggest that the mechanism we have uncovered here may be universal for different condensate or membrane compositions, depending on whether the condensate can wet (has an affinity for) the membrane. We cannot rule out the possibility that other components (missing either in the membrane or in the condensation process in the *in vitro* or *in silico* studies) may play a role, as evidenced by the observed inhibition by eIF2a. In this regard, we see no contradiction with the *in cellulo* data, since at the cellular level the mechanism will be more complex, involving several protein networks and interactions (10.1016/j.bbamcr.2020.118876, 10.1016/j.cell.2020.03.049) that will contribute to the effective nucleation, interaction, and wetting of membranes by SG. Given the important relevance of lysosomal damage in a wide range of diseases (10.1016/j.tcb.2023.01.001), we believe that addressing the specific molecular machinery regulating SG formation, composition, and wetting during lysosomal damage represents an exciting research avenue that this study opens up. We believe these studies are beyond the scope of this study.

Statement in the introduction that the mechanism to stabilize lysosome is elusive is not accurate, as the known pathways for lysosomal membrane repair including lipids (e.g., sphingomyelin, cholesterol) and proteins (e.g., ESCRT, Gal3) also stabilize membranes in addition to repairing lysosomal. (<https://doi.org/10.1016/j.ejcb.2017.01.002>, 10.4161/cc.9.12.12052, <https://doi.org/10.1038/s41594-020-0404-x>).

We thank the reviewer for this comment. We believe there may be a misunderstanding due to the wording and specific context we are referring to. In that sentence in the introduction, we are referring to ruptured membrane stabilisation in the context of vesicle (*in vitro*) and lysosomal membrane damage (*in cellulo*), and we are not generalising to any membrane stabilisation process or considering it equivalent to the term "lysosome stabilisation." We understand the concept of "lysosome stabilisation" (as used in the first two manuscripts mentioned by the reviewer) as a more general concept encompassing the lysosome as a whole and including broader targets, such as the impact on lysosomal proteolytic activity, and not exclusively to the membrane. For example, in the study by Eriksson et al. (first manuscript), the authors observed that increased lysosomal cholesterol accumulation (which they correlate with increased lysosomal stability) decreases lysosomal cathepsin activity compared to control cells.

We did not intend to disregard previous important work uncovering the role of Hsp70 in lysosomal function. However, in Kirkegaard et al. (10.1038/nature08710, and second manuscript), the authors highlight that the function they uncovered may not be related to an effect on the membrane, as they state: "*Taken together, our data indicate that the Hsp70–BMP interaction stabilizes lysosomes by a mechanism involving the regulation of sphingomyelin metabolism rather than direct physical stabilization of the membrane.*"

We are not sure how our work relates to the third study mentioned by the reviewer, which shows the role of ESCRT during membrane fission. The manuscripts discussed above do not address the dynamics of ruptured membrane stabilisation or the connection with the lysosomal repair machinery. What we show here is that, *in cellulo*, SG formation occurs rapidly after lysosomal damage induction, and we propose that this is a new mechanism of ruptured membrane stabilisation by a molecular condensate that prevents lysosomal leakage and allows for their efficient repair. Although we believe the wording, we use reflects the current literature to the best of our knowledge, we changed "remain to be identified" to "poorly understood" to avoid any misunderstandings.

If a pore forms in a membrane, there are at least two possible outcomes: either the pore is resealed, which can be the case for neutral membranes, or if the membrane is charged (as in the case of the lysosomal membrane), the lower edge tension will increase the lifetime of the pore, leading to membrane burst and collapse (see Figure 2f and 10.1002/adv.202004068). In this manner, the mechanism of SG formation upon poration we uncovered here constitutes an immediate response to damage that allows to stabilise the pores, preventing collapse.

In CLEM images (figure 3h), the disruption sites appear to be drawn on the image. But there is no note about it in the legends, and it also obscures the data.

We thank the reviewer for raising this point. We have indicated the areas of membrane disruption with a dashed line, as we believe this would help a broader audience of this journal to better understand the figure. Following the reviewer's feedback, we have now changed the dashed lines for asterisks. We hope the image is clearer and we apologise if this was not clearly stated in the figure legend. We have corrected it now.

The extended data 9c and 9d, the mixed font color is used to label the "U2OS G3BP DKO".

Thanks, we have now corrected U2OS G3BP DKO and labelled all in red.

Referee #5

This is a very interesting series of findings around endomembrane damage during Mtb infection. Many of the Mtb findings are strong, specifically the findings that these stress granules associate with Mtb in an RD1 dependent manner.

We are grateful to the reviewer for their encouraging comments.

The bacterial growth differences are however a bit modest and I agree with reviewer 1 that it appears that G3BP has an overall impact on Mtb growth and it appears that the G3BP impact is stronger than the RD1 impact. The use of bacterial area to quantify is a nice way to utilize their microscopy based assay however, it is quite hard to translate what appears to be maybe a 1.25-1.6 fold difference in area to a difference in bacterial burden. These data don't seem necessary to really make the strong point that is the major observation that Mtb is in association with stress granules. [Please note that in their confidential comments, the reviewer suggests removal of Fig. 4m from the manuscript.]

We are grateful to the reviewer for raising this point. As we mentioned to Reviewer 1 in our previous reply, *Mycobacterium tuberculosis* lacking RD1 (Mtb DRD1) is severely restricted in its ability to induce endosomal damage. However, the bacterium still possesses the lipid PDIM, which enhances macrophage phagosomal permeabilization and membrane damage (10.1111/cmi.12726). This means that in long-term experiments, macrophages infected with Mtb DRD1 will still exhibit some degree of endolysosomal damage. These characteristics explain why we observed the main differences in Mtb DRD1 growth between WT and G3BP1/2nf- macrophages after 72 hours of culture but not at earlier time points (in contrast with Mtb WT). We have included the growth curve with the earlier time points, where the growth difference between Mtb WT and Mtb DRD1 is more clearly illustrated and evident already after 48 hours of infection.

We would like to emphasise that, considering the slow-growing nature of Mtb, even small increases in the growth rate can have a significant impact on host cell viability in the long term. As we are working in conditions where macrophages are viable (10.1038/s41564-023-01335-9, 10.1038/s41467-022-34632-8), we consider differences of almost 50% (such as the one observed for Mtb WT in G3BP-nf macrophages) to be very significant. We believe that future work evaluating the role of SG in Mtb infection *in vivo* will provide further insights of clinical relevance, and we are currently exploring this possibility.

Due to space constraints and the focus of this work, we did not expand further on this finding. However, we acknowledge the reviewer's point and, in line with the most recent findings, we state in the text that Mtb DRD1 is a strain that is "severely restricted" in its ability to induce endomembrane damage, but not "unable." We have also updated the text to mention the role of the bacterial lipids.

The final version of the manuscript was seen by the referees